# Targeting histone H2B acetylated enhanceosomes via p300/CBP degradation in prostate cancer

Jie Luo[1,2,12], Zhixiang Chen [3,4,5,12], Yuanyuan Qiao [1,2,3,12], Jean Ching-Yi Tien[1,2], Eleanor Young [1], Rahul Mannan [1,2], Somnath Mahapatra [1,2], Rupam Bhattacharyya[1,2], Lanbo Xiao[1,2], Tongchen He[1,2], Sanjana Eyunni[1,2,6], Yuping Zhang[1,2], Yang Zheng [1,2], Fengyun Su[1,2], Xuhong Cao[1,2,7], Rui Wang [1,2], Yunhui Cheng[1,2], Rithvik Seri[1], James George [1], Miriam Shahine[1], Stephanie J. Miner [1,2], Matthew G. Rees [8], Melissa M. Ronan [8], Jennifer A. Roth [8], Ulka Vaishampayan[3,4], Mi Wang[3,4,9,10], Shaomeng Wang [3,4,9,10,13] ✉, Abhijit Parolia[1,2,3,11,13] ✉ & Arul M. Chinnaiyan [1,2,3,7,11,13] ✉

Prostate cancer is driven by oncogenic transcription factor enhanceosomes comprising chromatin and epigenetic regulators. The lysine acetyltransferases p300 and CREB-binding protein (CBP) are key cofactors that activate enhancers through histone acetylation. Here we identify p300/CBP-mediated multisite histone H2B N-terminal acetylation (H2BNTac) as a defining feature of oncogenic enhanceosomes in androgen receptor (AR)-positive prostate cancer. p300/CBP are essential for AR and ETS transcription factor ERG transcriptional activity, and their dual degradation eliminates H2BNTac and histone H3 lysine 27 acetylation at hyperactive enhancers, leading to stronger suppression of oncogenic transcription than targeting either paralog or bromodomain alone. Cytotoxicity profiling across >900 cell lines revealed that tumors with high H2BNTac, including AR-positive prostate cancer, are selectively dependent on p300/CBP. In preclinical models, systemic p300/CBP degradation inhibited tumor growth, synergized with AR antagonists and showed no evident toxicity. These findings position H2BNTac as an epigenetic marker of enhancer addiction and establish dual p300/CBP degradation as a promising therapeutic strategy for enhancer-driven cancers.

Cancer is characterized by a major reconfiguration of the epigenetic landscape[1,2]. A key change noted in metastatic castration-resistant prostate cancer (CRPC) is an escalated dependence on an aberrant enhancer landscape[3,4]. Oncogenic transcription factors (TFs), such as the androgen receptor (AR), cooperate with epigenetic coregulators at cancer-specific enhancer complexes, or neo-enhanceosomes, to drive oncogene expression by promoter looping[4–9]. Earlier research demonstrated that degradation of the ATPase subunits (SMARCA2/4) of the switch/sucrose non-fermentable (SWI/SNF) complex by a proteolysis-targeting chimera (PROTAC) compacts enhancers, blocks TF access and inhibits CRPC growth[4].

In addition to nucleosome remodeling, histone modifications are essential for enhancer activation[10]. The acetyltransferases p300 and CREB-binding protein (CBP) are key coactivators in metastatic CRPC[11,12], modifying histones and transcriptional regulators, scaffolding enhancer complexes and promoting transcription[13–15]. They function as

coregulators of AR[16–20], and their inhibition impairs AR-positive prostate cancer growth[17,21,22]. Two major types of inhibitors have been developed: bromodomain inhibitors, which block recognition of acetylated lysines, and histone acetyltransferase (HAT) inhibitors, which suppress acetyltransferase activity[17,21,22]. Both domains are required for complete p300/CBP activities, and dual inhibition has shown enhanced efficacy[23]. However, bromodomain inhibitors act on a narrow set of substrates[12,24], and HAT inhibitors fail to block non-enzymatic functions[25]. Currently, only one bromodomain inhibitor, CCS1477, is in clinical trials[17].

Here, we demonstrate that p300 functions as the essential HAT, defining active oncogenic enhanceosomes in prostate cancer by hyperacetylating the N terminus of histone H2B (H2BNT), thereby promoting oncogenic transcriptional programs. Histone H2B N-terminal acetylation (H2BNTac) is markedly elevated in prostate cancer lesions relative to normal tissue. Using CBPD-409 (ref. 26), a potent and orally bioavailable PROTAC designed to degrade both p300 and CBP, we achieve robust and selective suppression of enhancer-driven, AR-positive prostate cancer, surpassing the efficacy of conventional domain-specific inhibitors. This enhanced activity is attributed to more complete silencing of AR signaling and H2BNTac-dependent oncogenes. Notably, systemic administration of p300/CBP degraders was well tolerated, with no observable toxicity in preclinical models. These findings underscore the central role of p300/CBP in sustaining oncogenic enhancer activity and establish PROTAC-mediated degradation as a promising therapeutic strategy for enhancer-addicted prostate cancers.

## Results

### H2BNTac is significantly elevated in prostate cancer lesions

We examined a range of histone post-translational modifications related to transcriptional regulation in matched benign and primary prostate cancer tissues from patients[27–31] and identified a notable increase in H2BNTac in prostate cancer tissues (Extended Data Fig. 1a). H2BNTac, including histone H2B lysine 5 acetylation (H2BK5ac), histone H2B lysine 12 acetylation (H2BK12ac), histone H2B lysine 16 acetylation (H2BK16ac) and histone H2B lysine 20 acetylation (H2BK20ac), is known to mark active intergenic enhancer regions[29]. Strikingly, immunofluorescent staining of prostatectomy samples revealed a marked increase in H2BK5ac and H2BK20ac levels in keratin 8 (KRT8)-positive malignant cells relative to patient-matched adjacent benign epithelia (Fig. 1a,b). Immunofluorescence analysis in a tissue microarray of benign and tumor specimens further confirmed the marked elevation of H2BK5ac and H2BK20ac levels in prostate cancer tissues (Extended Data Fig. 1b).

Next, we performed cumulative analyses of essentiality scores for all HATs and histone deacetylases (HDACs) in AR-positive prostate cancer cell lines (VCaP and LNCaP) from DepMap (v.24Q2; https://depmap.org/portal). This analysis identified p300, which is responsible for H2BNTac[29], as the most essential gene (Extended Data Fig. 1c). By contrast, p300 is dispensable in AR-negative prostate cancer cell lines (PC3 and DU145) (Extended Data Fig. 1c). Further assessment revealed a concordant increase of p300 expression, and that of its paralog, CBP, in primary prostate cancer tissues compared with the benign epithelia (Extended Data Fig. 1d–f). H2BNTac signals also showed a strong positive correlation with p300/CBP levels in prostate cancer tissues (Extended Data Fig. 1g). H2BK5ac, H2BK20ac, histone H3 lysine 18 acetylation (H3K18ac), histone H3 lysine 27 acetylation (H3K27ac) and p300 chromatin immunoprecipitation sequencing (ChIP-seq) confirmed that the majority of H2BK5ac (82.3%) and H2BK20ac (76%) histone marks overlapped with p300 peaks in prostate cancer cells, while a smaller fraction of H3K18ac (46.7%) and H3K27ac (46.3%) sites overlapped with p300 peaks, supporting the dependency of H2BNTac on p300 (Extended Data Fig. 1h,i). Collectively, these data reveal hyperacetylation of H2BNT to be associated with p300 in prostate cancer.

### p300 is an oncogenic enhanceosome determinant in prostate cancer

p300 co-localizes with TFs to regulate gene expression[14,32,33]. We explored the interplay between p300 and AR or ETS transcription factor ERG (ERG) in VCaP cells[34]. Based on normalized read densities, we categorized ChIP-seq peaks of oncogenic TFs (AR, ERG and forkhead box A1 (FOXA1)) and transcriptional cofactors (SMARCA4, p300 and bromodomain-containing 4 (BRD4)) at non-promoter regions into quartiles (Q4 represents the top quartile). Subsequently, we assessed their presence at the binding sites of AR or ERG. As expected, the majority of the strongest AR or ERG peaks (Q4 peaks) were co-occupied by pioneer TF FOXA1 (ref. 7) and the cofactor SMARCA4 (ref. 4) (Extended Data Fig. 2a–c). Notably, unlike BRD4 (ref. 35), we found that p300 co-localized with over 60% of the strongest AR and ERG Q4 peaks (Extended Data Fig. 2a–c). Moreover, most p300 peaks overlapping with AR were high-confidence (Q3 or Q4) signals.

Next, we profiled the p300 cistrome in VCaP cells and categorized AR binding sites as shared (AR/p300 co-bound) or exclusive (AR only) (Extended Data Fig. 2d). Although AR binding intensity was similar between groups, AR/p300 co-bound sites (48% of the AR cistrome) showed higher chromatin accessibility and greater recruitment of Mediator complex subunit 1 (MED1) and BRD4, along with increased H3K27ac and H2BK20ac levels (Fig. 2a). By contrast, AR-only sites had lower accessibility and reduced cofactor binding and histone acetylation (bottom heatmaps in Fig. 2a). AR/p300 co-bound sites more frequently formed super-enhancers[36], including at recurrent oncogenic loci such as *TMPRSS2*, *AR* and *MYC* (Fig. 2b and Extended Data Fig. 2e). Notably, Hi-ChIP analysis[4] revealed that a greater proportion of AR/p300-shared enhancers looped to active promoters compared with AR-only sites (Extended Data Fig. 2f). Several key AR target genes, including *KLK2*, *TMPRSS2*, *CCND1* and *ZBTB16*, were exclusively regulated by AR/p300-shared enhancers (Supplementary Table 1). Nascent RNA sequencing (RNA-seq) confirmed that AR/p300-shared enhancers had a significantly higher transcriptional output (Extended Data Fig. 2g). Similar patterns were observed for ERG, with p300 having a critical role in regulating ERG-bound enhancers (Extended Data Fig. 2h,i).

AR and ERG cooperate at *cis*-regulatory elements to form hyperactivated enhancers in prostate cancer[37,38]. Consistently, AR/ERG shared sites showed stronger enrichment of H2BNTac and p300 than AR-only or ERG-only regions (Fig. 2c,d and Extended Data Fig. 2j). Notably, most ERG/AR co-bound nucleosomes were p300-enriched, with increased cofactor occupancy, active histone marks and chromatin accessibility (Fig. 2e,f). Together, these findings suggest that p300 serves as a key determinant in defining active oncogenic enhanceosomes in prostate cancer cells.

### p300/CBP degradation erases H2BNTac

In LNCaP cells, CRISPR-mediated p300 knockout markedly reduced histone H3 and H2B acetylation, and combined targeting of both p300 and CBP resulted in a more pronounced loss of histone acetylation (Fig. 3a). In 22Rv1 cells, inhibition of both p300 and CBP was necessary to suppress histone acetylation (Extended Data Fig. 3a), suggesting a functional compensation between the paralogs and the need to target both for complete blocking of H2BNTac in prostate cancer.

p300/CBP bromodomain inhibitor CCS1477 is currently in early-phase clinical trials; thus, we tested the efficacy of CCS1477 and another bromodomain inhibitor, GNE-049, in AR-positive prostate cancer cells[17,22]. Although both compounds reduced H3K27ac, we saw little to no decrease in the abundance of H2BNTac (Fig. 3b and Extended Data Fig. 3b), suggesting that bromodomain inhibitors lead to only a partial inhibition of p300/CBP activity. To achieve complete suppression, we developed CBPD-409, a cereblon (CRBN)-dependent PROTAC degrader based on the GNE-049 warhead[26] (Extended Data Fig. 3c). In multiple cell lines, CBPD-409

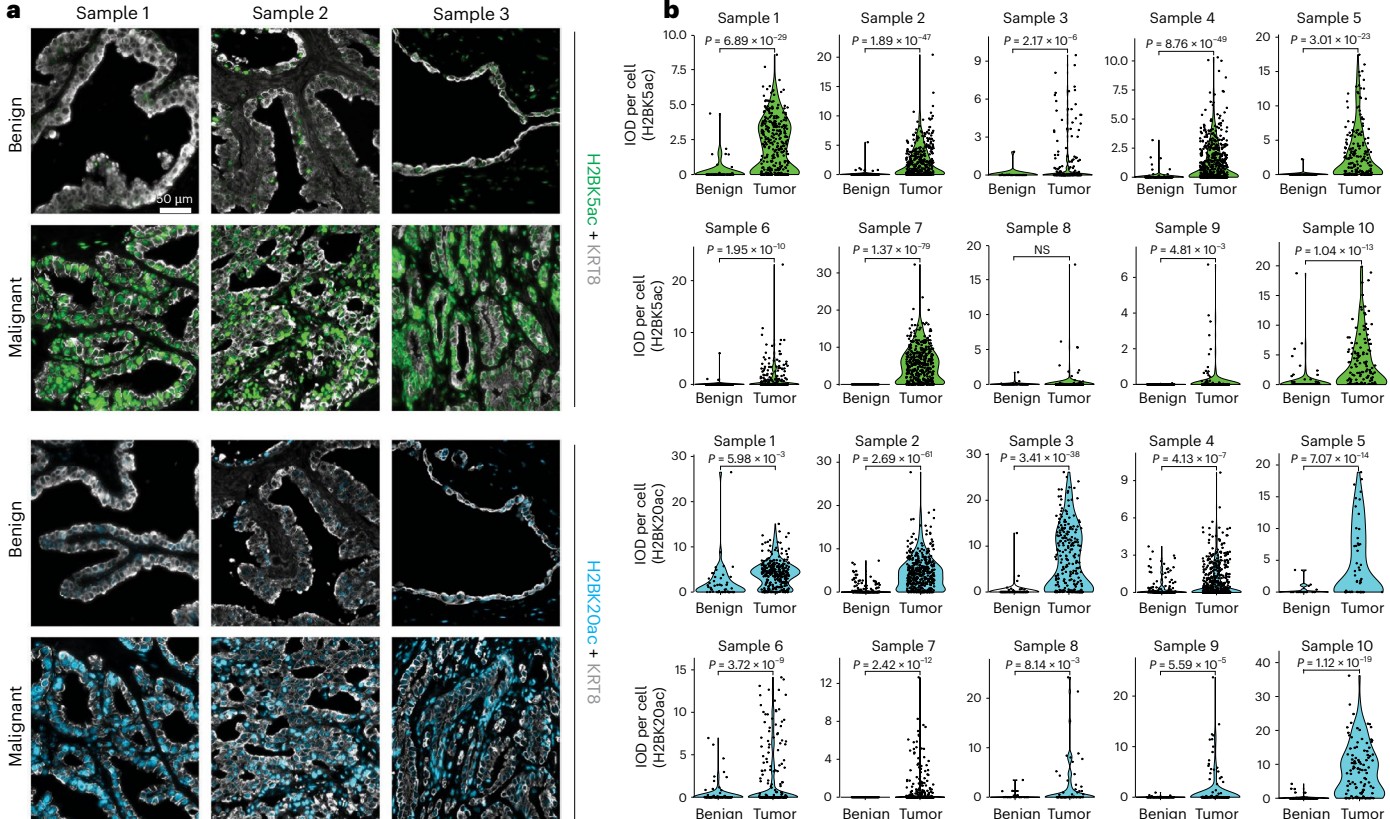

**Fig. 1 | p300/CBP-catalyzed H2BNTac is significantly elevated in prostate cancer lesions compared with normal prostate epithelia. a**, Representative multiplex immunofluorescence images of H2BK5ac/KRT8 (green and gray) and H2BK20ac/KRT8 (cyan and gray) staining in patient-matched adjacent benign and tumor tissues. Magnification, ×200. **b**. H2BK5ac and H2BK20ac IF mean intensity per cell (integrated optical density, IOD) of each case (*n* = 10) from images in **a** (two-sided *t*-test). NS, not significant.

markedly degraded both p300 and CBP proteins within 1 h (Fig. 3c and Extended Data Fig. 3d–g). Mass-spectrometry-based proteomics analyses in prostate cancer cells confirmed that CBPD-409 specifically degraded p300 and CBP (Fig. 3d and Extended Data Fig. 3h) without affecting other bromodomain proteins, including bromodomain and extraterminal protein (BET) family members[39] (Extended Data Fig. 3i,j) or known CRBN neo-substrates, including G1 to S phase transition 1 (GSPT1) and Ikaros[40] (Extended Data Fig. 3k). Competition with the free thalidomide ligand or proteasomal inhibition using carfilzomib completely blocked degradation of p300/CBP by CBPD-409 (Extended Data Fig. 3l,m). The inactive analog CBPD-409-me did not degrade its targets (Extended Data Fig. 3n,o). Notably, compared with previously reported p300/CBP degraders dCBP-1 (ref. 41) and JQAD1 (ref. 42), CBPD-409 exhibited enhanced degradation effects in prostate cancer cell lines (Extended Data Fig. 3p). Collectively, our results confirm CBPD-409 as a potent and selective tool to probe p300/CBP function.

Degradation of p300/CBP with CBPD-409 abolished H3K27ac and H2BNTac without affecting the carboxy-terminal histone H2B lysine 120 acetylation (H2BK120ac) mark (Fig. 3e and Extended Data Fig. 4a,b). By contrast, neither GNE-049 nor CCS1477 extinguished H2BNTac levels (Fig. 3e). Immunoprecipitation-based mass spectrometry analysis uncovered several acetyl marks on histone tails, including H2BN-Tac, which were rapidly lost with CBPD-409 treatment (Fig. 3f and Extended Data Fig. 4c,d). ChIP-seq revealed that CBPD-409 resulted in a complete loss of H2BK5ac and H2BK20ac peaks on the chromatin, while GNE-049 showed only a modest impact on H2BK5ac and H2BK20ac (Fig. 3g and Extended Data Fig. 4e). CBPD-409 most effectively suppressed H2BNTac and H3K27ac compared with p300/CBP

HAT inhibitor A485 and other p300/CBP degraders dCBP-1 and JQAD1 (Extended Data Fig. 4f). Altogether, these findings suggest that p300/CBP retain partial catalytic function in the presence of bromodomain inhibitors and highlight the efficacy of CBPD-409 in completely extinguishing p300/CBP oncogenic histone acetylation program in prostate cancer cells.

### p300/CBP degradation disrupts oncogenic histone acetylation

Although both CBPD-409 and GNE-049 triggered a similar decrease in overall abundance of H3K27ac histone modifications (Extended Data Fig. 5a), motif analyses revealed that CBPD-409 preferentially abolished the H3K27ac peaks at binding sites of prostate cancer-specific TFs (Fig. 3h). Consistent with this observation, CBPD-409 triggered an almost complete loss of H3K27ac and H2BNTac at AR and ERG enhancers, particularly AR/ERG co-bound hyperactivated enhancers (Fig. 3i and Extended Data Fig. 5b,c). On the other hand, the bromodomain inhibitor GNE-049 modestly suppressed H3K27ac and H2BNTac at AR and ERG enhanceosomes (Fig. 3i and Extended Data Fig. 5b,c). CBPD-409 diminished H2BK20ac at enhancers that are co-bound by ERG/AR/p300 as well as those solely bound by p300, suggesting that CBPD-409 broadly disrupts H2BNTac-marked enhancers (Extended Data Fig. 5d). Concordant findings were observed at AR super-enhancers, where CBPD-409 more effectively repressed H3K27ac and H2BNTac levels compared with GNE-049 (Extended Data Fig. 5e). To confirm our findings in patients with prostate cancer, we leveraged AR cistrome data from patient tissues to distinguish CRPC-specific AR neo-enhancers from normal AR enhancers[5,6] (Extended Data Fig. 5f). We then analyzed the p300-binding status at

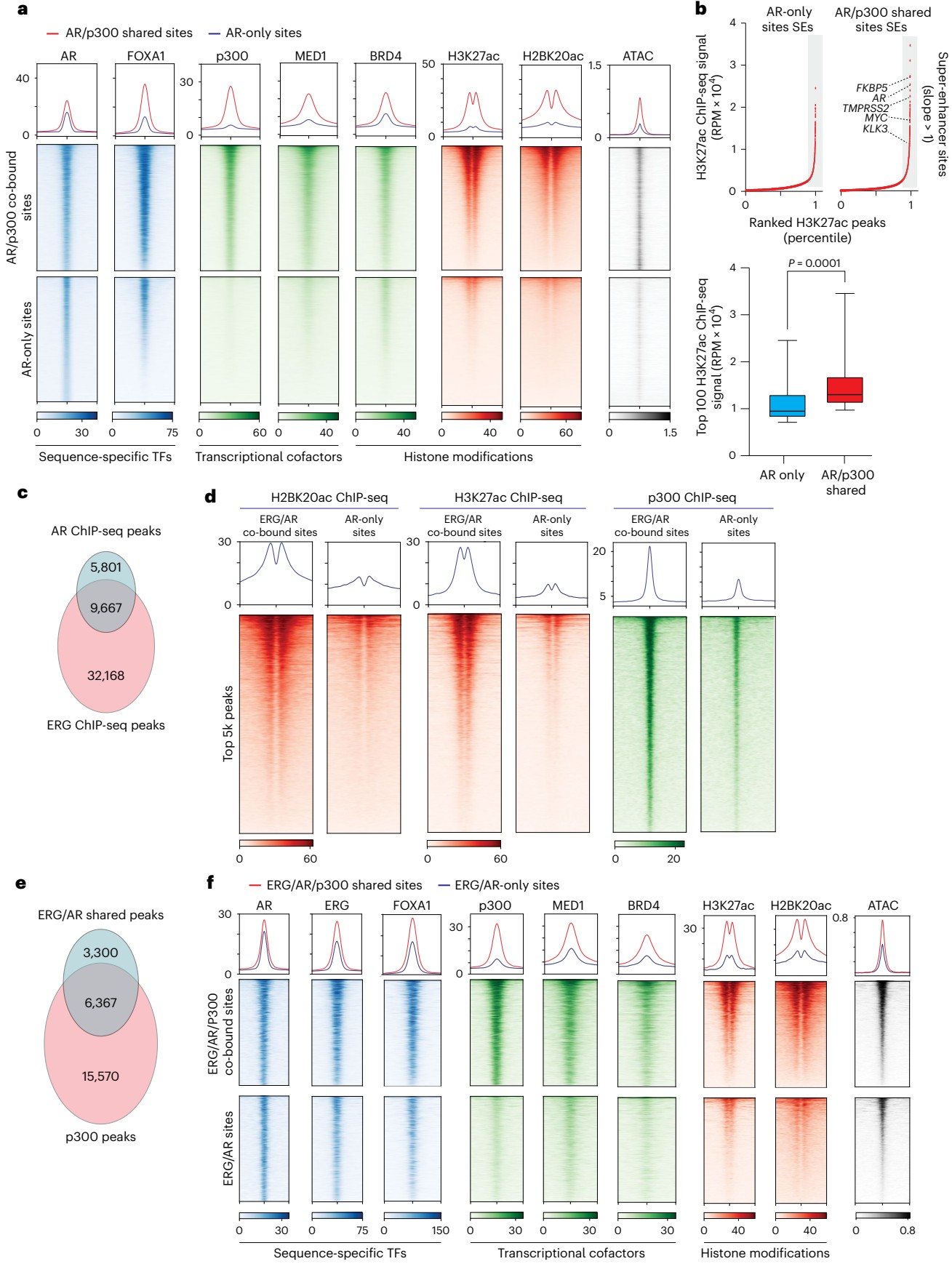

**Fig. 2 | p300 is a determinant cofactor of the oncogenic enhanceosome in prostate cancer. a**, ChIP-seq and ATAC-seq read-density heatmaps at AR/p300 co-bound and AR-only binding sites in VCaP cells. TFs, transcriptional cofactors and respective histone modifications are indicated. **b**, Ranked ordered plot of H3K27ac ChIP-seq signal (reads per million (RPM) × 10^4) of super-enhancers (SEs) on p300/AR co-bound sites and AR-only sites in VCaP cells identified by HOMER. Bottom: box plot depicting the H3K27ac ChIP-seq signals (RPM × 10^4) of super-enhancers comprising either AR/p300 co-bound or AR-only sites (two-sided *t*-test, *n* = 100 binding sites). Box plot: center line, median; box, 25th–75th

percentiles; whiskers, ±1.5× interquartile range. **c**, Venn diagram illustrating the overlap of genome-wide AR and ERG ChIP-seq peaks at non-promoter regions in VCaP cells. **d**, ChIP-seq read-density heatmaps of indicated histone marks and p300 at ERG/AR co-bound and AR-only binding sites in VCaP cells. **e**, Venn diagram illustrating the overlap of genome-wide ERG/AR and p300 ChIP-seq peaks at non-promoter regions in VCaP cells. **f**, ChIP-seq and ATAC-seq read-density heatmaps at ERG/AR/p300 co-bound and ERG/AR-only binding sites in VCaP cells. TFs, transcriptional cofactors and respective histone modifications are indicated.

these sites in VCaP cells. Notably, no significant difference in p300 enrichment between AR CRPC-specific neo-enhancers and normal enhancers was observed (Extended Data Fig. 5g). However, p300/CBP degradation severely attenuated H3K27ac at CRPC-specific AR neo-enhancers, suggesting that CBPD-409 disables the oncogenic AR enhanceosome in CRPC (Extended Data Fig. 5h).

Next, we performed assay for transposase-accessible chromatin sequencing (ATAC-seq) to explore the effects of p300/CBP degradation on chromatin accessibility. In contrast to the SMARCA2/4 degrader AU-15330 (ref. 4), CBPD-409 did not alter chromatin accessibility (Extended Data Fig. 5i). We also detected no change in the chromatin binding of AR or FOXA1 upon degradation of p300/CBP (Extended Data Fig. 5i), suggesting that p300/CBP degradation does not alter transcription factor access to chromatin. By contrast, CBPD-409 treatment significantly suppressed cofactor BRD4 (ref. 35) recruitment to AR enhancers (Extended Data Fig. 5j). Additionally, co-immunoprecipitation assays revealed that CBPD-409 treatment markedly increased the interaction between AR and HDAC3, a core component of the nuclear receptor corepressor complex[43,44], suggesting that degradation of p300/CBP not only disrupts coactivator recruitment, but also enhances corepressor loading at AR enhanceosomes (Extended Data Fig. 5k). Altogether, these findings suggest that p300/CBP degradation rapidly depletes histone acetylation and supports a hierarchical assembly of the AR enhancer complex in a p300-dependent manner.

**p300/CBP degrader eliminates oncogenic transcription**

Gene set enrichment analysis (GSEA) based on RNA-seq data revealed that p300/CBP degradation suppressed androgen response and proliferation-related pathways (Fig. 4a and Extended Data Fig. 6a,b). CRISPR-mediated knockout of both p300 and CBP in LNCaP cells phenocopied the PROTAC degrader in significantly suppressing AR and MYC target genes (Extended Data Fig. 6c). Degradation of p300/CBP also significantly reduced the protein levels of AR, prostate-specific antigen (PSA) and MYC in a time-dependent manner (Fig. 4b). *MYC* and *NKX3-1* (an AR target gene) transcripts were rapidly repressed by CBPD-409 (Extended Data Fig. 6d), which is in line with our proteomics data (Fig. 3d). Notably, the inhibitory effects of CBPD-409 were more pronounced than the bromodomain inhibitors or HAT inhibitor (Fig. 4c and

Extended Data Fig. 6e). A 1 h pre-treatment with CBPD-409 completely attenuated ligand-induced transcriptional activity of AR in prostate cancer cells (Fig. 4d). This was paralleled by a marked reduction in loading of RNA polymerase II (Pol II) specifically at the promoters of AR up-regulated genes in CBPD-409 treated cells (Extended Data Fig. 6f,g). In nascent RNA-seq data, we discovered hallmark AR-regulated transcripts among the topmost significantly downregulated genes (Fig. 4e and Extended Data Fig. 6h). A large proportion of the downregulated nascent transcripts comprising enhancer RNAs also templated from AR binding *cis*-regulatory elements (Fig. 4e). In summary, p300/CBP degradation triggers a rapid loss of acetylation marks at AR enhancers and impairs the subsequent transcription of enhancer elements as well as their distal target genes.

**p300/CBP degradation blocks oncogenic gene programs**

Over 45% of the genes downregulated by CBPD-409 were not comparably suppressed by bromodomain inhibitors (Extended Data Fig. 7a), with p300/CBP degradation showing a stronger inhibition of gene expression (Fig. 4f and Extended Data Fig. 7b). GSEA revealed unique CBPD-409 repressed genes to be associated with growth-associated signaling pathways in prostate cancer cells (Fig. 4g,h and Extended Data Fig. 7c,d). Among genes uniquely downregulated by CBPD-409, we identified *NKX3-1*, *CITED2* and *CCND1*, which are known for their critical roles in prostate cancer progression[45–47] (Fig. 4f and Extended Data Fig. 7e). We were able to confirm the contrasting effects on these genes at the protein level (Fig. 4i and Extended Data Fig. 7f,g). Notably, co-treatment with free thalidomide, which blocks the degradation activity of CBPD-409, reversed the loss in H2BNTac as well as NKX3-1, CBP/p300-interacting transactivator with Glu/Asp-rich carboxy-terminal domain 2 (CITED2) and cyclin D1 (CCND1) expression, but still led to reduced H3K27ac and MYC repression, phenocopying the effect of bromodomain inhibitors (Fig. 4j and Extended Data Fig. 7h). These results suggest the residual H2BNT acetyltransferase activity of bromodomain-inhibited p300/CBP continues to support the expression of important cancer-promoting genes. As an orthogonal approach, treatment with the inactive analog of CBPD-409, CBPD-409-me, significantly reduced H3K27ac, yet had modest effects on H2BNTac histone modifications and failed to repress degradation-specific gene targets (Fig. 4k). Evaluation of the *NKX3-1* and *CCND1* loci confirmed a significant depletion of H2BK20ac and

**Fig. 3 | Degradation of p300/CBP, but not inhibition of their reader bromodomains, abolishes histone acetylation activity at AR enhanceosome. a**, Immunoblot analysis of p300, CBP and the indicated histone marks in LNCaP wild-type (WT), p300 knockout (KO), CBP KO and p300 KO with small interfering RNA (siCBP or siNC) cells. **b**, Immunoblot analysis of indicated histone acetylation marks in VCaP cells treated with dimethylsulfoxide (DMSO), 1 μM GNE-049 or 1 μM CCS1477 for 24 h. **c**, Immunoblot analysis of p300 and CBP in VCaP cells treated with 100 nM CBPD-409 for the indicated durations. **d**, Tandem mass tag mass spectrometry assay to evaluate the effects of CBPD-409 (100 nM, 4 h) on the proteome of VCaP cells. Data are plotted as log_2 of the fold change (FC) versus DMSO control against −log_2 of the *P* value per protein from *n* = 3 independent experiments. All *t*-tests were two-tailed, assuming equal variances. p300 and CBP proteins, highlighted in red, are indicated for emphasis. **e**, Immunoblot analysis of labeled H2B N-terminal and C-terminal acetylation in VCaP cells treated with the indicated concentration of CBPD-409, GNE-049 or

CCS1477 for 4 h. **f**, Acetyl-lysine proteomics analysis of VCaP cells. FC heatmap illustrating alterations in acetyl-lysine levels of core histone proteins (H2B, H2A, H3 and H4) in VCaP cells treated with 100 nM CBPD-409 at the indicated time points, compared with DMSO-vehicle-treated cells. Data plotted from *n* = 3 independent samples. **g**, Venn diagrams of genome-wide changes of H2BK5ac and H2BK20ac ChIP-seq peaks after CBPD-409 (100 nM, 4 h) or GNE-049 (1 μM, 4 h) treatment of VCaP cells. **h**, Analyses of %target versus %background and significance levels for HOMER motifs enriched at H3K27ac sites depleted by CBPD-409 and GNE-049. Prostate cancer (PCa)-specific TF motifs are highlighted in red; other TF motifs are highlighted in blue. *P* values were calculated using HOMER's binomial test (two-sided) and adjusted for multiple comparisons using the Benjamini–Hochberg method. **i**, ChIP-seq read-density heatmaps of H3K27ac, H2BK5ac and H2BK20ac at ERG/AR co-bound non-promoter regions in VCaP cells with 4 h of 100 nM CBPD-409 or 1 μM GNE-049 treatment.

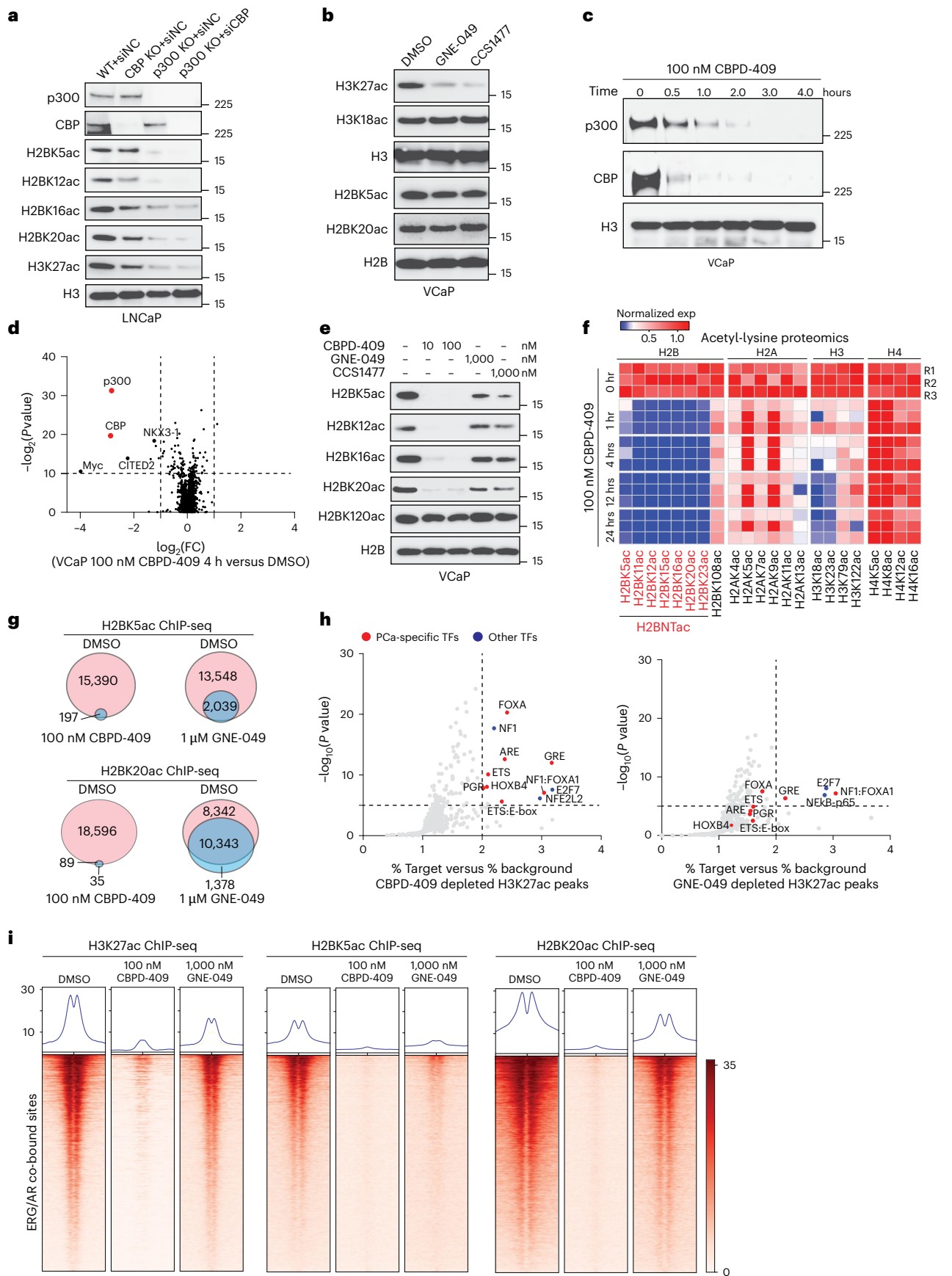

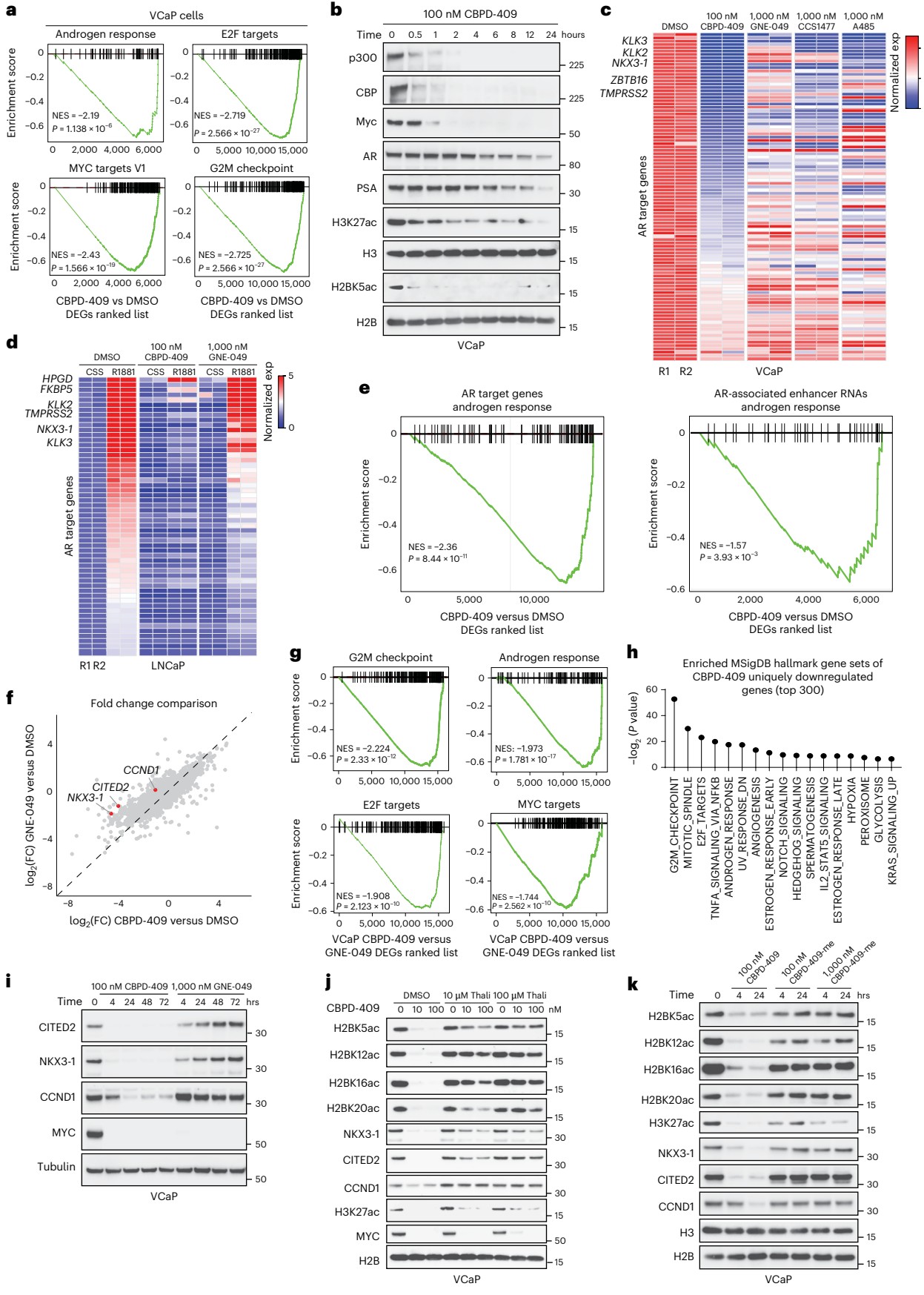

**Fig. 4 | Degradation of p300/CBP eliminates the AR-regulated oncogenic transcriptional program and leads to stronger suppression of oncogenic gene programs compared with bromodomain inhibition. a**, GSEA plots for AR, MYC, E2F and G2M checkpoint pathway-related genes from the CBPD-409-treated VCaP cells. $n = 2$ biological replicates. Statistical significance was assessed using a two-sided GSEA permutation test with adjustment for multiple comparisons. NES, normalized enrichment score; DEGs, differentially expressed genes. **b**, Immunoblot analysis of indicated proteins and histone marks in VCaP cells treated with 100 nM CBPD-409 for the indicated times. **c**, RNA-seq heatmaps for AR target genes in VCaP cells treated with 100 nM CBPD-409, 1 μM GNE-049, 1 μM CCS1477 or 1 μM A485 for 24 h. $n = 2$ biological replicates. **d**, RNA-seq heatmaps of AR target genes in LNCaP cells treated with 100 nM CBPD-409 or 1 μM GNE-049. Cells were cultured in 10% charcoal-stripped serum (CSS) medium, pre-treated for 1 h and then stimulated with 1 nM R1881 for 12 h. $n = 2$ biological replicates. **e**, GSEA plots of AR signaling pathway based on nascent RNA (left, AR target genes; right, AR-associated enhancer RNAs) expression in VCaP cells treated with 100 nM CBPD-409 for 4 h. Statistical significance was assessed using a two-sided

GSEA permutation test with adjustment for multiple comparisons. **f**, Comparison of whole-transcriptome in VCaP cells treated with 100 nM CBPD-409 and 1 μM GNE-049 for 24 h. $n = 2$ biological replicates. *CCND1*, *NKX3-1* and *CITED2* are highlighted in red. **g**, GSEA plots for G2M checkpoint, AR, E2F and MYC signaling pathways in VCaP cells: comparison of 24 h treatment with 100 nM CBPD-409 versus 1 μM GNE-049. Statistical significance was assessed using a two-sided GSEA permutation test with adjustment for multiple comparisons. **h**, Uniquely downregulated genes in CBPD-409 relative to GNE-409-treated VCaP cells analyzed for overlap with molecular signatures database (MSigDB) hallmark gene sets. Statistical significance was assessed using a two-sided hypergeometric test with adjustment for multiple comparisons. **i**, Immunoblot analysis of indicated proteins in VCaP cells treated with 100 nM CBPD-409 or 1 μM GNE-049 for the indicated times. **j**, Immunoblot analysis of labeled proteins and histone marks in VCaP cells pre-treated with different concentrations of thalidomide (Thali) for 1 h, then treated with CBPD-409 at indicated concentrations for 4 h. **k**, Immunoblot analysis of indicated proteins and histone marks in VCaP cells treated with CBPD-409 or CBPD-409-me (inactive analog) for indicated times.

Pol II loading at these genes upon treatment with CBPD-409 without affecting H3K27ac, which was not observed upon treatment with GNE-049 (Extended Data Fig. 7i). Notably, compared with other p300/CBP PROTAC degraders and the HAT domain inhibitor, CBDP-409 exhibited the most prominent effects in repressing NKX3-1, CITED2 and CCND1 levels (Extended Data Fig. 7j).

Concordant with previous reports that CITED2 promotes prostate cancer metastasis[45], Boyden chamber Matrigel invasion assays showed that degradation of p300/CBP markedly reduced the invasion ability of LNCaP and 22Rv1 cells, whereas significant changes were not observed with p300/CBP bromodomain inhibitors (Extended Data Fig. 7k). Altogether, these findings suggest that p300/CBP complexes retain their acetyltransferase activity for the H2BNT lysine residues despite the inhibition of the bromodomain, which sustains oncogenic enhancer activity in prostate cancer.

## p300/CBP degradation inhibits tumor growth without toxicity

Using CRISPR knockout and small interfering RNA techniques, we confirmed that growth of AR-positive prostate cancer cells was significantly suppressed by targeting p300/CBP (Fig. 5a). In line with the genetic approaches, treatment with CBPD-409 resulted in stronger cytotoxicity in all tested AR-positive prostate cancer cell lines relative to GNE-049 (Fig. 5b and Extended Data Fig. 8a,b). Notably, CBPD-409 showed a hook effect, with reduced cytotoxicity at the highest doses in LNCaP and 22Rv1 cells (Fig. 5b), consistent with the PROTAC mechanism, in which high concentrations hinder ternary complex formation and ubiquitination owing to target and E3 ligase saturation[48]. Despite target degradation, CBPD-409 showed no efficacy in AR-negative prostate cancer (PC3 and DU145), neuroendocrine prostate cancer (NCI-H660 and LTL-331R-CL) or normal human prostate-derived cell lines (WPMY-1, PNT2 and RWPE1; Fig. 5b

and Extended Data Fig. 8c,d). CBPD-409 also exhibited superior cytotoxicity when compared with other published p300/CBP degraders, bromodomain inhibitors or HAT domain inhibitors in AR-positive prostate cancer cell lines (Fig. 5c and Extended Data Fig. 8e). As expected, the inactive PROTAC, CBPD-409-me, had a similar cytotoxicity profile as GNE-049 (Extended Data Fig. 8f). CBPD-409 exhibited strong cytotoxicity even in GNE-049-resistant LNCaP cells (Extended Data Fig. 8g).

Impressively, assessment of the cytotoxicity of CBPD-409 in a large panel of 136 cell lines from 20 distinct lineages found AR-positive prostate cancer cells to be among the most sensitive models (Fig. 5d). This screen also revealed multiple myeloma and neuroblastoma cell lines, which show acute dependence on p300/CBP[42,49], to be markedly sensitive to CBPD-409 (Fig. 5d). By contrast, degradation of p300/CBP in HEK293FT and RWPE1 cells did not show any significant suppression of growth-related pathways, supporting the non-essential role of p300/CBP in these normal cell lines (Extended Data Fig. 8h,i).

Next, we used the PRISM multiplexed screening platform to assess the efficacy of CBPD-409 in an even larger panel of 887 cancer cell lines[50]. Among these, 6% (52 out of 887) displayed strong sensitivity (area under the dose–response curve (AUC) < 0.3), while 51% (449 out of 887) were completely resistant to CBPD-409 (AUC > 0.8), underscoring the remarkable selectivity of CBPD-409 (Extended Data Fig. 8j). Lineage enrichment analysis revealed that enhancer-driven AR-positive prostate cancer, neuroblastoma, rhabdoid tumor and acute myeloid leukemia[51-53] were among the most sensitive to CBPD-409 treatment (Fig. 5e). PRISM data integration with the DepMap Drug Repurposing dataset[54] showed the strongest correlation between CBPD-409 and the HAT inhibitor A485 (Extended Data Fig. 8k). Notably, bromodomain inhibitors showed weaker correlations, reflecting their partial

**Fig. 5 | p300/CBP degradation inhibits prostate cancer cell growth. a**, IncuCyte live-cell analysis of LNCaP and 22Rv1 cells with respective CRISPR KO and short interfering RNAs (siRNAs) indicated. siNT, non-targeting siRNA. Data are presented as means; error bars, s.d. ($n = 6$ biologically replicated wells); two-sided *t*-test. **b**, Dose–response curves and half-maximal inhibitory concentration ($IC_{50}$) of prostate cancer cells treated with CBPD-409 and GNE-049. Data are presented as means; error bars, s.d. ($n = 6$ biologically replicated wells). **c**, Dose–response curves and $IC_{50}$ of VCaP and LNCaP cells treated with p300/CBP degraders (CBPD-409, dCBP-1 and JQAD1), bromodomain inhibitor CCS1477 and HAT inhibitor A485. Data are presented as means; error bars, s.d. ($n = 6$ biologically replicated wells). **d**, Rank-order plot of $IC_{50}$ values for CBPD-409 across 136 human normal and cancer cell lines following 5 days of treatment. Models of AR-positive (AR+) prostate cancer, AR-negative (AR−) prostate cancer, non-neoplastic prostatic cells, multiple myeloma and neuroblastoma are highlighted in specific colors. The originating tissue lineages are indicated below. ERMS, embryonal rhabdomyosarcoma. NSCLC, non-small cell lung cancer;

AML, acute myeloid leukemia. **e**, Scatterplot illustrating CBPD-409 lineage enrichment from PRISM screening results ($\log_2(AUC)$), with effect size on the *x* axis indicating sensitivity and $-\log_{10}(q$ value) on the *y* axis representing statistical significance. **f**, Scatterplot depicting the correlation between CBPD-409 sensitivity (from PRISM screening) and gene dependency (from DepMap CRISPR KO project) across 887 cancer cell lines. **g**, Top: immunoblot analysis of indicated histone post-translational modifications (PTMs) in CBPD-409-sensitive and-insensitive cells. MM1S, multiple myeloma; RPMI8402, T cell acute lymphoblastic leukemia (ALL); NB1, neuroblastoma; Kelly, neuroblastoma; VCaP, AR-positive prostate cancer; 22Rv1, AR-positive prostate cancer; LNCaP, AR-positive prostate cancer; PC3, AR-negative prostate cancer; DU145, AR-negative prostate cancer; MIAPaCa-2, pancreatic cancer cell; Dang, pancreatic cancer cell; BPH1, benign prostatic hyperplasia cell; PrEC, primary benign prostatic epithelia cell; RWPE1, benign prostatic epithelia cell; HEK293FT, embryonic kidney cell. Bottom: heatmap depicting the intensities of indicated histone marks and $IC_{50}$ of CBPD-409 in individual cell lines.

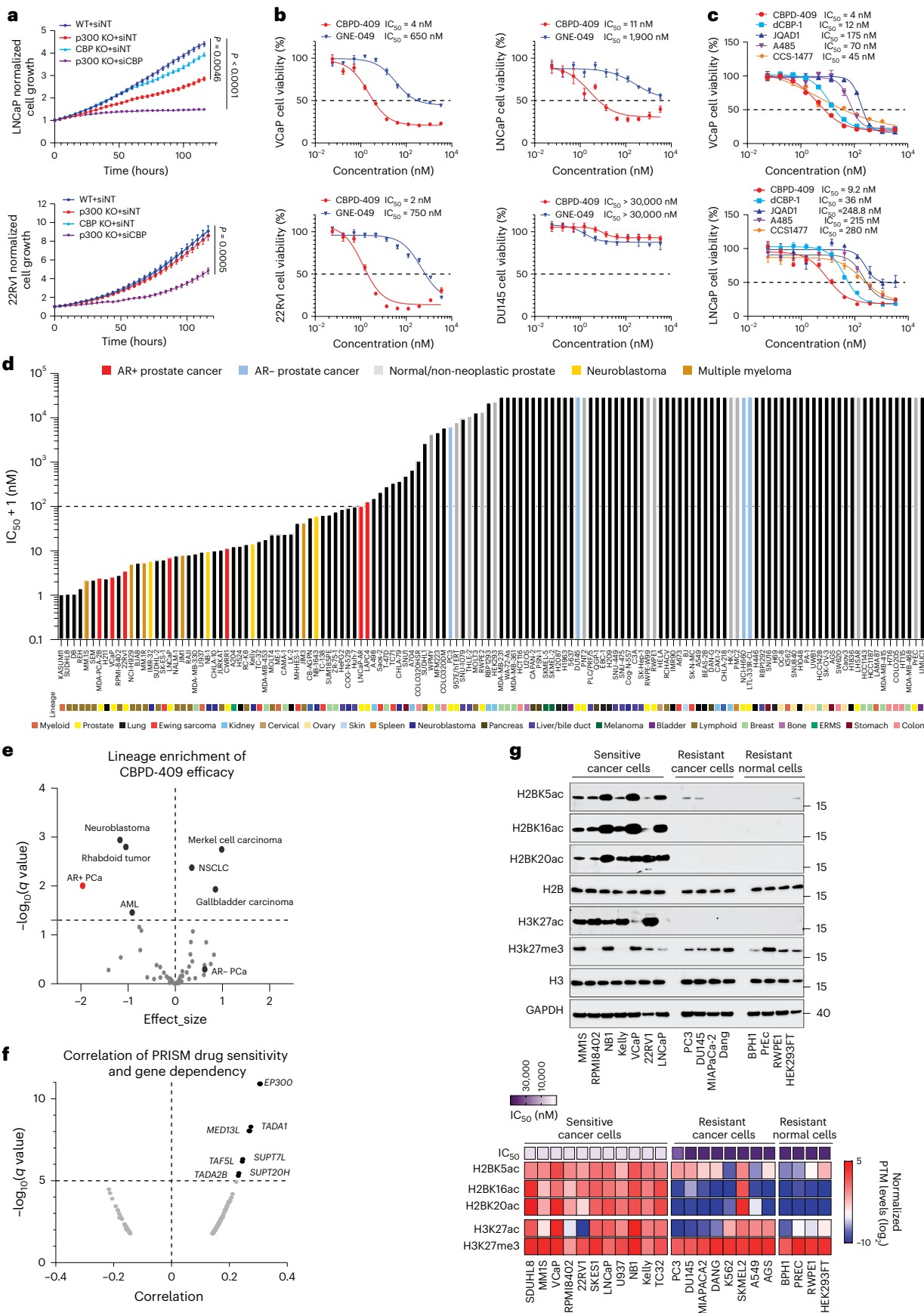

inhibition of p300/CBP activity. (Extended Data Fig. 8k). The PRISM screening also identified the strongest positive correlation between p300 dependency (CRISPR gene essentiality score) and CBPD-409 sensitivity (Fig. 5f). However, we did not observe a significant correlation between CBP dependency and CBPD-409 sensitivity, suggesting that p300 dependency is a stronger determinant of CBPD-409 efficacy. Interestingly, transcriptional adaptor 1 (*TADA1*), SPT7 like STAGA complex subunit gamma (*SUPT7L*), TATA-box binding protein associated factor 5 like (*TAF5L*), SPT20 homolog (*SUPT20H*) and transcriptional adaptor 2B (*TADA2B*), which are subunits of the Spt– Ada–Gcn5 acetyltransferase (SAGA) complex[55], showed a significant positive association with CBPD-409 sensitivity and were among the top co-dependencies with p300, underscoring the heavy reliance of enhancer-driven malignancies on histone hyperacetylation (Fig. 5f and Extended Data Fig. 8l). This was supported by elevated H2BNTac levels in CBPD-409-sensitive cells compared with the resistant normal as well as cancer cells (Fig. 5g and Extended Data Fig. 8m). A spike-in ChIP-seq assay performed in CBPD-409-sensitive cells (AR-positive VCaP) and CBPD-409-resistant cells (AR-negative DU145 and PC3) further confirmed significantly higher enrichment of H2BNTac at AR/ p300-binding enhancer sites, suggesting that H2BNTac could serve as a biomarker for predicting the efficacy of p300/CBP-targeted therapy (Extended Data Fig. 8n,o).

To assess the safety of p300/CBP degradation, we conducted toxicity studies in immune-competent CD1 mice. CBPD-409 effectively degraded p300/CBP in multiple organs (Extended Data Fig. 9a) without affecting body or organ weights (Extended Data Fig. 9b,c). Histological examination at the endpoint (day 32) revealed no toxicity in vital organs (Extended Data Fig. 9d). CBPD-409 did not affect any of the prostatic lobes, and no evidence of atrophy, hyalinization, fibrosis or emergence of any neoplastic phenomenon was identified (Extended Data Fig. 9d). Blood analyses confirmed no adverse effects on liver or kidney function or blood composition (Extended Data Fig. 9e,f). Notably, unlike p300/CBP bromodomain inhibitors or BET inhibitors[56,57], CBPD-409 did not cause thrombocytopenia or megakaryocyte loss[58] (Extended Data Fig. 9f,g) or goblet cell depletion[59–62] (Extended Data Fig. 9h). The only observed side effect was a reversible defect in germ cell maturation and testicular atrophy (Extended Data Fig. 9i). Further assessment in humanized *Crbn*[V380E/I391V] mice[63] (hCRBN mice) showed that CBPD-409 does not induce evident toxicity (Extended Data Fig. 9j–n). Despite efficient p300/CBP degradation, no toxicity was observed in vital organs, even in CD rats (Extended Data Fig. 9o,p). Liver and kidney functions, as well as platelet counts, remained unchanged upon CBPD-409 treatment (Extended Data Fig. 9q, r). Unlike the BRD4 degrader ZBC-260 (ref. 64) in human primary CD3+ T cells, natural killer cells and immortalized B cells (GM24694), CBPD-409 showed no cytotoxicity (Extended Data Fig. 9s). Altogether, our results revealed that p300/ CBP degradation selectively inhibits the growth of enhancer-driven cancer cells while sparing the normal tissues.

## p300/CBP degrader synergizes with enzalutamide in CRPC

CBPD-409 was designed for high oral bioavailability (50%) and favorable pharmacokinetics[26], distinguishing it from the previously reported p300/CBP PROTACs[41,42]. In an initial in vivo efficacy study using VCaP-derived xenograft tumors in intact severe combined immunodeficient (SCID) mice, oral administration with 3 mg kg⁻¹ of CBPD-409 significantly inhibited tumor growth (Extended Data Fig. 10a–c) without evident toxicities (Extended Data Fig. 10d–i).

Next, in a castration-resistant VCaP-CRPC model (Fig. 6a), although CBPD-409 alone significantly suppressed tumor growth, combination with enzalutamide led to regression in over 60% of animals (11 out of 18) (Fig. 6b,c and Extended Data Fig. 10j). Immunohistochemistry (IHC) staining and western blotting of tumor xenografts 5 days post treatment confirmed significant downregulation of p300/CBP, AR, PSA, MYC, Ki67, CCND1, NKX3-1, CITED2, H3K27ac and H2BK20ac (Fig. 6d and Extended Data Fig. 10k,l).

Consistent with the in vivo data, combination treatment with CBPD-409 and enzalutamide showed marked synergism in inhibiting VCaP cells (Bliss score, 23.8; Fig. 6e). CBPD-409 retained comparable efficacy between parental and an enzalutamide-resistant LNCaP cell line (Fig. 6f). In treating an enzalutamide-resistant patient-derived xenograft model, MDA-PCa-146-12 (ref. 65), the combination therapy profoundly suppressed tumor growth (Fig. 6g and Extended Data Fig. 10m–o). We further used an in-house CRPC patient-derived xenograft model, WA-74, that is inherently resistant to enzalutamide. Here, the combined treatment of CBPD-409 and enzalutamide significantly inhibited the growth of WA-74 tumors (Fig. 6h). In a more aggressive WA-74 CRPC model, the combination of CBPD-409 and enzalutamide markedly suppressed tumor growth and enhanced the survival of tumor-bearing mice (Fig. 6i and Extended Data Fig. 10p,q). Taken together, our in vitro and in vivo efficacy data demonstrate that targeting p300/CBP proteins for degradation using CBPD-409 represents a promising therapeutic strategy for advanced AR-dependent prostate cancer while exhibiting minimal on or off-target toxicity.

## Discussion

The 'histone code' underlies epigenetics, positing that DNA activity is regulated by histone post-translational modifications, commonly referred to as histone marks[66–68]. These chemical modifications create various chromatin states with specific effects on gene expression and cellular fate. For instance, at *cis*-regulatory enhancers, H3K27ac on flanking nucleosomes is associated with active transcription[28]. More recently, H2BNTac marks were shown to demarcate active enhancers and are exclusively catalyzed by p300/CBP[29]. In this study, screening a set of histone marks in prostatectomy specimens, we uncovered H2BNTac to be specifically and consistently elevated in prostate cancer lesions compared with the adjacent benign epithelia, suggesting that H2BNTac is an important signature of enhancer-addicted cancer. Importantly, H2BNTac preferentially marks prostate cancer lineage-specific transcription factor binding regions, underscoring

**Fig. 6 | p300/CBP degradation inhibits CRPC tumor growth and synergizes with enzalutamide in vivo. a**, Schematic of the CBPD-409 in vivo efficacy study using the VCaP-CRPC xenograft model. mpk, mg kg⁻¹. Tx, treatment. **b**, Tumor volume curves and weights in the VCaP-CRPC model treated with vehicle, enzalutamide (Enza) (10 mg kg⁻¹, orally, five times per week), CBPD-409 (3 mg kg⁻¹, orally, three times per week) or both. Data are means; error bars, s.e.m.; *n* = 18 per group, two-sided *t*-test. **c**, Waterfall plot illustrating the change in tumor volume after 33 days of treatment from the VCaP-CRPC study. **d**, Representative IHC images from the VCaP-CRPC xenograft study for the indicated protein. Scale bar, 50 μm. **e**, VCaP cells were treated with the noted concentrations of CBPD-409 and/or enzalutamide to evaluate drug synergism using the Bliss independence method. The average synergy score of CBPD-409 and Enza is 23.876 (>10 indicates drug synergism). Data representation includes the mean of four replicates. **f**, Dose–response curves and IC₅₀ of LNCaP parental and LNCaP enzalutamide-resistant (Enza-R) cells treated with CBPD-409. Data

are presented as means; error bars, s.d. (*n* = 6 biologically replicated wells). **g**, Tumor volume curves and weights in the MDA-PCa-146-12 CRPC patient-derived xenograft (PDX) model treated with vehicle, enzalutamide or Enza + CBPD-409. Measurements were taken biweekly. Data are means; error bars, s.e.m.; *n* = 10 (vehicle), *n* = 12 (Enza), *n* = 14 (Enza + CBPD-409). Two-sided *t*-test. **h**, Tumor volume curves in the intact WA-74 PDX model treated with vehicle, enzalutamide or Enza + CBPD-409, measured biweekly. Data are means; error bars, s.e.m.; *n* = 12 (vehicle), *n* = 12 (Enza), *n* = 14 (Enza + CBPD-409). Two-sided *t*-test. **i**, Kaplan–Meier survival plot illustrating the survival rates of mice bearing WA-74 CRPC PDX tumors, treated with vehicle, enzalutamide or Enza + CBPD-409. Survival is measured up to the point where tumor volume reaches 2,000 mm³. **j**, Mechanism of action of p300/CBP degrader versus bromodomain inhibitor in disrupting activated AR enhanceosome in prostate cancer cells. Panel **j** created with BioRender.com.

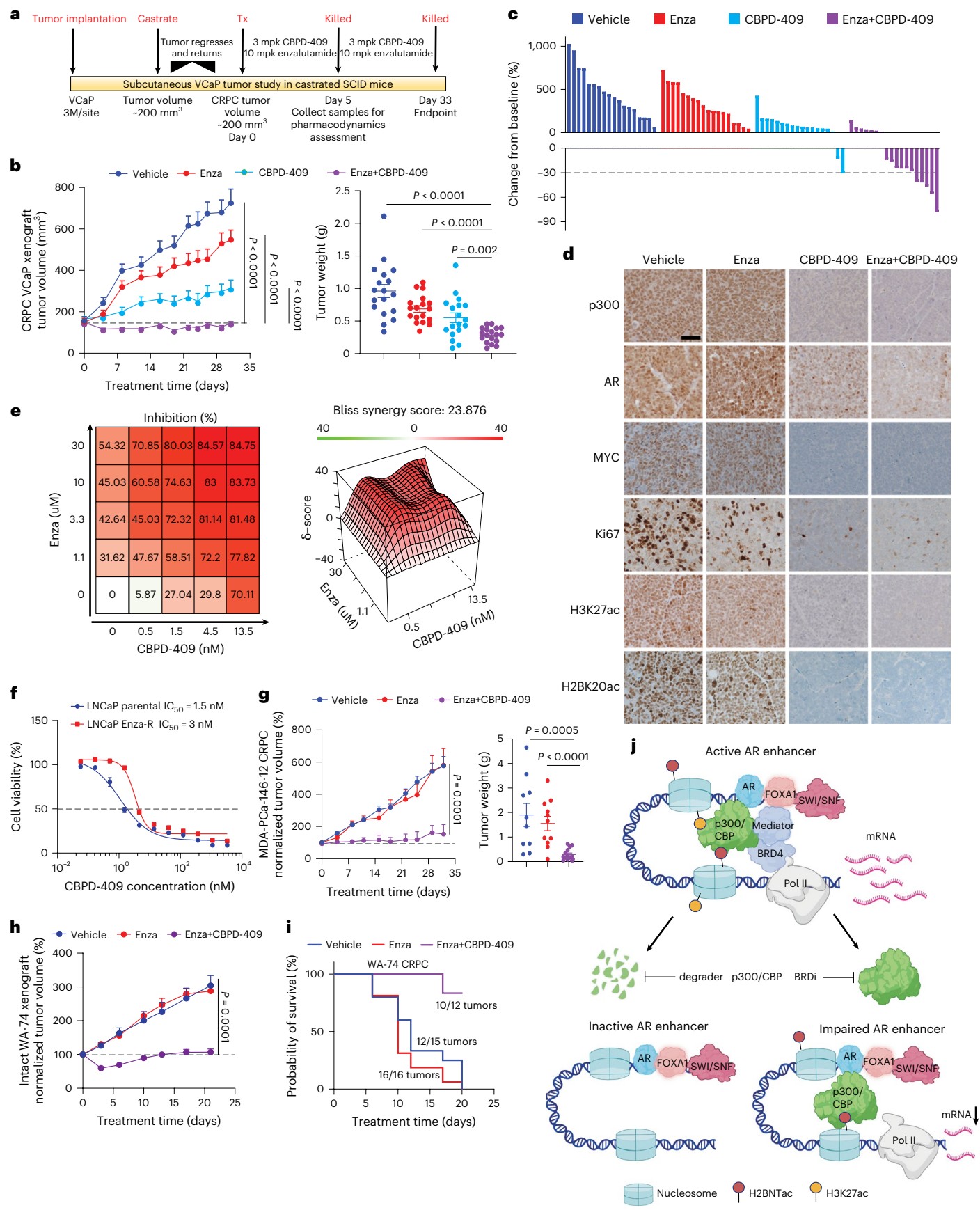

the pivotal role of H2BNTac in wiring the oncogenic transcriptome. Consistently, CRISPR-based dependency maps (DepMap) identify p300 as the most essential HAT in AR-driven prostate cancer, implying a pivotal role of p300/H2BNTac on prostate cancer cell viability.

Notably, only a subset of AR enhancers were co-occupied by p300 in prostate cancer cells, which was deterministic of stronger transcriptional activation, evidenced both by higher accessibility and subsequent recruitment of the Mediator and Pol II complexes. The genome-wide distribution profiles of H2BK5ac and H2BK20ac in prostate cancer cells confirmed a specific enrichment of H2BNTac at AR/p300 co-bound enhancers. Given recent evidence that H3K27ac loss does not affect enhancer-associated gene expression[69–71], our data suggest that H2BNTac may have a more critical role in enhancer activities. Importantly, our data reveal that H2BNTac levels correlated with CBPD-409 sensitivity, indicating its potential as a predictive biomarker for enhancer-driven tumors and p300/CBP-targeted therapies.

Current p300/CBP inhibitors exhibit distinct mechanisms of action: bromodomain inhibitors impair chromatin reader functions[17,22], HAT inhibitors reduce catalytic activities[21] and KIX domain inhibitors disrupt protein–protein interactions[72,73]. Among these, bromodomain inhibitors are known for their high specificity and drug-like properties[74], with CCS1477 being tested in clinical trials[17]. However, we show that p300/CBP deposit H2BNTac independently of their bromodomain, aligning with prior reports that bromodomain inhibition does not dislodge p300 from chromatin[24,75] and fails to extinguish its oncogenic gene programs in prostate cancer cells. Therefore, our findings support a mechanism-based rationale for the development of p300/CBP PROTACs. Using related bromodomain inhibitor and PROTAC compounds (that is, GNE-049 and CBPD-409), we demonstrate only p300/CBP degradation to completely extinguish its catalytic activity and potently inhibit prostate cancer growth. Compared with other p300/CBP degraders, including CBPD-268 and JET-209 (refs. [41,42,76,77]), CBPD-409 is the only compound with good oral bioavailability, high target selectivity and better target degradation kinetics. Unlike the p300-selective PROTAC targeting the HAT domain (JQAD1)[42], bromodomain-binding PROTACs maintain the specificity and binding affinity of bromodomain inhibitors, thus also ensuring their strong on-target effects and safety. Notably, our CRISPR knockout data show that targeting both p300 and CBP is required to inhibit 22Rv1 cell growth, underscoring the potential of dual degraders to achieve more comprehensive inhibition than p300-selective approaches.

Given that p300/CBP function as critical transcription cofactors, the concern of broad cytotoxicity has hindered the clinical development of p300/CBP targeting therapies. Here, through screening diverse cell lineages, we demonstrate that p300/CBP degraders are selectively cytotoxic, primarily in cancer driven by enhancer-binding TFs. A recent study supports this observation, showing that p300/CBP degradation selectively impairs rhabdomyosarcoma cells harboring the *PAX3–FOXO1* fusion[78]. Importantly, targeting p300/CBP does not inhibit the growth of any tested non-neoplastic cells. Furthermore, despite the effective degradation of p300/CBP, no dose-limiting toxicities were observed with prolonged CBPD-409 treatment in CD1 mice, rats or primary human cells. CBPD-409 also lacked goblet or platelet cell toxicity, unlike BRD4 inhibitors or degraders[59–62] or p300/CBP bromodomain inhibitors[56]. Transcriptomic analyses showed limited alterations in non-neoplastic cells, but marked suppression of growth-related pathways in cancer cells with CBPD-409 treatment. These findings define the lineage selectivity of p300/CBP-targeted therapy and support further safety and efficacy studies in higher-order models and early-phase clinical trials.

In summary, we identify a striking elevation of p300/CBP-catalyzed H2BNTac as a hallmark of prostate cancer, with p300 co-occupancy serving to define hyperactivated AR enhancers. Importantly, we show that conventional bromodomain inhibitors fail to deplete H2BNTac or fully suppress p300/CBP-dependent transcriptional programs. As an alternative, we developed CBPD-409, a potent, orally bioavailable PRO-TAC degrader that selectively targets p300/CBP, exhibiting favorable pharmacokinetics and preferential cytotoxicity in AR-driven prostate cancer over normal prostate and other tumor types. Mechanistically, CBPD-409 induces complete ablation of p300/CBP-mediated histone acetylation at AR enhancers and disrupts recruitment of key coactivators, including BRD4 and Pol II, thereby dismantling the enhancer-associated transcriptional machinery (Fig. 6j). These findings establish H2BNTac as a critical epigenetic dependency in prostate cancer and position p300/CBP degradation as a compelling therapeutic strategy. Given its potency, selectivity and tolerability, CBPD-409, along with other emerging p300/CBP degraders[26,76,77], represents a promising class of therapeutics for the treatment of enhancer-addicted malignancies.

## Online content

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

[1]Michigan Center for Translational Pathology, University of Michigan, Ann Arbor, MI, USA. [2]Department of Pathology, University of Michigan, Ann Arbor, MI, USA. [3]Rogel Cancer Center, University of Michigan, Ann Arbor, MI, USA. [4]Department of Internal Medicine, University of Michigan, Ann Arbor, MI, USA. [5]Interdisciplinary Research Center on Biology and Chemistry, Shanghai Institute of Organic Chemistry, Chinese Academy of Sciences, Shanghai, China. [6]Molecular and Cellular Pathology Program, University of Michigan, Ann Arbor, MI, USA. [7]Howard Hughes Medical Institute, University of Michigan, Ann Arbor, MI, USA. [8]Broad Institute of MIT and Harvard, Cambridge, MA, USA. [9]Department of Pharmacology, University of Michigan, Ann Arbor, MI, USA. [10]Department of Medicinal Chemistry, University of Michigan, Ann Arbor, MI, USA. [11]Department of Urology, University of Michigan, Ann Arbor, MI, USA. [12]These authors contributed equally: Jie Luo, Zhixiang Chen, Yuanyuan Qiao. [13]These authors jointly supervised this work: Shaomeng Wang, Abhijit Parolia, Arul M. Chinnaiyan. ✉e-mail: shaomeng@med.umich.edu; aparolia@umich.edu; arul@umich.edu

## Methods

### Ethical statement
This study complies with all relevant ethical regulations and was approved by the Institutional Review Board and Institutional Animal Care and Use Committee (IACUC) at the University of Michigan (protocol ID PRO00012191).

### Statement on use of human specimens
Formalin-fixed, paraffin-embedded prostate tumor specimens were obtained from the University of Michigan pathology archives. These archived tissues were used for IHC and multiplex immunofluorescence analyses to evaluate H2BNTac, p300 and CBP expression in tumor and adjacent normal prostate cells. Use of these de-identified specimens was approved by the University of Michigan Institutional Review Board and did not require patient consent.

### Animal procurement
Animal care and use was conducted in accordance with institutional guidelines and relevant regulatory standards. In vivo experiments were performed using male NCI SCID/NCr athymic nude mice (strain code 561, Charles River), CD1 mice (Charles River), CD rats (Charles River) and $Crbn^{V380E/I391V}$ C57BL/6 mice, which were generously provided by B. L. Ebert (Dana-Farber Cancer Institute). All experiments were initiated using male animals aged 5–8 weeks. Mice and rats were housed in pathogen-free barrier facilities under standard conditions: 12 h light–dark cycle, ambient temperature of 18–23 °C and humidity of 40–60%. The maximum tumor size permitted by the IACUC was 2.0 cm in any dimension or a total tumor volume of 2,000 mm³. This threshold was not exceeded in any animal during the course of the study.

### Cell lines and compounds
All cell lines were originally obtained from the American Type Culture Collection, DSMZ, European Collection of Authenticated Cell Cultures, Lonza or internal stock. CWR-R1 and LNCaP parental/enzalutamide-resistant (LNCaP-EnzR) cells were gifts from D. Vander Griend (University of Illinois at Chicago). All cells were genotyped every 6 months to ensure their identity at the University of Michigan Sequencing Core and tested every 2 weeks for mycoplasma contamination. Gibco RPMI 1640 + 10% FBS (ThermoFisher) was used for LNCaP, 22Rv1, CWR-R1, PC3 and DU145 cells. VCaP was grown in Gibco DMEM Glutamax + 10% FBS (ThermoFisher). CBPD-409 and CBPD-409-me were synthesized in S. Wang's lab (Supplementary Notes). GNE-049, CCS1477, A485, JQAD1, enzalutamide, carfilzomib and thalidomide were purchased from Selleck Chemicals. dCBP-1 was purchased from MedChemExpress.

### Antibodies
For immunoblotting, the following antibodies were used: p300 (1:500; Invitrogen, MA1-16608); CBP (1:500; Invitrogen, PA5-27369); AR (1:1,000; Abcam, ab133273); H2BK5ac (1:1,000; Cell Signaling Technology, 12799S); H2BK20ac (1:1,000; Cell Signaling Technology, 34156S); H2BK12ac (1:1,000; Abcam, ab40883); H2BK16ac (1:1,000; Abcam, ab177427); H3K27ac (1:1,000; Cell Signaling Technology, 8173S); H3K18ac (1:1,000; Active Motif, 39755); H3K4me1 (1:1,000; Abcam, ab8895); H3K4me3 (1:1,000; Active Motif, 39060); H3K27me3 (1:1,000; Millipore, 07-449); H2B (1:1,000; Active Motif, 39210); H3 (1:1,000; Cell Signaling Technology, 3638S); H2BK120ac (1:1,000; Active Motif, 39119); Myc (1:1,000; Cell Signaling Technology, 9402S); KLK3/PSA (1:1,000; Dako, A0562); CITED2 (1:1,000; Abcam, ab108345); NKX3-1 (1:1,000; Cell Signaling Technology, 83700S); CCND1 (1:1,000; Abcam, ab16663); FOXA1 (1:1,000; Thermo Fisher Scientific, PA5-27157); Vinculin (1:1,000; Cell Signaling Technology, 18799S); GAPDH (1:1,000; Santa Cruz Biotechnology, sc-47724); BRD2 (1:1,000; Bethyl Laboratories, A700-008); BRD3 (1:1,000; Bethyl Laboratories, A302-368A); BRD4 (1:1,000; Bethyl Laboratories, A700-004CF); GSPT1 (1:1,000; Proteintech, 28130-1-ap); Aiolos (1:1,000; Cell Signaling Technology, 15103S); Ikaros (1:1,000; Cell Signaling Technology, 14859S); HDAC3 (1:1,000; Abcam, ab137704); goat anti-rabbit IgG (H+L)-horseradish peroxidase (HRP) conjugate (1:10,000; Biorad, 1706515); and ECL peroxidase (HRP) anti-mouse IgG (1:10,000; Cytiva, NA931-100UL).

For ChIP-seq, the following antibodies were used: H3K27ac (1 µg per million cells; Diagenode, C15410196); p300 (2 µg per million cells; Abcam, ab14984); H2BK20ac (1 µg per million cells; Cell Signaling Technology, 34156S); H2BK5ac (1 µg per million cells; Cell Signaling Technology, 12799S); AR (2 µg per million cells; Millipore, 06-680); FOXA1 (2 µg per million cells; Thermo Fisher Scientific, PA5-27157); BRD4 (2 µg per million cells; Diagenode, C15410337); H3K18ac (1 µg per million cells; Active Motif, 39755); MED1 (2 µg per million cells; Active motif, 61065); ERG (2 µg per million cells; Cell Signaling Technology, 97249S); and Pol II (1 µg per million cells; Active motif, 39097).

For immunohistology and immunofluorescent staining, the following antibodies were used: p300 (1:250; Invitrogen, 33-7600); CBP (1:100; Invitrogen, PA5-27369); AR (1:4,000; Abcam, ab133273); H3K27ac (1:200; Cell Signaling Technology, 8173S); Ki67 (predilute; Ventana Medical Systems, 790-4286); H2BK20ac (1:50; Abcam, ab177430); H2BK5ac (1:250; Abcam, ab40886); CCND1 (predilute; Cell Marque, 241R-18); KRT8 (1:50; Abcam, ab53280); anti-mouse HRP (predilute; OmniMap, 760-4310); anti-rabbit HRP (predilute; OmniMap, 760-4311).

### Cell viability assay
Cells (4,000 per well) were seeded into poly-D-lysine-coated 96-well plates (Corning) in culture medium and incubated at 37 °C with 5% $CO_2$. After 24 h, cells were treated with a serial dilution of compounds, with six replicates per concentration. Following a 120 h incubation, cell viability was assessed using the CellTiter-Glo assay (Promega) according to the manufacturer's instructions. Luminescence was measured using an Infinite M1000 Pro plate reader (Tecan), and data were analyzed using GraphPad Prism (GraphPad Software).

### IncuCyte proliferation assays
A total of 4,000 cells per well were plated in clear, flat-bottom poly-D-lysine-coated 96-well plates. After 24 h of incubation, the cells were treated with different concentrations of compounds. Every 4 h, phase object confluence (percentage area) for proliferation was measured.

### Immunoblotting
Cells were lysed in RIPA buffer (ThermoFisher) supplemented with 1× Halt Protease Inhibitor Cocktail (ThermoFisher), then denatured in NuPAGE 1× LDS/reducing agent buffer (Invitrogen) by incubating at 70 °C for 15 min. Protein concentrations were determined using the Pierce BCA Protein Assay Kit (ThermoFisher). A total of 15–30 µg protein per sample was loaded onto either NuPAGE 3–8% Tris-Acetate or 4–12% Bis-Tris Protein Gels (ThermoFisher), followed by transfer to 0.45 µm nitrocellulose membranes (ThermoFisher). Membranes were blocked with 5% non-fat dry milk in TBS-T (0.1% Tween-20) for 1 h, incubated with primary antibodies overnight at 4 °C and then with HRP-conjugated secondary antibodies. Blots were imaged using an Odyssey CLx Imager (LI-COR Biosciences).

### Co-immunoprecipitation assay
The co-immunoprecipitation assay was performed with modifications[4]. In brief, cells were lysed in IP lysis buffer (ThermoFisher) supplemented with 1× Halt Protease Inhibitor Cocktail. For each immunoprecipitation reaction, 500 µg of lysate was incubated overnight with 5 µg of AR antibody (Abcam) and 75 µl of Protein G Dynabeads (ThermoFisher). Beads were then washed three times with IP lysis buffer and subsequently denatured.

## Tandem mass tag-based proteomics sample preparation and mass spectrometry

**Sample preparation.** For each condition, 75 μg of cell lysate was submitted to the Proteomics Resource Facility at the University of Michigan for tandem mass tag-based mass spectrometry. Proteins were first reduced with 5 mM dithiothreitol (30 min, 45 °C) and alkylated with 15 mM 2-chloroacetamide (30 min, 25 °C). After protein precipitation using six volumes of ice-cold acetone and overnight incubation at −20 °C, pellets were centrifuged, air-dried and resuspended in 0.1 M TEAB buffer. Proteins were digested overnight at 37 °C using trypsin/Lys-C (Promega) at a 1:50 enzyme-to-protein ratio with constant mixing.

Peptides were labeled using tandem mass tag 16-plex reagents (Thermo Fisher Scientific, A44521) following the manufacturer's instructions. After 1 h of labeling at room temperature, reactions were quenched with 5% hydroxylamine for 15 min. Labeled samples were pooled, dried and fractionated into 12 fractions using high-pH reversed-phase chromatography (Zorbax 300Extend-C18 column; Agilent 1260 Infinity II high-performance liquid chromatography). Each fraction was dried and reconstituted in 0.1% formic acid/2% acetonitrile for liquid chromatography–mass spectrometry.

**Liquid chromatography–mass spectrometry analysis.** Samples were analyzed on an Orbitrap Ascend Tribrid mass spectrometer with field asymmetric ion mobility spectrometry (FAIMS) and a Vanquish Neo ultra HPLC system. Peptides were separated on a 75 μm × 50 cm PepMap Neo column using a multi-step acetonitrile gradient over 130 min at 300 nl min⁻¹. Mass spectrometry analysis used multinotch-MS3 to enhance quantification accuracy and reduce ratio distortion. FAIMS was operated at compensation voltages of −45 V and −65 V. MS1 scans were acquired at 120K resolution, and selected precursors (charge state 2–6) were fragmented using CID for MS2. Ten MS2 fragments per scan were selected for higher-energy collisional dissociation fragmentation and analyzed in Orbitrap MS3 at 45,000 resolution.

## RNA-seq and data analysis

RNA extraction was performed as described in our prior studies[4]. Ribosomal RNA was depleted using the RiboErase module of the KAPA RNA Hyper+RiboErase HMR Kit (Roche, cat. no. 08098140702), and the ribosomal RNA-depleted RNA was used for library preparation following the manufacturer's protocol. Library quality and concentration were assessed using the Agilent 2100 Bioanalyzer, and libraries were sequenced on an Illumina NovaSeq 6000 using paired-end 150 bp reads at a depth of 15–20 million paired reads per sample.

RNA-seq data were handled using Kallisto (v.0.50.1)[79]. Normalized and filtered read counts (counts >10) were generated using EdgeR (v.3.39.6)[80], and differential expression analysis was performed using Limma-Voom (v.3.58.1)[81]. GSEA was conducted using fgsea (v.1.28.0). Additional visualization was done using tidyverse (v.2.0.0), gtable (v.0.3.5), gplots (v.3.1.3.1), ggplot2 (v.3.5.1) and EnhancedVolcano (v.1.15.0).

## ChIP-seq and data analysis

ChIP-seq was conducted as in our prior studies[4]. In brief, ChIP experiments were conducted using the Ideal ChIP-seq Kit for Transcription Factors (for AR, FOXA1, p300, ERG, BRD4 and MED1 ChIP-seq) or Histones (for H3K27ac, H2BK5ac, H3K20ac and Pol II ChIP-seq) (Diagenode) following the manufacturer's protocol. In brief, $4 \times 10^6$ cells (for transcription factors) and $1 \times 10^6$ cells (for histones) for each ChIP reaction were applied for cross-linking for 10 min in 1% formaldehyde solution, followed by 1/10 volume 1.25 M glycine for 5 min at room temperature to quench the formaldehyde. Chromatin was extracted by cell lysis and sonication (Bioruptor, Diagenode) to break down chromatin into fragments of 200–600 bp. Sheared chromatin was then used for immunoprecipitation with the individual antibody (8 μg for transcription factor or 1 μg for histone), with overnight incubation at 4 °C. DNA fragments were de-crosslinked and purified. Purified DNA was then subjected to sequencing following the manufacturer's instructions (Illumina). ChIP-seq libraries were prepared from purified ChIP samples (1–10 ng) as described previously[4]. Libraries were quantified and quality checked using the Bioanalyzer 2100 (Agilent) and sequenced on the Illumina NovaSeq 6000 Sequencer (2 × 150-nucleotide read length with sequence depth of 25–35 M paired reads).

ChIP-seq data analysis started with trimming using Trimmomatic (v.0.39) (settings TruSeq3-PE-2.fa:2:30:10, minlen 50)[82]. BWA was used to align reads to hg38 (GRCh38) human genome reference ('bwa mem' command with options -5SP -T0, version 0.7.17-r1198-dirty)[83]. Alignments were filtered using SAMtools (quality score cutoff of 20) and Picard MarkDuplicates (removed duplicates)[84,85]. MACS2 was used for peak calling with narrowpeak setting for narrow peaks and a second set of parameters for histone peaks (H3K27Ac,–broad -B–cutoff-analysis–broad-cutoff 0.05–max-gap 500)[86]. In addition, bedtools was used to remove blacklisted regions of the genome from the peaks list (Encode's exclusion list ENCFF356LFX.bed)[87,88]. UCSC's tool wigtoBigwig was used for conversion to bigwig formats[89].

## ATAC-seq and data analysis

ATAC-seq was conducted similarly to our previous studies[4]. In brief, 50,000 cancer cells were lysed using CER-I cytoplasmic lysis buffer from the NE-PER kit (Invitrogen) and incubated for 5 min on ice with occasional gentle mixing. After centrifugation at 1,300g for 5 min at 4 °C, the nuclear pellets were isolated. The nuclei were then treated with 50 μl of 1× TD buffer and 2 μl Tn5 enzyme for 30 min at 37 °C, using the Nextera DNA Library Preparation Kit. After transposition, samples were immediately purified using a Qiagen minElute column and subjected to PCR amplification with NEBNext High-Fidelity 2X PCR Master Mix (New England Biolabs). Optimal PCR cycles were determined via qPCR to avoid over-amplification. The amplified libraries were further purified using a Qiagen minElute column and SPRI beads (Beckman Coulter). Libraries were quantified and quality checked using the Bioanalyzer 2100 (Agilent). Finally, the ATAC-seq libraries were sequenced on the Illumina HiSeq 2500 platform, using a 2 × 50 nucleotide paired-end read length with sequence depth of 30–35 million reads per sample.

Sequencing of ATAC-seq libraries generated fastq files, which were initially processed using Trimmomatic (v.0.39) for trimming[82]. These files were then aligned to the GRCh38/hg38 human genome using bwa mem (v.0.7.17-r1198-dirty)[83], and the alignments were converted to binary format with SAMtools (v.1.9)[84]. We next eliminated reads from mitochondrial DNA and duplicated reads, using SAMtools and Picard MarkDuplicates (v.2.9)[84,85]. Peaks in the ATAC-seq data were identified using MACS2 (v.2.1.1.20160309)[86]. Finally, conversion of data to bigwig format was accomplished using the UCSC tool wigtoBigwig[89].

## Super-enhancer analysis

Regions were selected from the overlap comparison of ChIP-seq peaks as peaks only in the AR sample, or peaks found in both the AR sample and the p300 sample. The AR sample-aligned BAM file was subset to regions from bed files. Super-enhancers were called using HOMER[90]. The subset alignment files were then converted to tag directories using the makeTagDirectory command, and peaks were called with findPeaks (options-style super-superSlope 1000 -typical). A line with slope for $y = 1$ cutoff was used to classify the results as super-enhancers and plotted in R (ggplot).

## IHC and immunofluorescence

IHC and immunofluorescence were carried out on 4 μm formalin-fixed, paraffin-embedded tissue slices on the Ventana ULTRA automated slide stainer platform. Antigen retrieval was done by heating tissues for cell conditioning media 1 (CC1) and 2 (CC2) with primary antibody incubation. Anti-rabbit or anti-mouse secondary antibodies, wherever applicable, were used to develop the immune complex. For IHC, the OmniMap and UltraMap Universal DAB RTU detection kits were

used; for immunofluorescence, Ventana FITC, Red 610 and Cy5 RTU detection kits were used to develop the signal. In the double immunofluorescence study, primary antibodies H2BK5ac, H2BK20ac, p300, CBP and KRT8 were used. The following duplex immunofluorescence combinations were used: H2BK5ac/CK8, H2BK20ac/CK8, CBP/CK8 and p300/CK8. Red 610/FITC fluorophores were used for H2BK5ac/CK8 and H2BK20ac/CK8; Cy5/FITC fluorophores were used for CBP/CK8 and p300/CK8. All double immunofluorescence assays were carried out using CC1 95 °C for antigen retrieval, followed by first antibody (H2BK5ac/H2BK20ac/CBP/p300) incubation, OmniMap anti-rabbit HRP and signal development using either the Discovery 610 Kit or Discovery Cy5 Kit (RTU; details below). The second antibody staining was performed consecutively with heat denaturation before the second primary antibody (KRT8) incubation, OmniMap anti-rabbit HRP Kit and signal development using Discovery FITC (RTU; details below). Similarly, H2BK20ac was used as the first primary antibody to develop Red 610, and KRT8 was used as the second primary antibody to develop FITC. An additional counterstaining step was performed using the DAPI Kit (Roche-Ventana, 760-4196). The following reagents from Roche-Ventana Medical Systems were used: CC1 (06414575001), CC2 (950-223), Inhibitor (760-4840), OmniMap anti-rabbit HRP (760-4311), OmniMap anti-mouse HRP (760-4310), Universal DAB Detection Kit (760-500), FITC Kit (760-232) and Red 610 Kit (760-245).

### PRISM multiplexed cell line screening

The PRISM drug screen using CBPD-409 (eight concentrations, threefold dilution, top dose 10 μM) was performed as previously described[50]. These assays were performed at the Broad Institute. To assess lineage-specific enrichment of CBPD-409 efficacy, we performed a two-sample $t$-test between the CBPD-409 $\log_2(AUC)$ values for the lineage of interest and all the other lineages. A Cohen's $d$ statistic was calculated for each such test as the effect size outlining the difference between that lineage and all others, and a corresponding $P$ value was calculated. To assess the univariate association between the PRISM sensitivity profile ($\log_2(AUC)$) of CBPD-409 and the corresponding gene dependencies across cell lines, we computed Pearson correlations and associated $P$ values between these two quantifications for each gene for which the gene dependencies were available, across all cell lines for which both quantifications were available. For each set of $P$ values from an analysis, the $q$ values were computed using the Benjamini–Hochberg algorithm.

### Human prostate tumor xenograft models

The establishment of human prostate tumor xenograft models was performed as in our prior studies[91]. Male SCID mice (6–8 weeks old) were obtained from the University of Michigan breeding colony. Bilateral subcutaneous tumors were established in the dorsal flanks, and tumor volumes were measured weekly using digital calipers with the formula $(\pi / 6) \times (\text{length} \times \text{width}^2)$. At the study endpoint, mice were killed, and tumors were excised and weighed. All procedures were approved by the University of Michigan IACUC.

For the VCaP non-castrated model, mice were injected subcutaneously with $3 \times 10^6$ VCaP cells suspended in serum-free medium mixed 1:1 with Matrigel (BD Biosciences). Once tumors reached ~200 mm³, mice were randomized and treated with CBPD-409 (3 mg kg⁻¹) or vehicle via oral gavage three times per week for 5 weeks.

For the VCaP castration-resistant model, mice were similarly injected with VCaP cells. After tumors became palpable (150–200 mm³), mice were surgically castrated. Upon tumor regrowth to baseline size, mice were treated with CBPD-409 (3 mg kg⁻¹) or vehicle three times per week, with or without enzalutamide (10 mg kg⁻¹) five times per week, for 5 weeks by oral gavage.

### Statistics and reproducibility

All key experiments were independently repeated at least three times, with most quantitative assays performed in biological triplicates unless otherwise noted. Representative data (for example, micrographs, immunoblots) reflect reproducible results. The number of biological replicates is indicated in the figure legends where applicable.

No statistical method was used to predetermine sample size. No data were excluded from the analyses. The experiments were not randomized. The investigators were not blinded to allocation during experiments and outcome assessment.

Statistical analyses were conducted using two-sided tests unless specified. Comparisons between two groups used unpaired $t$-tests or Wilcoxon rank-sum tests. Pearson correlation was calculated using a two-sided test. Multiple comparisons were adjusted using the Benjamini–Hochberg method. Where applicable, exact $P$ values are reported in the figure legends. For values shown as $P < 0.0001$, this reflects the lower reporting limit of GraphPad Prism.

### Reporting summary

Further information on research design is available in the Nature Portfolio Reporting Summary linked to this article.

### Data availability

All data are available in the paper and its Supplementary Information. All sequencing data generated in this study have been deposited in the Gene Expression Omnibus (GEO) repository GSE255134. ChIP-seq datasets for normal, primary prostate cancer and metastatic CRPC were obtained from GEO accessions GSE130408 (ref. 92) and GSE70079 (ref. 5). Gene dependency scores were analyzed using CRISPR knockout datasets available from the DepMap portal (https://depmap.org). The proteomics data have been deposited to the ProteomeXchange Consortium through the PRIDE[93] partner repository with the dataset identifier PXD065228 for tandem mass tag mass spectrometry and PXD065292 for acetyl-lysine proteomics. Source data are provided with this paper.

### Code availability

No custom code or software was used in this study. Details of all code used in the study are provided in Methods.

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

## Acknowledgements

We thank I. Apel, J. Hu, L. McMurry, A. Miller, C. Caldwell-Smith, K. Gu, J. Cai, X. Wang, P. Gajjala, S. Li, R. Mehra and J. Siddiqui from the Michigan Center for Translational Pathology at the University of Michigan, as well as V. Basrur and the Rogel Cancer Center Proteomics Shared Resource for providing technical assistance. This work was supported by the following: Prostate Cancer Foundation (PCF) 2023 Challenge Award (A.M.C., S.W., A.P., U.V.), National Cancer Institute (NCI) Specialized Programs of Research Excellence Grant P50-CA186786 (A.M.C.), NCI Outstanding Investigator Award R35-CA231996 (A.M.C.), NCI Early Detection Research Network Grant U2C-CA271854 (A.M.C.), National Institutes of Health (NIH) R01CA289013 (S.W.), NIH F99/K00 Fellowship 5-K00-CA245825 (A.P.), University of Michigan Rogel Fellow Award (A.P.) and PCF 2021 and 2023 Young Investigator Awards (A.P. and J.L., respectively). A.M.C. is a Howard Hughes Medical Institute Investigator, A. Alfred Taubman Scholar and American Cancer Society Professor.

## Author contributions

J.L., A.P., Y.Q., S.W. and A.M.C. conceived of and designed the studies. J.L. performed all in vitro and functional genomics experiments with assistance from A.P., S.E., R.S. and M.W. Y.Q., J.C.T. and J.L. performed all animal efficacy and toxicity studies with assistance from T.H., Y. Zheng and Y.C. E.Y., A.P. and Y. Zhang carried out all bioinformatics analyses with assistance from J.G., M.S. and S.E. R.M. and S.M. carried out all histopathological evaluations of drug toxicity and quantified all histology-based data. R.M. and S.M. performed all IHC and immunofluorescence staining. L.X., M.G.R., M.M.R. and J.A.R. performed the PRISM drug screening assay. R.B. carried out all analyses of the PRISM screening. F.S. generated next-generation sequencing libraries, and R.W. and X.C. performed the sequencing. U.V. guided the design of drug efficacy studies. Z.C. and M.W. were involved in the discovery, synthesis and initial profiling of the CBPD-409 compound. J.L., A.P., S.J.M. and A.M.C. wrote the paper and organized the final figures.

## Competing interests

S.W. was a co-founder and served as a paid consultant to Oncopia Therapeutics. S.W. and the University of Michigan owned equity in Oncopia Therapeutics, which was acquired by Roivant Sciences and Proteovant Therapeutics. S.W. is a paid consultant to Proteovant Therapeutics. The University of Michigan has received a research contract from Oncopia Therapeutics, for which S.W. serves as the principal investigator. A.M.C. is a co-founder and serves on the scientific advisory board of Lynx Dx, Oncopia Therapeutics, Flamingo Therapeutics, Medsyn Pharma and Esanik Therapeutics. A.M.C. serves as an advisor to Tempus, Aurigene Oncology, Proteovant and Ascentage. The other authors declare no competing interests. The authors have filed patents on (1) orally active p300/CBP degraders (US patent application no. 63/563,534) and (2) elevation of H2NTac in prostate cancer.

## Additional information

**Extended data** is available for this paper at https://doi.org/10.1038/s41588-025-02336-6.

**Correspondence and requests for materials** should be addressed to Shaomeng Wang, Abhijit Parolia or Arul M. Chinnaiyan.

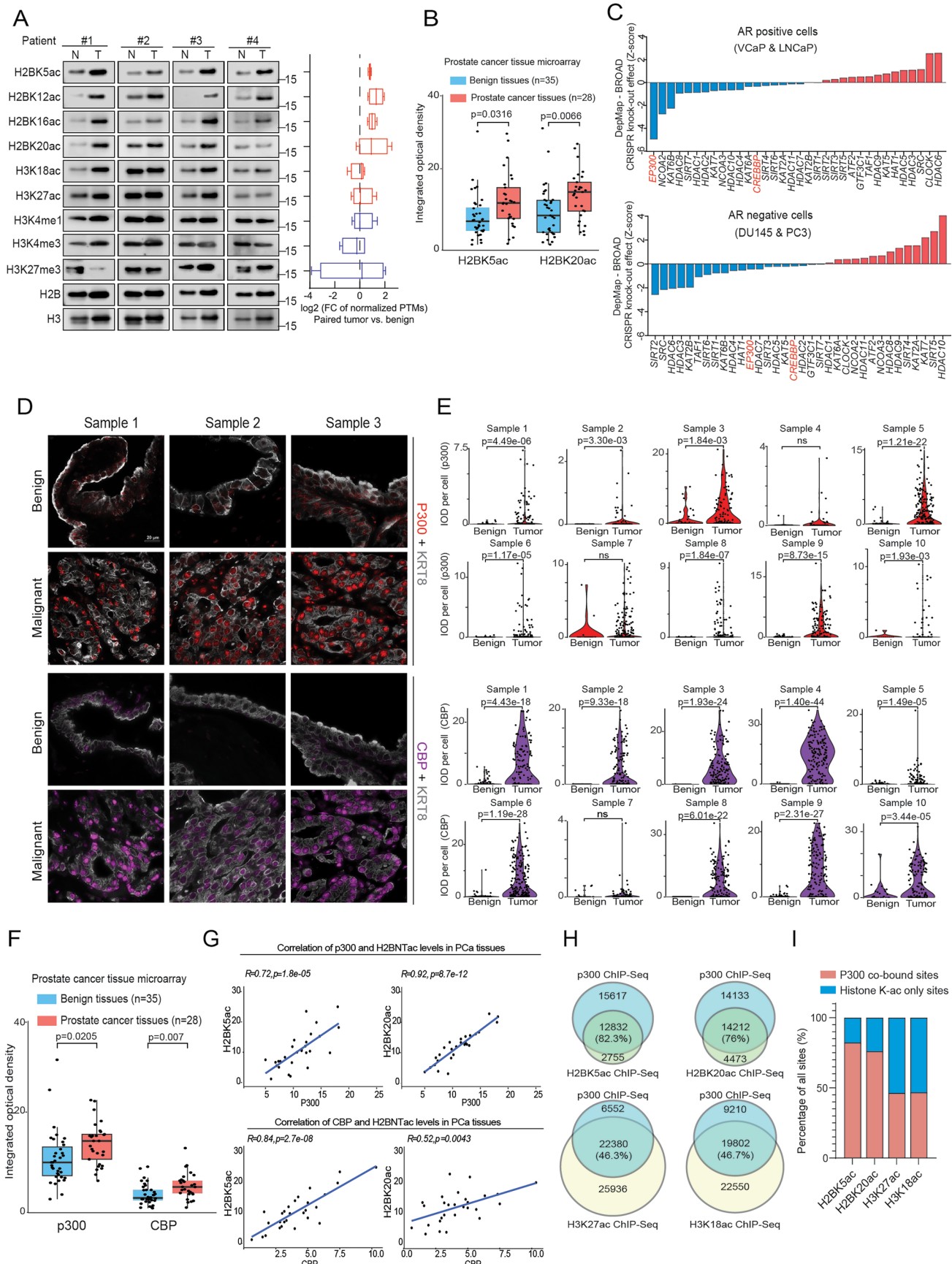

**Extended Data Fig. 1 | See next page for caption.**

**Extended Data Fig. 1 | p300/CREB-binding protein (CBP) and histone H2B N-terminus acetylation (H2BNTac) levels increase in prostate cancer lesions.** **a.** Immunoblot analysis of key histone marks in four pairs of matched prostate cancer (T) and benign adjacent tissues (N). Quantitation and fold change (FC) of the respective histone marks is provided to the right, n=4 pairs of patient samples. Box plot: center line, median; box, 25th–75th percentiles; whiskers, ±1.5× interquartile range (IQR). **b.** Histone H2B lysine 5 acetylation (H2BK5ac) and histone H2B lysine 20 acetylation (H2BK20ac) immunofluorescence (IF) mean intensity (integrated optical density) per sample of benign prostate tissues (n=35) and prostate cancer tissues (n=28) from tissue microarray (two-sided t-test). Box plot: center line, median; box, 25th–75th percentiles; whiskers, ±1.5× IQR, each dot represents an individual tissue core. **c.** Cumulative DepMap clustered regularly interspaced short palindromic repeats (CRISPR) knockout dependency scores of histone acetyltransferases and deacetylases in androgen receptor (AR)-positive and AR-negative prostate cancer cell lines. Aggregated z-scores for each gene across individual cell lines derived from LNCaP and VCaP; or DU145 and PC3 DepMap screens are shown, with p300 and CBP highlighted in red. **d.** Representative multiplex IF images of p300 (red)/keratin 8 (KRT8, gray), and CBP (purple)/KRT8 (gray) staining in patient-matched adjacent benign and tumor tissues. Magnification: 200x. Scalebar = 50 μm. **e.** p300 and CBP IF mean intensity per cell (integrated optical density) of each case (n=10) from images in left panel (two-sided t-test). **f.** p300 and CBP IF mean intensity (integrated optical density) per sample of benign prostate tissues (n=35) and prostate cancer tissues (n=28) from tissue microarray (two-sided t-test). Box plot: center line, median; box, 25th–75th percentiles; whiskers, ±1.5× IQR, each dot represents an individual tissue core. **g.** Pearson's correlation coefficient analyses of p300/H2BNTac and CBP/H2BNTac levels in prostate tumor tissues from the cohort in panels **b** & **f**. Two-sided Pearson correlation tests were used to assess statistical significance. **h.** Venn diagrams illustrating overlaps of genome-wide p300, H2BK5ac, H2BK20ac, histone H3 lysine 18 acetylation (H3K18ac), and histone H3 lysine 27 acetylation (H3K27ac) chromatin immunoprecipitation sequencing (ChIP-seq) peaks in VCaP cells. **i.** Bar charts illustrating overlaps of genome-wide p300, H2BK5ac, H2BK20ac, H3K18ac, and H3K27ac ChIP-seq peaks in VCaP cells.

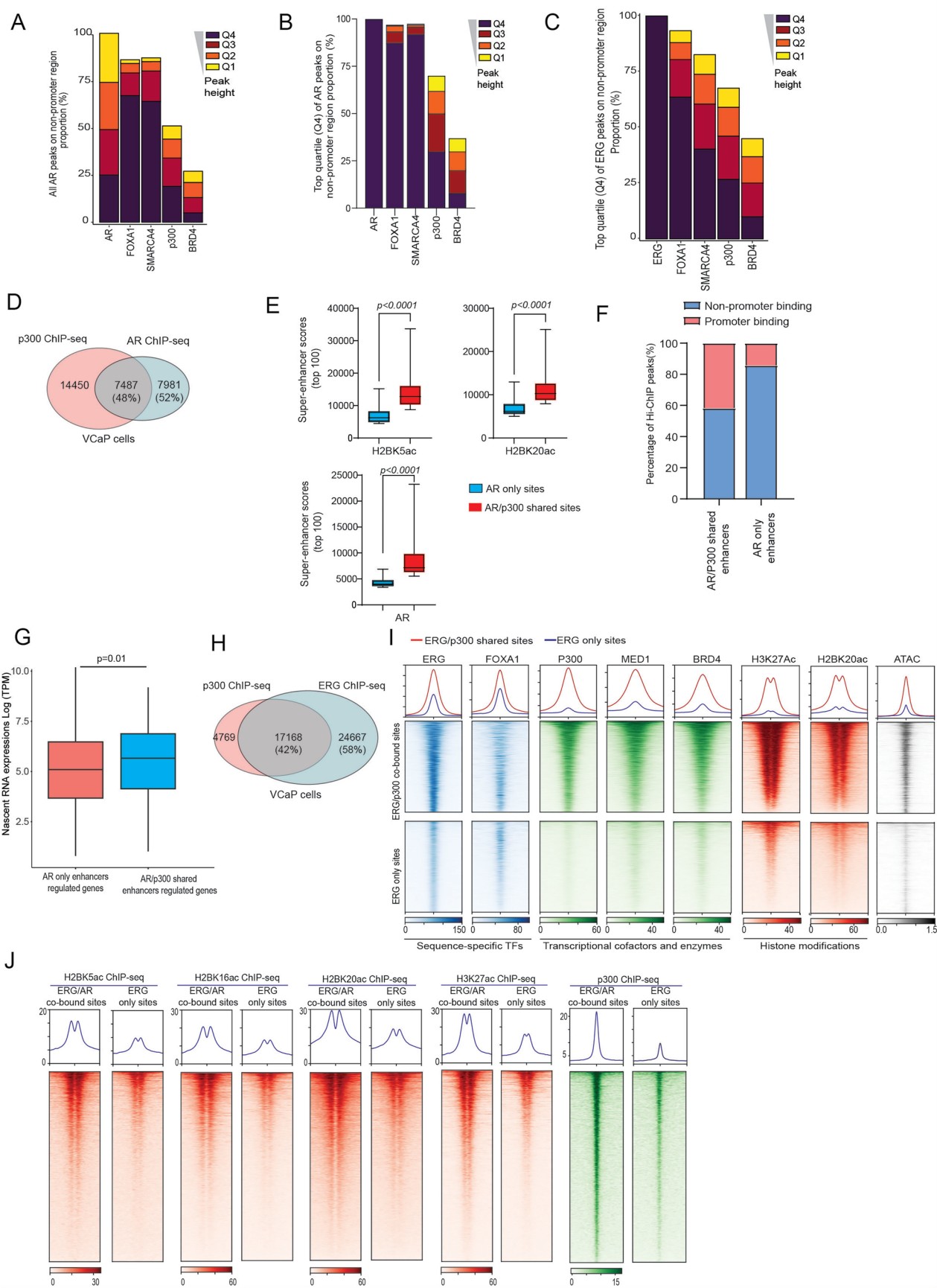

**Extended Data Fig. 2 | See next page for caption.**

**Extended Data Fig. 2 | p300 is the critical histone acetyltransferase for assessment of the active AR and ETS transcription factor ERG (ERG) cistromes in prostate cancer. a**. Bar charts depicting the proportion of forkhead box A1 (FOXA1), SMARCA4, p300, and bromodomain-containing 4 (BRD4) ChIP-seq peaks that map to all AR peaks located in non-promoter regions. **b**. Bar charts depicting the proportion of FOXA1, SMARCA4, p300, and BRD4 ChIP-seq peaks that map to the top quartile of AR ChIP-seq peaks located in non-promoter regions in VCaP cells. **c**. Bar charts depicting the proportion of FOXA1, SMARCA4, p300, and BRD4 peaks that map to the top quartile of ERG ChIP-seq peaks located in non-promoter regions in VCaP cells. **d**. Venn diagram illustrating overlaps of genome-wide p300 and AR ChIP-seq peaks in VCaP cells. **e**. Bar charts depicting the H2BK5ac, H2BK20ac, and AR super-enhancer scores at AR only or AR/p300 co-bound sites (two-sided t test, n=100 binding sites). Box plot: center line, median; box, 25th–75th percentiles; whiskers, ±1.5× IQR. **f**. The histone H3 lysine 4 trimethylation (H3K4me3) HiChIP-seq data (GSM5229035) was analyzed to

identify enhancer looping events associated with both AR/p300 and AR-only regions. Bar chart showing the percentage of enhancers looping to promoter versus non-promoter regions. **g**. Bar chart illustrating the nascent RNA expression levels of genes regulated by AR-only enhancers versus AR/p300-shared enhancers. Statistical significance was assessed using a two-sided t-test. n=180 for AR-only enhancers and n=141 for AR/p300 shared enhancers. Box plot: center line, median; box, 25th–75th percentiles; whiskers, ±1.5× IQR. **h**. Venn diagram illustrating overlaps of genome-wide p300 and ERG ChIP-seq peaks in VCaP cells. **i**. ChIP-seq and assay for transposase-accessible chromatin sequencing (ATAC-seq) read-density heatmaps at ERG/p300 co-bound and ERG only binding sites in VCaP cells. Transcription factors (TFs), transcription cofactors, and respective histone marks are indicated. **j**. ChIP-seq read-density heatmaps of indicated histone marks and p300 at ERG/AR co-bound and ERG only binding sites in VCaP cells.

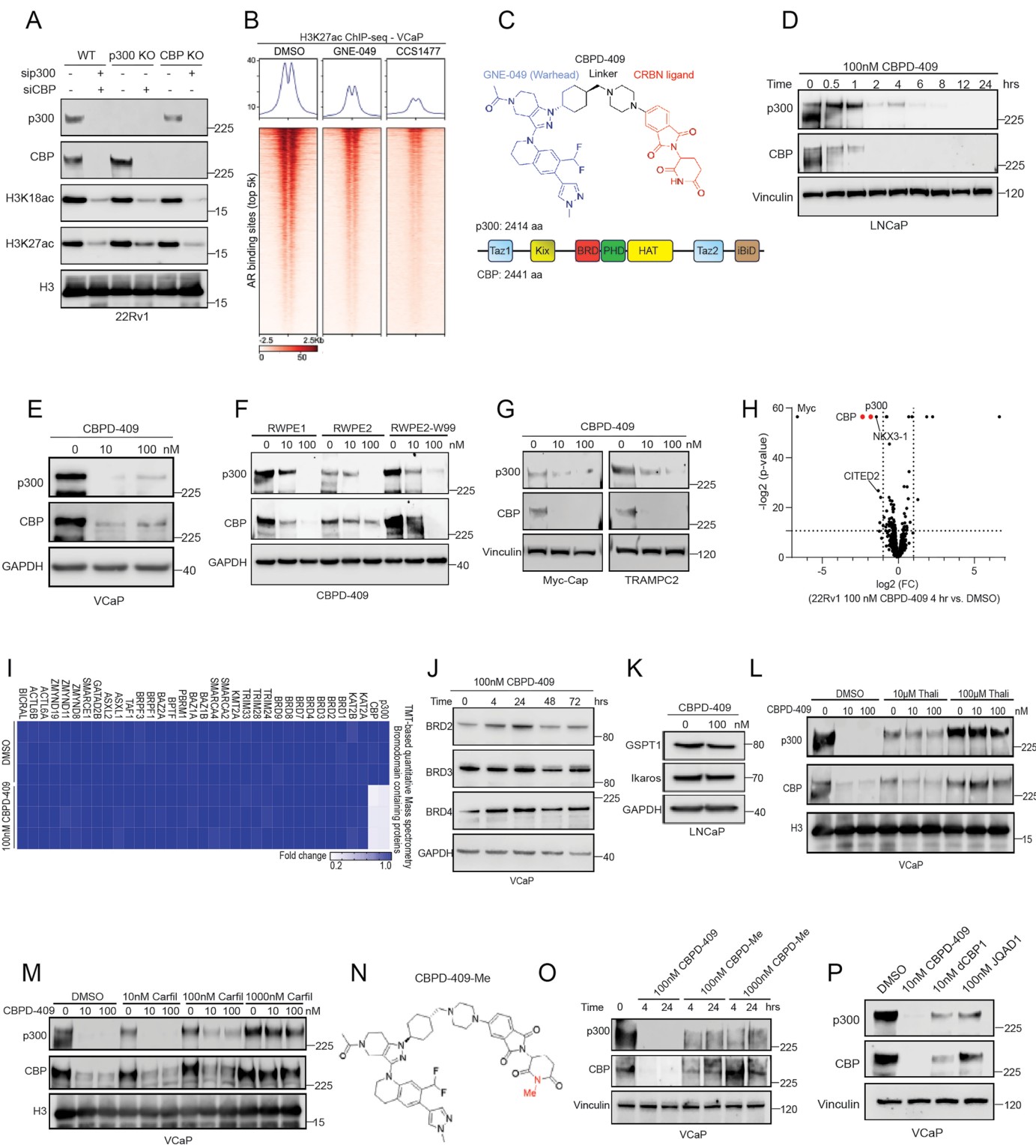

**Extended Data Fig. 3 | See next page for caption.**

**Extended Data Fig. 3 | Characterization of the on-target degradation effects of the p300/CBP degrader, CBPD-409. a.** Immunoblot analysis of p300, CBP, and indicated histone marks in 22Rv1 wild type (WT), p300 knockout (KO), CBP KO, and p300 KO with siCBP cells. **b.** ChIP-seq read-density heatmaps of H3K27ac at top 5000 AR binding sites on non-promoter regions in VCaP cells with 4 h of 1 μM GNE-049 or 1 μM CCS1477 treatments. **c.** Structure of CBPD-409 and schematic of p300 and CBP domains. **d.** Immunoblot analysis of p300 and CBP in LNCaP cells treated with 100 nM CBPD-409 for the indicated durations. **e.** Immunoblot analysis of p300 and CBP in VCaP cells treated with 10 and 100 nM CBPD-409 for 4 h. **f.** Immunoblot analysis of p300 and CBP in non-neoplastic prostatic cells treated with 10 nM or 100 nM CBPD-409 for 4 h. **g.** Immunoblot analysis of p300 and CBP in Myc-Cap and TRAMPC2 treated with 10 nM or 100 nM CBPD-409 for 4 h. **h.** Tandem mass tag (TMT) mass spectrometry (MS) assay to evaluate effects of CBPD-409 (100 nM, 4 h) on the proteome of 22Rv1 cells. N = 3 independent experiments. All t-tests performed were two-tailed t-tests assuming equal variances. **i.** Heatmap of relative abundance of all bromodomain-containing proteins detected in TMT-based quantitative MS after 4 h CBPD-409 treatment of VCaP cells. **j.** Immunoblot analysis of bromodomain and extraterminal (BET) proteins (bromodomain-containing 2 (BRD2), bromodomain-containing 3 (BRD3), and BRD4) in VCaP cells treated with 100 nM CBPD-409 for the indicated durations. **k.** Immunoblot analysis of cereblon (CRBN) neo-substrates (GSPT1 and Ikaros) in LNCaP cells treated with 100 nM CBPD-409 for 24 h. **l.** Immunoblot analysis of p300 and CBP in VCaP cells pre-treated with thalidomide (Thali), then treated with CBPD-409 at the indicated concentrations for 4 h. **m.** Immunoblot analysis of p300 and CBP in VCaP cells pre-treated with carfilzomib (Carfil), then treated with CBPD-409 at the indicated concentrations for 4 h. **n.** Schematic of CBPD-409-Me structure. **o.** Immunoblot analysis of p300 and CBP in VCaP cells treated with CBPD-409 and CBPD-409-Me at indicated concentrations for the noted durations. **p.** Immunoblot analysis of p300 and CBP in VCaP cells treated with 10 nM CBPD-409, 10nM dCBP1 and 100nM JQAD1 for 4 h.

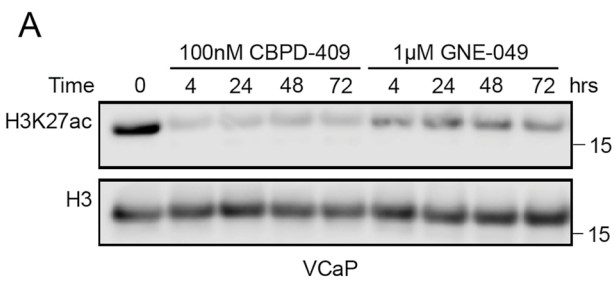

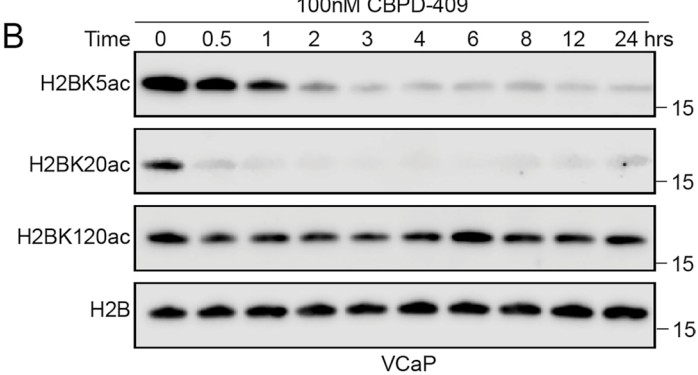

C

Summary of differential acetylated sites (acetylated proteins)

| Group Name | Up-regulated (FC>1.5, P<0.05) | Down-regulated (FC<1/1.5, P<0.05) |
|---|---|---|
| DMSO vs CBPD 1hr | 16 (11) | 134 (80) |
| DMSO vs CBPD 4hrs | 4 (2) | 243 (151) |
| DMSO vs CBPD 12hrs | 5 (5) | 302 (197) |
| DMSO vs CBPD 24hrs | 11 (8) | 337 (214) |

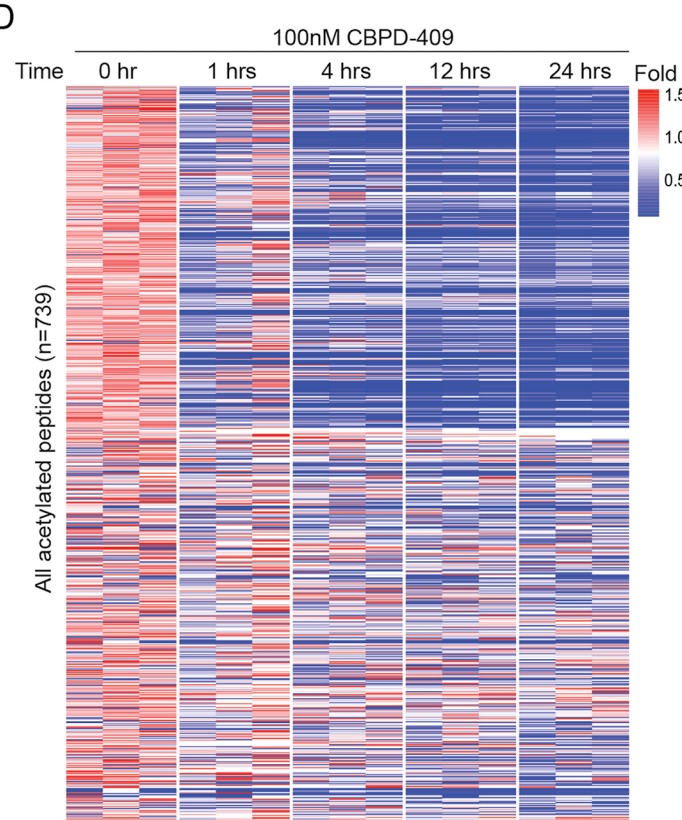

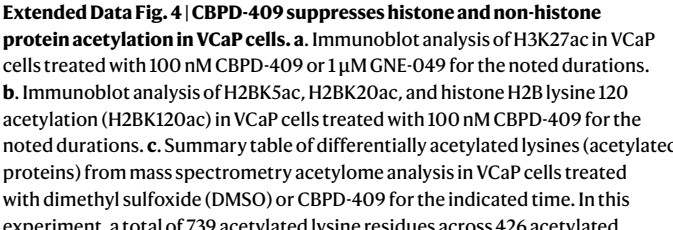

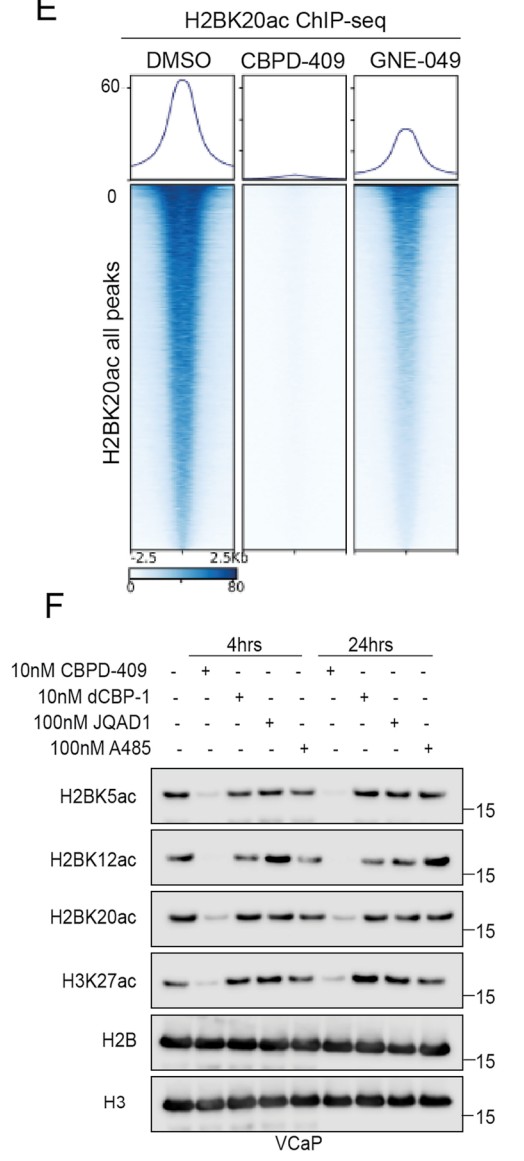

**Extended Data Fig. 4 | CBPD-409 suppresses histone and non-histone protein acetylation in VCaP cells. a.** Immunoblot analysis of H3K27ac in VCaP cells treated with 100 nM CBPD-409 or 1 µM GNE-049 for the noted durations. **b**. Immunoblot analysis of H2BK5ac, H2BK20ac, and histone H2B lysine 120 acetylation (H2BK120ac) in VCaP cells treated with 100 nM CBPD-409 for the noted durations. **c**. Summary table of differentially acetylated lysines (acetylated proteins) from mass spectrometry acetylome analysis in VCaP cells treated with dimethyl sulfoxide (DMSO) or CBPD-409 for the indicated time. In this experiment, a total of 739 acetylated lysine residues across 426 acetylated proteins were identified with high confidence (localization probability > 0.75). Differential acetylation was determined using a two-sided t-test with adjustment for multiple comparisons. **d**. Fold change heatmap of all detected acetylated peptides in VCaP cells treated with or without 100 nM CBPD-409 for the indicated durations. The results were obtained from three replicated experiments for each condition. **e.** ChIP-seq read-density heatmaps of genome-wide H2BK20ac in VCaP cells with 4 h of 100 nM CBPD-409 or 1 µM GNE-049 treatments. **f.** Immunoblot analysis of indicated histone marks in VCaP cells treated with 10 nM CBPD-409, 10 nM dCBP-1, 100 nM JQAD1, or 100 nM A485 for 4 or 24 h.

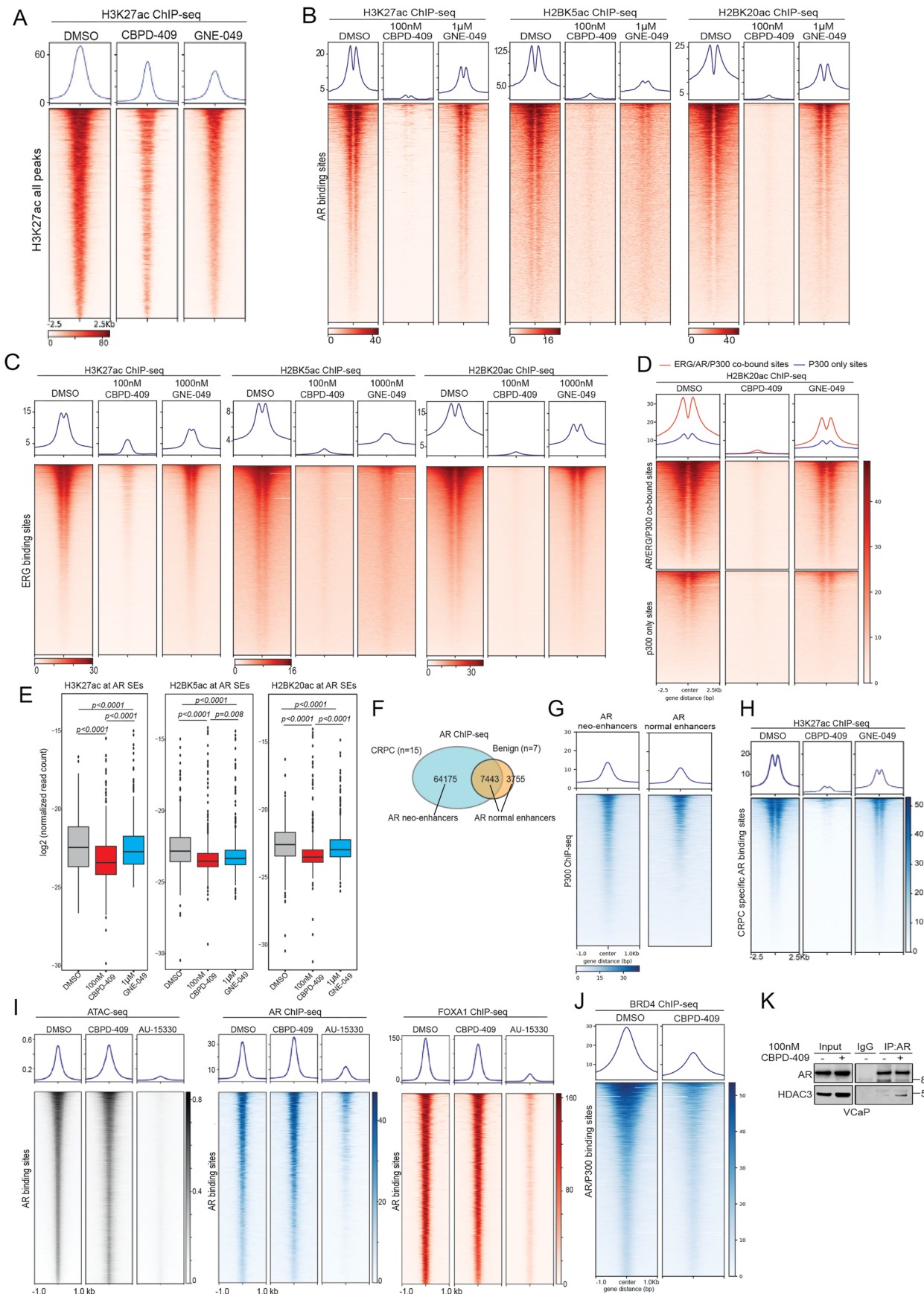

**Extended Data Fig. 5 | See next page for caption.**

**Extended Data Fig. 5 | CBPD-409 inhibits histone acetylation at the AR and ERG enhanceosomes without affecting chromatin accessibility and AR/FOXA1 chromatin distribution. a**. ChIP-seq read-density heatmaps of genome-wide H3K27ac in VCaP cells with 4 h of 100 nM CBPD-409 or 1 μM GNE-049 treatments. **b**. ChIP-seq read-density heatmaps of H3K27ac, H2BK5ac, and H2BK20ac at AR cis-regulatory elements in VCaP cells with 4 h of 100 nM CBPD-409 or 1 μM GNE-049 treatment. **c**. ChIP-seq read-density heatmaps of H3K27ac, H2BK5ac, and H2BK20ac at ERG cis-regulatory elements in VCaP cells with 4 h of 100 nM CBPD-409 or 1 μM GNE-049 treatment. **d**. ChIP-seq read-density heatmaps of H2BK20ac at ERG/AR/p300 co-bound sites and p300 only binding sites in VCaP cells with 4 h of 100 nM CBPD-409 or 1 μM GNE-049 treatment. **e**. H3K27ac, H2BK5ac, and H2BK20ac ChIP-seq intensities at super-enhancers associated with AR non-promoter binding sites in VCaP cells with 4 h of 100 nM CBPD-409 or 1 μM GNE-049 treatment (Two-sided Wilcox test, n=100 binding sites). Box

plot: center line, median; box, 25th–75th percentiles; whiskers, ±1.5× IQR, dots, outliers. **f**. Venn diagram showing number of AR ChIP-seq peaks in castration-resistant prostate cancer (CRPC) tissues and benign prostate tissues. **g**. ChIP-seq read-density heatmaps of p300 at AR CRPC-specific enhancers and AR normal enhancers in VCaP cells. **h**. ChIP-seq read-density heatmaps of H3K27ac within CRPC-specific AR binding sites in VCaP cells after 4 h of 100 nM CBPD-409 or 1 μM GNE-049 treatments. **i**. ATAC-seq read-density heatmaps and ChIP-seq read-density heatmaps of AR and FOXA1 peaks at AR cis-regulatory elements in VCaP cells with 100 nM CBPD-409 and 1 μM switch/sucrose non-fermentable (SWI/SNF) ATPase degrader AU-15330 treatments for 4 h. **j**. ChIP-seq read-density heatmaps of BRD4 at AR/p300 co-bound cis-regulatory elements in VCaP cells with 4 h of 100 nM CBPD-409 treatment. **k**. Immunoblot analysis of the co-immunoprecipitation (Co-IP) assay showing that 100 nM CBPD-409 increases the interaction between AR and histone deacetylase 3 (HDAC3) proteins.

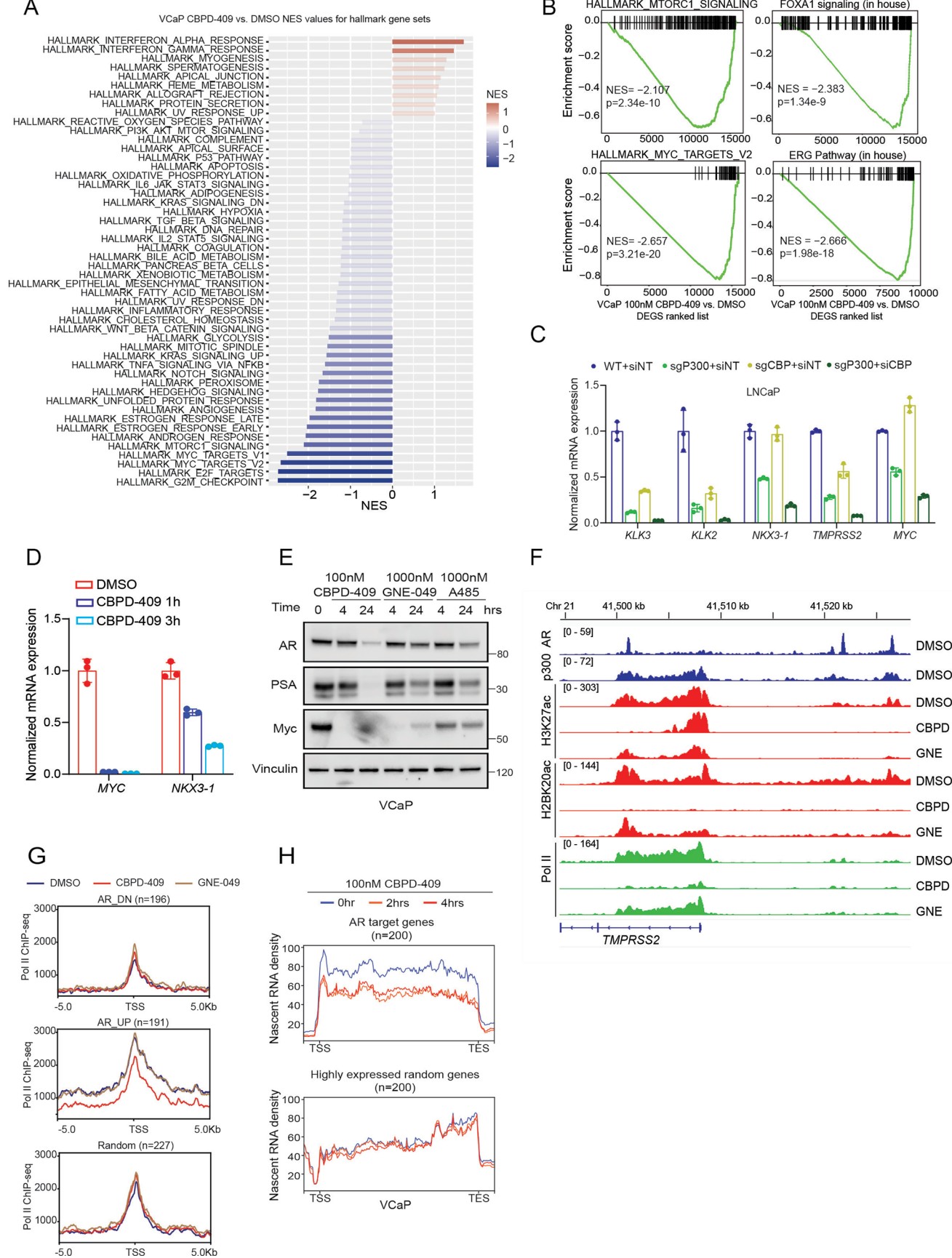

**Extended Data Fig. 6 | See next page for caption.**

**Extended Data Fig. 6 | CBPD-409 significantly suppresses AR transactivation.**
**a**. Gene set enrichment analysis (GSEA) net enrichment score (NES) plot of significantly altered hallmark pathways in VCaP cells treated with 100 nM CBPD-409 for 24 h. **b**. GSEA plots for mTORC1, FOXA1, MYC, and ERG signaling pathways using the fold change rank-ordered gene signature from VCaP cells treated with CBPD-409 for 24 h. Statistical significance was assessed using a two-sided GSEA permutation test with adjustment for multiple comparisons.
**c**. Quantitative PCR (qPCR) of *KLK3*, *KLK2*, *NKX3-1*, *TMPRSS2*, and *MYC* expression in LNCaP WT, p300 KO, CBP KO, and p300 KO with siCBP cells, n=3 biological replicates, data are presented as mean ± standard deviation (SD). **d**. qPCR of *MYC* and *NKX3-1* expression in VCaP cells treated with CBPD-409 for 1 and 3 h, n=3 biological replicates, data are presented as mean ± SD. **e**. Immunoblot analysis of indicated proteins in VCaP cells treated with 100 nM CBPD-409, 1000 nM

GNE-049, or 1000 nM A485 for noted durations. **f**. ChIP-seq tracks of AR, p300, H3K27ac, H2BK20ac, and RNA Polymerase II (Pol II) within the *TMPRSS2* gene locus in VCaP cells treated with 100 nM CBPD-409 or 1 μM GNE-049 for 4 h.
**g**. Average Pol II ChIP-seq coverage profiles of AR down-regulated genes (AD_DN), AR activated genes (AR_UP), and random genes in VCaP cells treated with 100 nM CBPD-409 or 1 μM GNE-049 for 4 h. **h**. Densities of 5-ethynyluridine (EU) labeled nascent RNA sequencing (RNA-seq) reads of classical AR target genes (n=200) and highly expressed random genes (n=200) in VCaP cells treated with or without 100 nM CBPD-409 for 2 and 4 h. Classical AR target genes were defined as those up-regulated by 1 nM R1881. Highly expressed genes were defined as those with log2(counts per million (CPM)) > 6 from RNA-seq data, from which 200 random genes were selected. The data were derived from two independent samples for each group.

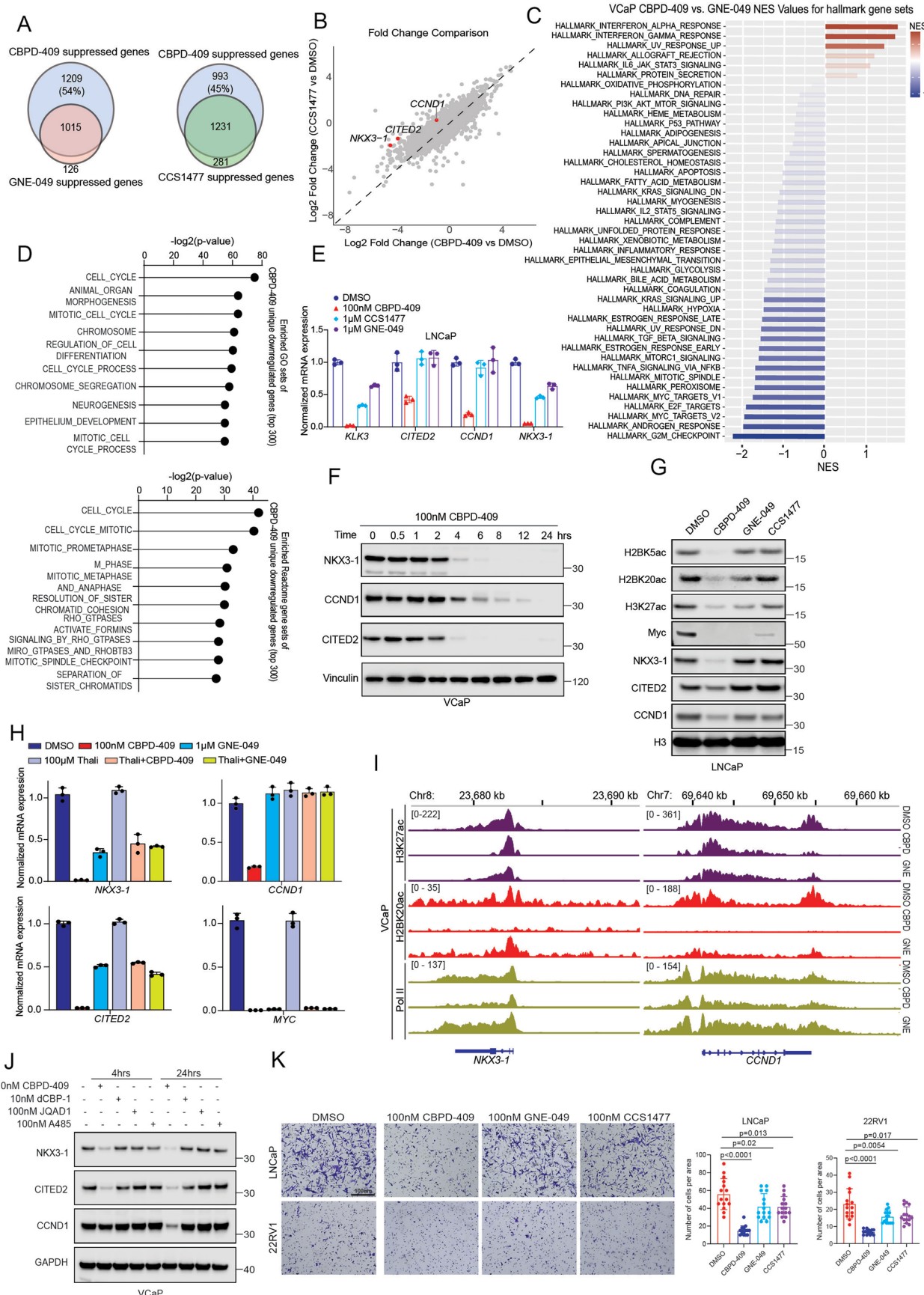

Extended Data Fig. 7 | See next page for caption.

**Extended Data Fig. 7 | CBPD-409 targets a unique set of cell cycle-related genes to suppress proliferation. a**. Venn diagrams showing overlaps between genes suppressed by CBPD-409 and those suppressed by GNE-049 or CCS1477 in VCaP cells. Data obtained from RNA-seq with two independent samples for each condition. **b**. Comparison of whole-transcriptome in VCaP cells treated with 100nM CBPD-409 and 1 μM CCS-1477 for 24 h. N=2 biological replicates. *NKX3-1*, cyclin D1 *(CCND1)* and CBP/p300-interacting transactivator with Glu/Asp rich carboxy-terminal domain 2 (*CITED2*) are highlighted in red. **c**. GSEA net enrichment score (NES) plot comparing hallmark pathway alterations in VCaP cells: 100 nM CBPD-409 vs. 1 μM GNE-049 treatment for 24 h. **d**. Compute overlaps of ranked top 300 CBPD-409 unique down-regulated genes with gene ontology (GO) and Reactome gene sets. P-values were calculated using a two-sided statistical test with adjustment for multiple comparisons. **e**. qPCR of indicated genes expression in LNCaP cells treated with CBPD-409, GNE-049, or CCS1477 for 24 h, n=3 biological replicates, data are presented as mean ± SD. **f**. Immunoblot analysis of NKX3-1, CCND1, and CITED2 in VCaP cells treated with

100 nM CBPD-409 for the noted durations. **g**. Immunoblot analysis of indicated proteins and histone marks in LNCaP cells treated with 100 nM CBPD-409, 1 μM GNE-049, or 1 μM CCS1477 for 24 h. **h**. qPCR of indicated genes expression in VCaP cells pre-treated with 100 μM thalidomide for 1 h, then treated with CBPD-409 or GNE-409 for 4 h, n=3 biological replicates, data are presented as mean ± SD. **i**. ChIP-seq tracks of H3K27ac, H2BK20ac, and Pol II within the *NKX3-1* and *CCND1* gene loci in VCaP cells treated with 100 nM CBPD-409 or 1 μM GNE-049 for 4 h. **j**. Immunoblot analysis of indicated proteins in VCaP cells treated with 10 nM CBPD-409, 10 nM dCBP-1, 100 nM JQAD1, or 100 nM A485 for noted durations. **k**. Left: representative images showing invasive cells from the Boyden chamber invasion assay. Each group included three biological replicates (independent transwell inserts). Right: Bar plot quantifying invasive cells. Five random fields were imaged per transwell, yielding n = 15 fields per group. Data are presented as mean ± SD. Statistical significance was assessed using a two-sided t-test. Scale bar = 100 μm.

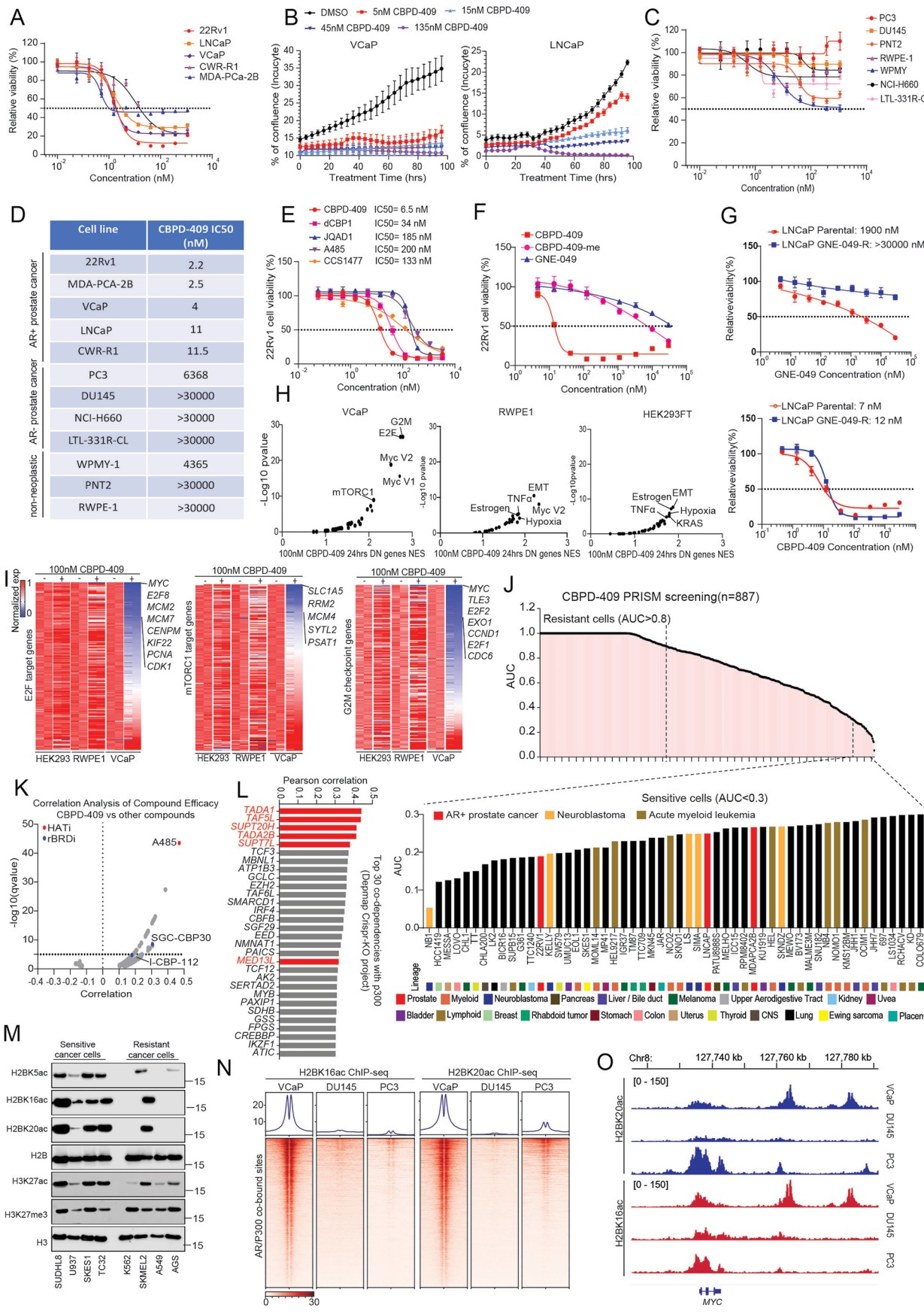

**Extended Data Fig. 8 | See next page for caption.**

**Extended Data Fig. 8 | CBPD-409 exhibits superior cytotoxicity in AR-positive prostate cancer cells compared to other p300/CBP inhibitors. a**. Dose-response curves of AR-positive prostate cancer cells treated with CBPD-409. Data are presented as mean ± SD (n = 6 biologically replicated wells). **b**. Incucyte growth curves of LNCaP and VCaP cells. Data are presented as mean ± SD (n = 6 biologically replicated wells). **c**. Dose-response curves of AR-negative prostate cancer cells and non-neoplastic prostate cells treated with CBPD-409. Data are presented as mean ± SD (n = 6 biologically replicated wells). **d**. Table summary of CBPD-409 IC50. **e**. Dose-response curves and IC50 of 22RV1 cells treated with CBPD-409, dCBP-1, and JQAD1, CCS1477, and A485. Data are presented as mean ± SD (n = 3 biologically replicated wells). **f**. Dose-response curves and IC50 of LNCaP cells treated with CBPD-409, CBPD-409-Me, and GNE-049. Data are presented as mean ± SD (n = 3 biologically replicated wells). **g**. Dose-response curves and IC50 of LNCaP parental and GNE-049 resistant (R) cells treated with GNE-049 (left) or CBPD-409 (right). Data are presented as mean ± SD (n = 3 biologically replicated wells). **h**. Scatterplot showing GSEA results for genes suppressed by 100 nM CBPD-409 in VCaP, RWPE1, and HEK293FT cells. Statistical significance was assessed using a two-sided permutation test with adjustment for multiple comparisons. **i**. RNA-seq heatmaps for E2F, mTORC1, and G2M checkpoint target genes in HEK293FT, RWPE1, and VCaP cells treated with 100 nM CBPD-409 for 24 h, n=2. **j**. Rank-order plot of area under the curve (AUC) values for CBPD-409 from PRISM assays. Left: the AUC values for CBPD-409 in 887 tested cancer cells. Right: the AUC values for CBPD-409 in 52 sensitive cancer cell lines. n=3. **k**. Scatterplot depicting the correlation between CBPD-409 sensitivity (log2-AUC; from PRISM screening) and other drugs sensitivities (from DepMap drug repurposing dataset) across 887 cancer cell lines. **l**. Rank-order plot depicting the top 30 co-dependencies with p300 from DepMap CRISPR-KO program. **m**. Immunoblot analysis of indicated histone marks in CBPD-409 sensitive and resistant cells. **n**. ChIP-seq read-density heatmaps of H2BK20ac and histone H2B lysine 16 acetylation (H2BK16ac) at non-promoter regions in different cells. **o**. ChIP-seq tracks of H2BK20ac and H2BK16ac within the *MYC* gene locus in VCaP, DU145, and PC3 cells.

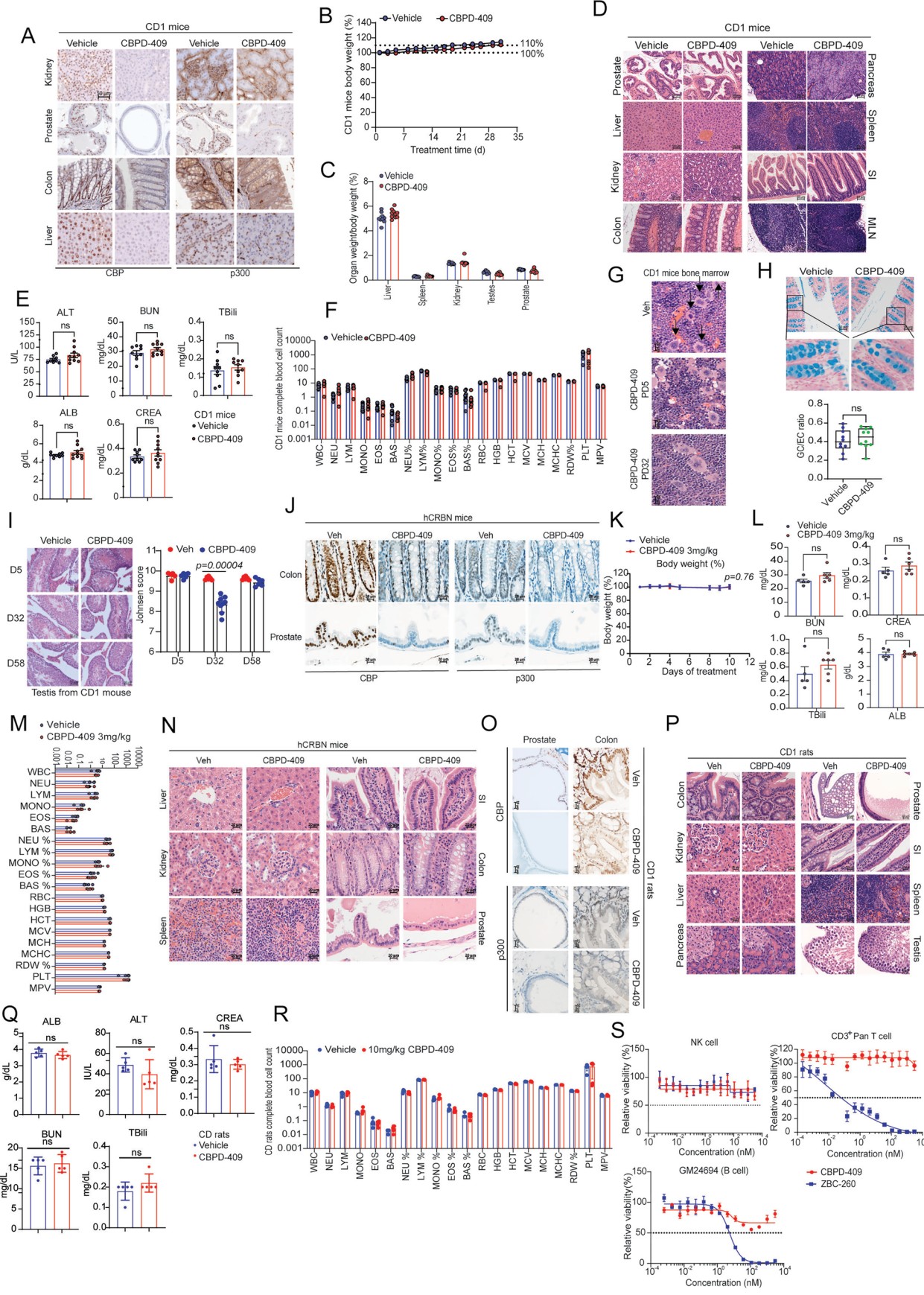

**Extended Data Fig. 9 | See next page for caption.**

**Extended Data Fig. 9 | CBPD-409 displays no evident toxicity in murine models. a**. Representative immunohistochemistry images of CBP/p300 in CD-1 mice. Scalebar = 50 μm. **b**. CD1 mice body weight (%) measurements (two-sided t-test, mean ± SEM). Vehicle: n = 10, CBPD-409: n=10. **c**. Major organ weights from CD1 mice (mean ± standard error of the mean (SEM)). N = 10, biological replicates. **d**. Representative organs H&E staining of CD1 mice. Scalebar = 50 μm. **e**. Liver/kidney function of CD1 mice (two-sided t-test, mean ± SEM). Vehicle: n=9; CBPD-409: n=9. **f**. Complete blood count of CD1 mice (mean ± SEM). N = 8, biological replicates. **g.** Representative hematoxylin and eosin (H&E) staining of femur bone marrow samples of CD1 mice. Megakaryocytes are indicated by arrows. Scalebar = 20 μm. **h**. Representative alcian-blue-staining images from CD1 mice large-intestinal (n = 10/treatment group). Right, quantification of goblet:epithelial cell (two-sided t-test, mean ± SEM). Box plot: center line, median; box, 25th–75th percentiles; whiskers, ±1.5× IQR, each dot represents an individual tissue core. **i**. Representative H&E images of mouse testes from CD1 study. Right: Johnsen score n=5 for Day 5 (D5) vehicle and CBPD-409; n=6

for Day 32 (D32) vehicle and n=9 for D32 CBPD-409; n=6 for D58 vehicle and CBPD-409 (two-sided t-test, mean ± SEM). Scalebar = 50 μm. **j**. Representative immunohistochemistry images of CBP/p300 in hCRBN mice, n=6 per treatment. Scalebar = 20 μm. **k**. hCRBN mice body weight measurements (two-sided t-test, mean ± SEM). Vehicle: n = 5, CBPD-409: n=6. **l**. Liver/kidney function of hCRBN mice (two-sided t-test, mean ± SEM). Vehicle: n=5; CBPD-409: n=6. **m**. Complete blood count of hCRBN mice (mean ± SEM). Vehicle: n=5; CBPD-409: n=6. **n**. Representative H&E staining of major organs of hCRBN mice. Scalebar = 20 μm. **o**. Representative immunohistochemistry images of CBP/p300 in CD rats treated with CBPD-409 (10 mg/kg, P.O). n=5 per treatment. Scalebar = 20 μm. **p**. Representative H&E staining of major organs from CD rat study. Scalebar = 20 μm. **q**. Liver/kidney function of CD rats (two-sided t-test, mean ± SEM). Vehicle: n=5; CBPD-409: n=5. **r**. Complete blood count from CD rats (mean ± SEM). N = 5, biological replicates. **s**. Dose-response curves of human primary CD3+ T cells, NK cells, and GM24694 treated with CBPD-409 and ZBC-260 (mean ± SD). N = 3 biologically replicated wells.

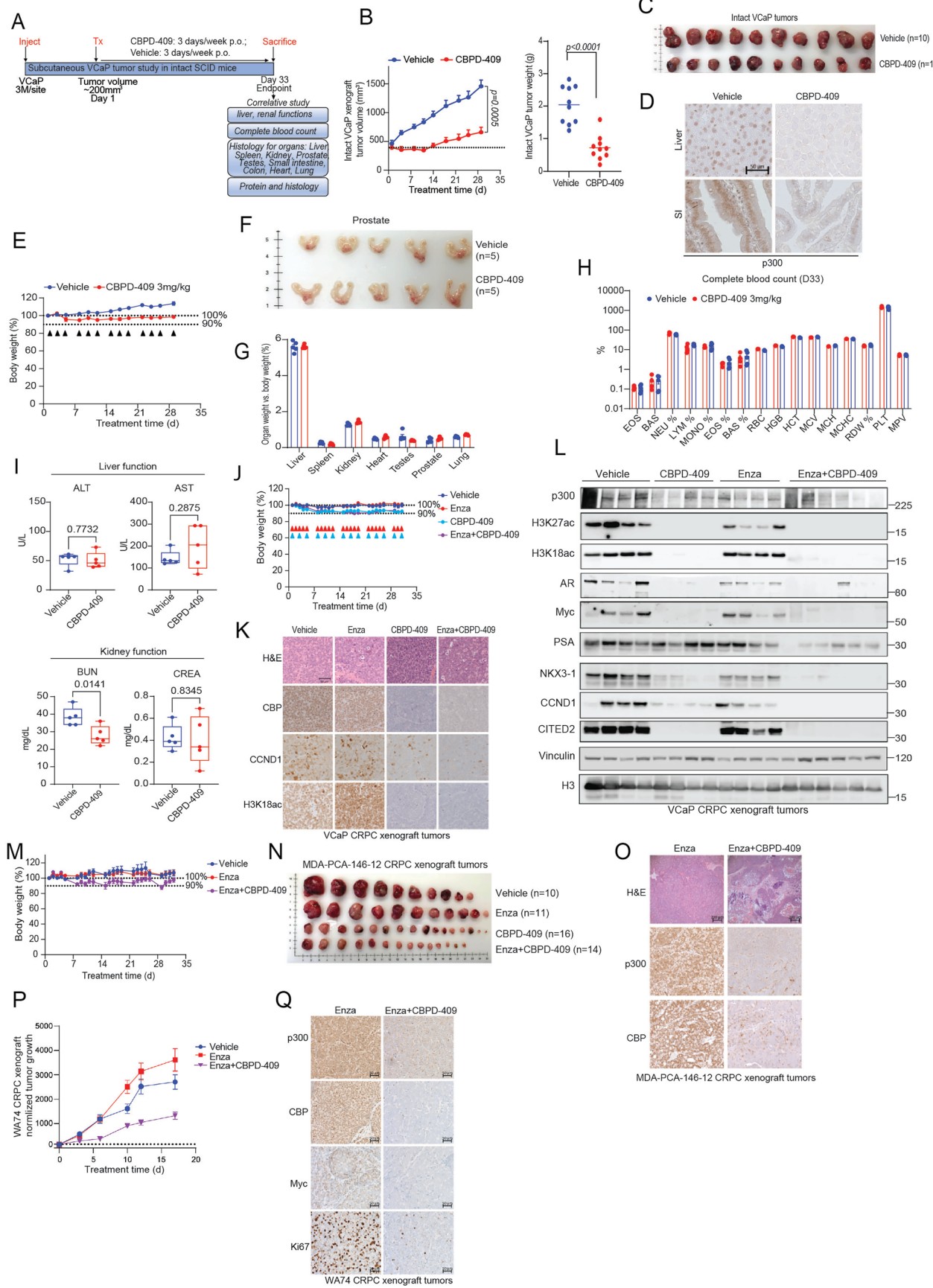

**Extended Data Fig. 10 | See next page for caption.**

**Extended Data Fig. 10 | CBPD-409 efficacy in intact VCaP and castrated CRPC tumor models. a**. Schematic of the CBPD-409 *in vivo* efficacy study in intact VCaP xenograft model. **b**. Tumor volume curve and tumor weights in the intact VCaP xenograft model. Statistical analysis was performed using a two-sided t-test. Data are presented as mean ± SEM. Sample sizes are as follows: vehicle, n = 10; CBPD-409, n = 10. **c.** Individual tumor images from vehicle and CBPD-409 groups in intact VCaP xenograft study. **d**. Representative immunohistochemistry images of p300. n=10 per treatment. Scalebar = 50 μm. **e**. Percent body weight measurement of intact VCaP tumor bearing mice (two-sided t-test, mean ± SEM). Vehicle: n = 10; CBPD-409: n = 10. **f.** Individual prostate images from intact VCaP xenograft study. **g.** Percent major organ weight measurements from intact VCaP xenograft study. Data are presented as mean ± SEM (n = 6 mice/group). **h**. Complete blood count from intact VCaP xenograft study. Data are presented as mean ± SEM (n = 6, biological replicates). **i**. Liver/kidney function tests from intact VCaP xenograft study. Vehicle: n=5; CBPD-409: n=5; two-sided t-test,

data are presented as mean ± SD. Box plot: center line, median; box, 25th–75th percentiles; whiskers, ±1.5× IQR, each dot represents an individual tissue core. **j**. Percent body weight measurement of VCaP-CRPC tumor bearing mice (two-sided t-test, mean ± SEM). Vehicle: n = 9; enzalutamide (Enza): n=9; CBPD-409: n = 9; Enza+CBPD-409: n=9. **k**. Representative H&E and immunohistochemistry for the indicated protein from the VCaP-CRPC xenograft study. Scalebar = 50 μm. **l**. Immunoblots from VCaP-CRPC xenograft PD5 tumors. **m**. Percent body weight measurement throughout the treatment period from MDA-PCa-146-12 CRPC xenograft study (two-sided t-test, mean ± SEM). Vehicle: n = 6; Enza: n=6; Enza+CBPD-409: n=7. **n**. Individual tumor images from MDA-PCa-146-12 CRPC study. **o**. Representative staining images from the MDA-PCa-146-12 CRPC xenograft study. Scalebar = 200 μm. **p**. Tumor volume curve in the WA-74 CRPC xenograft model (two-sided t-test, mean ± SEM). Vehicle, n = 15; Enza, n=16; CBPD-409, n = 12. **q**. Representative immunohistochemistry from the WA-74 CRPC xenograft study. Scale bar = 50 μm.

# Reporting Summary

## Statistics

For all statistical analyses, confirm that the following items are present in the figure legend, table legend, main text, or Methods section.

| n/a | Confirmed | |
|---|---|---|
| ☐ | ☒ | The exact sample size (*n*) for each experimental group/condition, given as a discrete number and unit of measurement |
| ☐ | ☒ | A statement on whether measurements were taken from distinct samples or whether the same sample was measured repeatedly |
| ☐ | ☒ | The statistical test(s) used AND whether they are one- or two-sided<br>*Only common tests should be described solely by name; describe more complex techniques in the Methods section.* |
| ☐ | ☒ | A description of all covariates tested |
| ☐ | ☒ | A description of any assumptions or corrections, such as tests of normality and adjustment for multiple comparisons |
| ☐ | ☒ | A full description of the statistical parameters including central tendency (e.g. means) or other basic estimates (e.g. regression coefficient) AND variation (e.g. standard deviation) or associated estimates of uncertainty (e.g. confidence intervals) |
| ☐ | ☒ | For null hypothesis testing, the test statistic (e.g. *F*, *t*, *r*) with confidence intervals, effect sizes, degrees of freedom and *P* value noted<br>*Give P values as exact values whenever suitable.* |
| ☒ | ☐ | For Bayesian analysis, information on the choice of priors and Markov chain Monte Carlo settings |
| ☒ | ☐ | For hierarchical and complex designs, identification of the appropriate level for tests and full reporting of outcomes |
| ☐ | ☒ | Estimates of effect sizes (e.g. Cohen's *d*, Pearson's *r*), indicating how they were calculated |

*Our web collection on statistics for biologists contains articles on many of the points above.*

## Software and code

Policy information about availability of computer code

| | |
|---|---|
| Data collection | No software was used for data collection. |
| Data analysis | Computational tools used:<br>GraphPad Prism 9 and in-built statistical tools<br>PICARD Mark Duplicates (version 2.9.0)<br>HOMER (version v.4.10)<br>MACS2 (version 2.1.1.20160309)<br>BWA (version 0.7.17-r1198-dirty)<br>EdgeR (version 3.34.1)<br>EnhancedVolcano (version 1.15.0)<br>Limma (version 3.58.1.)<br>Fgsea (version 1.28.0)<br>Gtable (version 0.3.5)<br>Kallisto (version 0.50.1)<br>Gplots (version 3.1.3.1)<br>Trimmomatic (version 0.39)<br>Samtools (version  1.19)<br>Bedtools (version 2.31.1)<br>Deeptools (version 3.5.4)<br>ChIPpeakAnno (version 3.6.5)<br>ChIPseeker (version 1.35.3) |

Lranges (version 2.36.0)
Tidyverse (v2.0.0)
Ggplot2 (v3.5.1)

For manuscripts utilizing custom algorithms or software that are central to the research but not yet described in published literature, software must be made available to editors and reviewers. We strongly encourage code deposition in a community repository (e.g. GitHub). See the Nature Portfolio guidelines for submitting code & software for further information.

## Data

Policy information about availability of data

All manuscripts must include a data availability statement. This statement should provide the following information, where applicable:
- Accession codes, unique identifiers, or web links for publicly available datasets
- A description of any restrictions on data availability
- For clinical datasets or third party data, please ensure that the statement adheres to our policy

All data are available in the manuscript and the supplementary information. All sequencing data generated in this study have been deposited in the following National Center for Biotechnology Information Gene Expression Omnibus (NCBI GEO) repository: GSE255134. ChIP-seq datasets for normal, primary PCa, and mCRPC were obtained from GEO accessions GSE130408 and GSE70079. Gene dependency scores were analyzed using CRISPR knockout datasets available from the DepMap portal (https://depmap.org). The proteomics data have been deposited to the ProteomeXchange Consortium via the PRIDE partner repository with the dataset identifier PXD065228 for TMT-mass spectrometry and PXD065292 for acetyl-lysine proteomics.

## Research involving human participants, their data, or biological material

Policy information about studies with human participants or human data. See also policy information about sex, gender (identity/presentation), and sexual orientation and race, ethnicity and racism.

| Reporting on sex and gender | All archival tissues come from male patients that were treated for prostate cancer at the the University of Michigan Hospital in Ann Arbor, Michigan. |
| Reporting on race, ethnicity, or other socially relevant groupings | All patients are Caucasian white men who received treatment in North America. |
| Population characteristics | The human samples used in this study were archival, formalin-fixed paraffin-embedded (FFPE) prostate cancer specimens obtained from the departmental archive. All samples were fully de-identified prior to analysis. As prostate cancer occurs in males, the sex of all participants is male by definition. Other covariate information (e.g., age, clinical history) was not available or applicable due to the de-identified nature of the samples. |
| Recruitment | Patients' tissues were taken from Pathology archives at the University of Michigan Hospital in Ann Arbor, Michigan. |
| Ethics oversight | The University of Michigan Institutional Review Board approved the acquisition and use of clinical FFPE specimens from the pathology archives in this study. |

Note that full information on the approval of the study protocol must also be provided in the manuscript.

# Field-specific reporting

Please select the one below that is the best fit for your research. If you are not sure, read the appropriate sections before making your selection.

☒ Life sciences          ☐ Behavioural & social sciences          ☐ Ecological, evolutionary & environmental sciences

For a reference copy of the document with all sections, see nature.com/documents/nr-reporting-summary-flat.pdf

# Life sciences study design

All studies must disclose on these points even when the disclosure is negative.

| Sample size | Sample sizes were empirically and statistically determined. For animal experiments, n=10-20 tumors were used for the pilot and efficacy studies. Using >10 tumors per treatment group, the statistical power to detect a 50% decrease in the mean tumor volume or metastatic burden in the treatment group is estimated to be 92.3% if the coefficient of variation (CV) is 40%. All in vitro experiments were performed with at least 3 technical replicates across two independent experiments. All samples sizes for various assays are listed in the Methods section or the figure legends. For in vitro experiments, no formal statistical method was used to predetermine sample size. Instead, sample sizes were selected based on empirical experience and commonly accepted standards in the field to ensure reproducibility and adequate statistical power.<br>All in vitro experiments were performed with at least three technical replicates and repeated across at least two independent experiments. These sample sizes are considered sufficient to detect robust and reproducible biological effects. Sample sizes for each assay, including replicates and experimental design, are detailed in the Methods section and figure legends. |

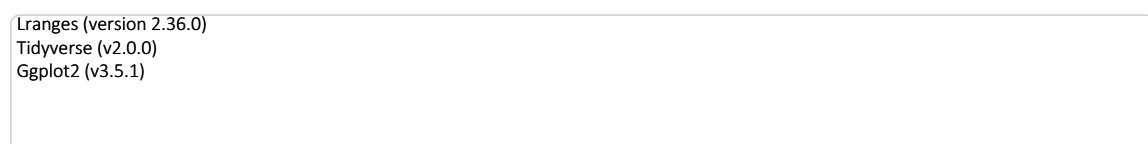

| Data exclusions | No data were excluded from the published publicly-available patient sequencing studies. For biological experiments, no data exclusions were made. |
| --- | --- |
| Replication | For all experiments, there are at least two independent biological repeats and multiple technical repeats in each. In all instances, all attempts at replicating the experiments produced similar results. |
| Randomization | For animal studies, mice were randomly assigned to treatment groups. For all other in vitro experiments, we used a common cell suspension to plate for both control and treatment groups.<br>For in vitro experiments, randomization was not applicable, as experiments were conducted using homogeneous cell populations under controlled conditions. Cells were allocated into experimental groups based on predefined treatment conditions (e.g., drug concentrations or genetic perturbations). All groups were processed in parallel using identical culture conditions to minimize variability. Given the uniform nature of the cell lines and experimental setup, there were no relevant covariates requiring additional control. |
| Blinding | All histo-pathological evaluations of tissues and IHC/staining-based scoring for drug toxicity studies were carried out in a blinded manner by two independent pathologists.<br>or all other experiments, blinding was not performed because data collection and quantification were conducted using automated instruments or standardized software-based pipelines (e.g., for imaging, RNA-seq, ChIP-seq, or proteomic analyses) with minimal to no subjective manual input. As such, investigator blinding was not relevant to these experiments. |

# Reporting for specific materials, systems and methods

We require information from authors about some types of materials, experimental systems and methods used in many studies. Here, indicate whether each material, system or method listed is relevant to your study. If you are not sure if a list item applies to your research, read the appropriate section before selecting a response.

## Materials & experimental systems

| n/a | Involved in the study |
| --- | --- |
| ☐ | ☒ Antibodies |
| ☐ | ☒ Eukaryotic cell lines |
| ☒ | ☐ Palaeontology and archaeology |
| ☐ | ☒ Animals and other organisms |
| ☒ | ☐ Clinical data |
| ☒ | ☐ Dual use research of concern |
| ☒ | ☐ Plants |

## Methods

| n/a | Involved in the study |
| --- | --- |
| ☐ | ☒ ChIP-seq |
| ☒ | ☐ Flow cytometry |
| ☒ | ☐ MRI-based neuroimaging |

## Antibodies

| Antibodies used | For WB: p300 (1:500, Invitrogen: MA1-16608, RW168, Vendor-validated); CBP (1:500, Invitrogen: PA5-27369, , Vendor-validated); AR (1:1000, Abcam: ab133273, ER179(2), Vendor-validated); H2BK5ac (1:1000, Cell Signaling Technology: 12799S, Vendor-validated); H2BK20ac (1:1000, Cell Signaling Technology: 34156S, vendor-validated); H2BK12ac (1:1000, Abcam: ab40883, EP858Y, Vendor-validated); H2BK16ac (1:1000, Abcam: ab177427, EPR17598, Vendor-validated); H3K27ac (1:1000, Cell Signaling Technology: 8173S, D5E4, Vendor-validated); H3K18ac (1:1000, Active Motif: 39755, Vendor-validated); H3K4me1 (1:1000, Abcam: ab8895, Vendor-validated); H3K4me3 (1:1000, Active Motif: 39060, Vendor-validated); H3K27me3(1:1000, Millipore: 07-449, Vendor-validated); H2B (1:1000, Active Motif: 39210, Vendor-validated); H3 (1:1000, Cell Signaling Technology: 3638S, 96C10, Vendor-validated); H2BK120ac (1:1000, Active Motif: 39119, Vendor-validated); Myc (1:1000, Cell Signaling Technology: 9402S, Vendor-validated); KLK3/PSA (1:1000, Dako: A0562, Vendor-validated); CITED2 (1:1000, Abcam: ab108345, EPR3416(2), Vendor-validated); NKX3-1 (1:1000, Cell Signaling Technology:83700S, D2Y1A, Vendor-validated); CCND1 (1:1000, Abcam: ab16663, SP4, Vendor-validated); FOXA1 (1:1000, Thermo Fisher Scientific: PA5-27157, Vendor-validated); Vinculin (1:1000, Cell Signaling Technology: 18799S, E1E9V, Vendor-validated) ; GAPDH (1:1000, Santa Cruz Biotechnology: sc-47724, 0411, Vendor-validated); BRD2 (1:1000, Bethyl Laboratories: A700-008,BL-167-2A2, Vendor-validated); BRD3 (1:1000, Bethyl Laboratories: A302-368A, Vendor-validated); BRD4 (1:1000, Bethyl Laboratories: A700-004CF, BL-149-2H5, Vendor-validated); GSPT1 (1:1000, Proteintech: 28130-1-ap, Vendor-validated); Aiolos (1:1000, Cell Signaling Technology: 15103S, D1C1E, Vendor-validated); Ikaros (1:1000, Cell Signaling Technology: 14859S, D6N9Y, Vendor-validated), HDAC3 (1:1000, Abcam, ab137704, Vendor-validated), Goat Anti-Rabbit IgG (H + L)-HRP Conjugate (1:10000, Biorad, #1706515), ECL Peroxidase (HRP) Anti-Mouse IgG (1:10000, Cytiva,NA931-100UL).<br>For ChIP-seq: H3K27ac (1ug/million cells, Diagenode: C15410196, Vendor-validated); p300 (2ug/million cells, Abcam: ab14984, G230 / NM-11, Vendor-validated); H2BK20ac (1ug/million cells, Cell Signaling Technology: 34156S, vendor-validated); H2BK5ac (1ug/million cells, Cell Signaling Technology: 12799S, Vendor-validated); H2BK16ac (1ug/million cells, Abcam, ab177427, EPR17598, Vendor-validated); AR (2ug/million cells, Millipore: 06-680, Vendor-validated); FOXA1 (2ug/million cells, Thermo Fisher Scientific: PA5-27157, Vendor-validated); BRD4 (2ug/million cells, Diagenode: C15410337, Vendor-validated); H3K18ac (1ug/million cells, Active Motif: 39755, Vendor-validated); MED1 (2ug/million cells, Active motif: 61065, Vendor-validated); ERG (2ug/million cells, Cell Signaling Technology: 97249S, A7L1G, Vendor-validated); RNA Pol II (1ug/million cells, Active motif: 39097, 4H8, Vendor-validated); Drosophila H2AV antibody for ChIP-seq Spike-in reference (1ug/25ug chromatin, Active Motif: 61686, Vendor-validated)<br>For IHC or IF: p300 (1:250, Invitrogen: 33-7600, NM11, Vendor-validated); CBP (1:100, Invitrogen: PA5-27369, Vendor-validated); AR (1:4000, Abcam: ab133273, EPR1535(2)); Ki67 (predilute, Ventana Medical Systems: 790-4286, 30-9, Vendor-validated); H2BK20ac (1:50, Abcam: ab177430, epr859, Vendor-validated); H2BK5ac (1:250, Abcam: ab40886, Vendor-validated); CCND1 (predilute, Cell Marque: 241R-18, SP4, Vendor-validated); KRT8 (1:50, Abcam: ab53280, EP1628Y); H3K27ac (1:200, Cell Signaling Technology: 8173S, D5E4, Vendor-validated), Anti-rabbit HRP (predilute, .OmniMap, 760-4311), Anti-mouse HRP (predilute, OmniMap, 760-4310); |
| --- | --- |

| Validation | All antibodies used in this study are from reputed commercial vendors and have been validated by the vendors (see website). QC data is directly available from all the vendors listed above, and these antibodies have been commonly used in other publications.
p300: https://www.thermofisher.com/antibody/product/p300-Antibody-clone-RW128-Monoclonal/MA1-16608, manufacturer states: This Antibody was verified by Knockdown to ensure that the antibody binds to the antigen stated.
CCND1:
CBP: https://www.thermofisher.com/antibody/product/CBP-Antibody-Polyclonal/PA5-27369, manufacturer states: This Antibody was verified by Knockdown to ensure that the antibody binds to the antigen stated.
AR:https://www.abcam.com/en-us/products/primary-antibodies/androgen-receptor-antibody-er1792-chip-grade-ab108341, manufacturer states: Advanced validation: For certain targets, additional application-specific testing is undertaken based on the most relevant methodologies for research into these targets. This additional validation testing data demonstrates efficacy to researchers when selecting products.
H2BK5ac: https://www.cellsignal.com/products/primary-antibodies/acetyl-histone-h2b-lys5-d5h1s-xp-rabbit-mab/12799?srsltid=AfmBOoo4Z-Or_MfINXDRYU6YN5FPHUPgGwDHpa3jE6n71wAZHvc8cc8B, manufacturer states: Acetyl-Histone H2B (Lys5) (D5H1S) XP® Rabbit mAb recognizes endogenous levels of histone H2B only when acetylated at Lys5. This antibody does not cross-react with other acetylated histones.
H2BK20ac: https://www.cellsignal.com/products/primary-antibodies/acetyl-histone-h2b-lys20-d7o9w-rabbit-mab/34156, manufacturer states: Acetyl-Histone H2B (Lys20) (D7O9W) Rabbit mAb recognizes endogenous levels of histone H2B protein when acetylated at Lys20. This antibody shows very slight cross-reactivity with histone H2B acetylated at Lys12.
H2BK12ac: https://www.abcam.com/en-us/products/primary-antibodies/histone-h2b-acetyl-k12-antibody-ep858y-ab40883?srsltid=AfmBOopfHUZXcMgiLrHng2CUt59dOszNQlpxaXx5KbSYUnRbA8yq3IIq. We validated this antibody by treating the cells with p300/CBP degraders and confirmed the loss of signal by wb.
H2BK16ac: https://www.abcam.com/en-us/products/primary-antibodies/histone-h2b-acetyl-k16-antibody-epr17598-chip-grade-ab177427#drawerView=highlights, manufacturer states: Advanced validation: For certain targets, additional application-specific testing is undertaken based on the most relevant methodologies for research into these targets. This additional validation testing data demonstrates efficacy to researchers when selecting products.
h3k27ac: https://www.cellsignal.com/products/primary-antibodies/acetyl-histone-h3-lys27-d5e4-xp-rabbit-mab/8173?srsltid=AfmBOorhibRWiQMGztdR2Yz3l9Nwia6wXCx0qi48PpbsDuPSyNE_pe_q, manufacturer states: Acetyl-Histone H3 (Lys27) (D5E4) XP® Rabbit mAb recognizes endogenous levels of histone H3 protein only when acetylated at Lys27. This antibody does not cross react with histone H3 acetylated at Lys9, 14, 18, 23, or 56. This antibody shows some cross-reactivity with acetyl-histone H2B lysine 5.
H3K18ac: https://www.activemotif.com/catalog/details/39755/histone-h3-acetyl-lys18-antibody-pab-3, manufacturer states: Applications Validated by Active Motif: ChIP: 10 µg per ChIP, ChIP-Seq: 4 µg each, ICC/IF: 1 µg/ml dilution, WB: 0.5 - 2 µg/ml dilution, DB: 0.1 - 0.5 µg/ml dilution
H3K4me1, https://www.abcam.com/en-us/products/primary-antibodies/histone-h3-mono-methyl-k4-antibody-chip-grade-ab8895?srsltid=AfmBOopgxryvWLRLafl8oXO_q5ZPjnd6RIXp74wxQUtkXG1ftMeoOQGN, the manufacturer states: Specific for Mono Methyl-K4. Does not Di Methyl-K4, Tri Methyl-K4 or Methyl-K9, Cited in over 1,100 publications, Trusted by researchers since 2002.
H3K4me3:https://www.activemotif.com/catalog/details/39159/histone-h3-trimethyl-lys4-antibody-pab, manufacturer states: Applications Validated by Active Motif: ChIP: 3 - 5 µl per ChIP, ChIP-Seq: 3 µl each, ICC/IF: 1:500 - 1:1,000 dilution, WB: 1:500 - 1:2,000 dilution, CUT&Tag: 1 µl per 50 µl reaction*, CUT&RUN: 1 µl per 50 µl reaction
H3K27me3: https://www.sigmaaldrich.com/US/en/product/mm/07449?srsltid=AfmBOorAvb EymMMPOF1qMbXr5HuR6mumQQRq-1Qb7NlrX04VESBTeJW, manufacturer states: Anti-trimethyl-Histone H3 (Lys27), also known as Anti-H3K27me3, is a highly published Rabbit Polyclonal Antibody. This protein A purified antibody is dot blot tested for trimethylated lysine 27 specificity and validated in WB, ICC, IP.
H2B: https://www.activemotif.com/catalog/details/39210/histone-h2b-antibody-pab, manufacturer states: Applications Validated by Active Motif: WB: 2 µg/ml dilution
H3: https://www.cellsignal.com/products/primary-antibodies/histone-h3-96c10-mouse-mab/3638?srsltid=AfmBOoqW82QOeXZGI5TVOfZNMZLO8lDnHkkR3yDERs8X9-kqtrnYd1jq, manufacturer states: Histone H3 (96C10) Mouse mAb detects endogenous levels of total Histone H3 protein, including isoforms H3.1, H3.2, and H3.3. The antibody does not cross-react with other histone proteins, including the Histone H3 variant CENP-A.
H2BK120ac: https://www.activemotif.com/catalog/details/39119/histone-h2b-acetyl-lys120-antibody-pab, manufacturer states: Applications Validated by Active Motif: ChIP: 10 µl per ChIP, ChIP-Seq: 5 µl each, WB*: 1:1,000 - 1:2,000 dilution, IF: 1:500 dilution.
Myc: https://www.cellsignal.com/products/primary-antibodies/c-myc-antibody/9402, manufacturer states: c-Myc Antibody detects endogenous levels of total c-Myc protein. This antibody is not recommended for detection of Myc-tagged fusion proteins (use Cell Signaling Technology cat. #2276 or #2278).
KLK3/PSA: https://www.citeab.com/antibodies/3382929-a0562-prostate-specific-antigen-psa, from citeab website: A polyclonal rabbit antibody, supplied by Dako, raised against Prostate-specific antigen (Human), cited in 99 publications, with 20 published images. Applications used include WB, IHC, ICC, ICC-IF, and 2 others.
CITED2: https://www.abcam.com/en-us/products/primary-antibodies/cited2-antibody-epr34162-ab108345?srsltid=AfmBOopYzwPzu1ZLHz2jH8km5rPGZNUMnviURy4BdRU3u-IVluOg92YK. We validated this antibody by using CITED2 siRNA and confirm the loss of signal by wb.
NKX3-1: https://www.cellsignal.com/products/primary-antibodies/nkx3-1-d2y1a-xp-rabbit-mab/83700, manufacturer states: NKX3.1 (D2Y1A) XP® Rabbit mAb recognizes endogenous levels of total NKX3.1 protein.
CCND1: https://www.abcam.com/en-us/products/primary-antibodies/cyclin-d1-antibody-sp4-ab16663?srsltid=AfmBOoryNSb9tl0ysvpoW6uCU1_v2iJV6hrFVJKaUnloONhtjMYGeeuq#drawerView=highlights. Manufacturer claims this is a KO validated antibody.
FOXA1: https://www.thermofisher.com/antibody/product/FOXA1-Antibody-Polyclonal/PA5-27157. Manufacturer states: This Antibody was verified by Relative expression to ensure that the antibody binds to the antigen stated  and Antibody specificity was demonstrated by detection of differential basal expression of the target across cell lines owing to their inherent genetic constitution. Enrichment of FOXA1 active binding region was observed in LNCaP cells in comparison to SW480 cells using Anti- FOXA1 Polyclonal Antibody (Product # PA5-27157) in chromatin immunoprecipitation.
Vinculin: https://www.cellsignal.com/products/antibody-conjugates/vinculin-e1e9v-xp-rabbit-mab-hrp-conjugate/18799?srsltid=AfmBOorV6mhoep8U_KdKI2_5v1XMGXsPElysTEwwEc-yalgv9CLE1v4A, manufacturer states: Vinculin (E1E9V) XP® Rabbit mAb (HRP Conjugate) recognizes endogenous levels of total vinculin protein. This antibody also reacts with metavinculin, a 145 kDa splice variant of vinculin.
GAPDH: https://www.scbt.com/p/gapdh-antibody-0411?srsltid=AfmBOopy1PEuNKckOJxVvbiiHZpv4mN-JksMdtk00a-8UOTvaEI7CZT_. |

This antibody has been cited in more than 3300 papers.

BRD2: https://www.fortislife.com/products/primary-antibodies/rabbit-anti-brd2-recombinant-monoclonal-antibody-bl-167-2a2/ BETHYL-A700-008. Manufacturer states: This antibody has been validated by Pillar 1: Independent Antibodies, Pillar 2: Complementary Assays, Pillar 4: Biological Characteristics.

BRD3: https://www.fortislife.com/products/primary-antibodies/rabbit-anti-brd3-antibody/BETHYL-A302-368?selected=A302-368A. Manufacturer states: this antibody has been validated to meet their strict performance standards. And this antibody has been cited in 52 papers.

BRD4: https://www.fortislife.com/products/primary-antibodies/rabbit-anti-brd4-recombinant-monoclonal-antibody-bl-149-2h5/ BETHYL-A700-004?selected=A700-004CF, manufacturer states that this antibody has been validated by Pillar 1: Independent Antibodies, Pillar 2: Complementary Assays. We also validated this antibody by treating the cells with BRD4 PROTAC degader and confirm the loss wb signal.

GSPT1: https://www.ptglab.com/products/GSPT1-Antibody-28130-1-AP.htm? srsltid=AfmBOopde6wBFxuVMSRA76-1gLZMH5u_IlD0JdnkwFjr6XZ8G91V5zur, manufacturer states: 28130-1-AP targets eRF3a/ GSPT1 in WB, IHC, IF/ICC, ELISA applications and shows reactivity with human, mouse, rat samples.

Aiolos: https://www.cellsignal.com/products/primary-antibodies/aiolos-d1c1e-rabbit-mab/15103, manufacturer states: Aiolos (D1C1E) Rabbit mAb recognizes endogenous levels of total Aiolos protein.

Ikaros: https://www.cellsignal.com/products/primary-antibodies/ikaros-d6n9y-rabbit-mab/14859, manufacturer states: Ikaros (D6N9Y) Rabbit mAb recognizes endogenous levels of total Ikaros protein.

HDAC3: https://www.citeab.com/antibodies/762917-ab137704-anti-hdac3-antibody. This antibody has been cited in 14 publications. We have validated this antibody by using siHDAC3.

MED1: https://www.activemotif.com/catalog/details/61065/med1-antibody-pab. We have validated this antibody by ChIP-seq and compared the MED1 peaks with a published dataset in the same cell line.

ERG: https://www.cellsignal.com/products/primary-antibodies/erg-a7l1g-rabbit-mab/97249. Manufacturer states: ERG (A7L1G) Rabbit mAb recognizes endogenous levels of total ERG protein. Based on sequence identity, this antibody should detect isoforms ERG1, ERG2 and ERG3. This antibody does not cross-react with Fli1. Non-specific staining was observed in human stomach epithelium by immunohistochemistry. This antibody is not recommended for immunohistochemical analysis of mouse tissues. We also validated this antibody by using siERG.

RNA Pol2: https://www.activemotif.com/catalog/details/39097/rna-pol-ii-antibody-mab. Manufacturer states: Applications Validated by Active Motif: ChIP: 10 μl per ChIP, ChIP-Seq: 6-20 μl each, WB: 1:2,000 - 1:5,000 dilution.

Drosophila H2AV antibody: https://www.activemotif.com/catalog/1091/chip-normalization. This antibody is a Spike-in reference antibody for ChIP-seq, which has been validated by Active motif.

p300: https://www.thermofisher.com/antibody/product/p300-Antibody-clone-NM-11-Monoclonal/33-7600. Manufacturer states: This Antibody was verified by Knockdown to ensure that the antibody binds to the antigen stated. Antibody specificity was demonstrated by siRNA mediated knockdown of target protein. HCT 116 cells were transfected with p300 siRNA and decrease in signal intensity was observed in ICC application using Anti-p300 Monoclonal Antibody (NM-11) (Product # 33-7600).

AR: https://www.abcam.com/en-us/products/primary-antibodies/androgen-receptor-antibody-epr15352-ab133273. We validated this antibody by siAR.

kI67: https://elabdoc-prod.roche.com/eLD/web/global/en/products/RTD000792? searchTerm=&catalog=ProductCatalog&orderBy=Relevance&fromResults=true. Manufacturer states: CONFIRM anti-Ki-67 (30-9) Rabbit Monoclonal Primary Antibody is intended for laboratory use in the qualitative immunohistochemical detection of Ki-67 protein by light microscopy in sections of formalin-fixed, paraffin-embedded tissue stained on a BenchMark IHC/ISH instrument. This product should be interpreted by a qualified pathologist in conjunction with histological examination, relevant clinical information, and proper controls. This antibody is intended for in vitro diagnostic (IVD) use.

CCND1: https://www.cellmarque.com/antibodies/CM/63/Cyclin-D1_SP4. This antibody has been validated by manufacturer and applied for the diagnosis of mantle cell lymphomas.

CK8: https://www.abcam.com/en-us/products/primary-antibodies/cytokeratin-8-antibody-ep1628y-cytoskeleton-marker-ab53280. Manufacturer states that this antibody has been validated by KO.

H3K27ac: https://www.abcam.com/en-us/products/primary-antibodies/histone-h3-acetyl-k27-antibody-chip-grade-ab4729? srsltid=AfmBOorViyxTaMVZTHNGhPvI93j11sGtG3b68nvTpWY_1ak3vjQQd9Xl. Manufacturer states: This antibody is tried and trusted by researchers since 2004. There are 2118 citations of this antibody.

# Eukaryotic cell lines

Policy information about cell lines and Sex and Gender in Research

| Cell line source(s) | Most cell lines were originally obtained from ATCC, DSMZ, ECACC, Lonza, or internal stock. CWR-R1 cells and a series of enzalutamide-resistant prostate cancer cell lines (LNCaP_Parental, LNCaP_EnzR) were provided by D. Vander Griend (University of Illinois at Chicago). Cells were grown in media conditions prescribed by ATCC, DSMZ, or ECACC. |
| --- | --- |
| Authentication | All cell lines were genotyped every month at the University of Michigan Sequencing Core using Profiler Plus (Applied Biosystems) and compared with corresponding short tandem repeat (STR) profiles in the ATCC database to authenticate their identity in culture between passages and experiments. |
| Mycoplasma contamination | All cell lines were biweekly tested to be free of mycoplasma contamination. |
| Commonly misidentified lines (See ICLAC register) | None |

# Animals and other research organisms

Policy information about studies involving animals; ARRIVE guidelines recommended for reporting animal research, and Sex and Gender in Research

| | |
|---|---|
| Laboratory animals | Efficacy studies: 4-6 week old male CB17 severe combined immunodeficiency (SCID) mice were procured from the University of Michigan breeding colony. Pharmacokinetics study: 9-11 week old CD-1 male mice, 9-12 week humanized cereblon C57BL/6 and 12 week-old male CD rats were used. All mice were maintained under the conditions of pathogen-free, 12 hours light/12 hours dark cycle, temperatures of 18-23°C, and 40-60% humidity. |
| Wild animals | No wild animals were used in the study. |
| Reporting on sex | Male animals were used since prostate cancer is specific to males. |
| Field-collected samples | No field collected samples were used in the study. |
| Ethics oversight | Institutional Review Board and Institutional Animal Care and Use Committee at the University of Michigan ensures that the highest animal welfare standards are maintained along with the conduct of accurate, valid scientific research through the supervision, coordination, training, guidance, and review of every project proposed to include the use of vertebrate animals at the University of Michigan. Protocl number PRO0001291 |

Note that full information on the approval of the study protocol must also be provided in the manuscript.

# Plants

| | |
|---|---|
| Seed stocks | Not applicable |
| Novel plant genotypes | Not applicable |
| Authentication | Not applicable |

# ChIP-seq

## Data deposition

☒ Confirm that both raw and final processed data have been deposited in a public database such as GEO.

☒ Confirm that you have deposited or provided access to graph files (e.g. BED files) for the called peaks.

| | |
|---|---|
| Data access links *May remain private before publication.* | All sequencing data generated in this study have been deposited in the following National Center for Biotechnology Information Gene Expression Omnibus (NCBI GEO) repository: GSE255134. |
| Files in database submission | These ChIP-seq Fastq files are submitted to the GEO: ChIP-seq: VCaP DMSO_H3K27ac VCaP CBPD_4hrs_H3K27ac VCaP GNE_4hrs_H3K27ac VCaP CCS_4h_H3K27ac VCaP_DMSO_AR VCaP_CBPD_4h_AR VCaP_GNE_4h_AR VCaP_DMSO_FOXA1 VCaP_CBPD_4h_FOXA1 VCaP_GNE_4h_FOXA1 VCaP_DMSO_H2BK5ac VCaP_CBPD_4hrs_H2BK5ac VCaP_GNE_4hrs_H2BK5ac VCaP_DMSO_H2BK20ac VCaP_CBPD_4hrs_H2BK20ac VCaP_GNE_4hrs_H2BK20ac VCaP_P300 VCaP_DMSO_BRD4 VCaP_MED1_AM VcaP_DMSO_Pol2 VcaP_CBPD_4h_Pol2 |

VcaP_GNE_4h_Pol2
VCaP_DMSO_H3K18ac
VCaP_CBPD_4hrs_H3K18ac
VCaP_GNE_4hrs_H3K18ac
VCap_Prathiba_May_HC7C2DRX3
DMSO_AR SI_28420
DMSO_FOXA1 SI_28421
AU15330_AR SI_28422
AU15330_FOXA1 SI_28423
VCaP_H2BK20ac_CST
DU145_H2BK20ac_CST
PC3_H2BK20ac_CST
VCaP_H2BK16ac_abcam
DU145_H2BK16ac_abcam
PC3_H2BK16ac_abcam
VCaP_DMSO_BRD4
VCaP_CBPD_4h_BRD4

**Genome browser session**
(e.g. UCSC)

N/A

## Methodology

**Replicates**

Multiple biological as well as technical replicates are included.

**Sequencing depth**

ATAC-seq: Sequenced to 65-70M total reads, paired-end mode, 125bp read lengths. Over 97% of uniquely mapped reads.
ChIP-seq: Sequenced to 50-70M total reads, paired-end mode, 125bp read lengths. Over 97% of uniquely mapped reads.
RNA-seq: Sequenced to 30-40M total reads, paired-end mode, 125bp read lengths. Over 97% of uniquely mapped reads.
EU-seq: Sequenced to 25-30M total reads, paired-end mode, 125bp read lengths. Over 97% of uniquely mapped reads.

**Antibodies**

For ChIP-seq, the following antibodies were used: H3K27ac (Diagenode: C15410196); p300 (Abcam: ab14984); H2BK20ac (Cell
Signaling Technology: 34156S); H2BK5ac (Cell Signaling Technology: 12799S); H2BK16ac (Abcam, ab177427); AR (Millipore: 06-680);
FOXA1 (Thermo Fisher Scientific: PA5-27157); BRD4 (Diagenode: C15410337); H3K18ac (Active Motif: 39755); MED1 (Active motif:
61065); ERG (Cell Signaling Technology: 97249S); RNA Pol II (Active motif: 39097), Drosophila H2AV antibody for ChIP-seq Spike-in
reference (Active Motif: 61686)

**Peak calling parameters**

MACS2 was used for peak calling with narrowpeak setting for narrow peaks and a second set of parameters for histone peaks (eg
H3K27Ac, --broad -B --cutoff-analysis --broad-cutoff 0.05 --max-gap 500).

**Data quality**

FastQC was used to quality check the raw sequencing data using standard metrics and default thresholds.

**Software**

ChIP-seq data analysis started with trimming using Trimmomatic version 0.39 (settings TruSeq3-PE-2.fa:2:30:10, minlen 50) [Trim].
BWA was used to align reads to hg38 (GRCh38) human genome reference ("bwa mem" command with options -5SP -T0, version
0.7.17-r1198-dirty) [bwa]. Alignments were filtered using samtools (quality score cutoff of 20) and picard MarkDuplicates (removed
duplicates) [picard, samtools]. MACS2 was used for peak calling with narrowpeak setting for narrow peaks and a second set of
parameters for histone peaks (eg H3K27Ac, --broad -B --cutoff-analysis --broad-cutoff 0.05 --max-gap 500) [MACS2]. In addition,
bedtools  was used to remove blacklisted regions of the genome from the peaks list (Encode's exclusion list ENCFF356LFX.bed)
[bedtools, encode]. UCSC's tool wigtoBigwig was used for conversion to bigwig formats [wigtoBigwig ].

