## [Peer Review File · Nature Genetics]

Targeting histone H2B acetylated enhanceosomes via p300/CBP degradation in prostate cancer

Corresponding Author: Dr Arul Chinnaiyan

Version 0:

Decision Letter:

31st Jan 2025

Dear Arul,

Your Article, entitled "Targeting histone H2B acetylated neo-enhanceosomes via p300/CBP degradation", has now been seen by 2 referees. You will see from their comments below that while they find your work of interest, some important points are raised. We are interested in the possibility of publishing your study in Nature Genetics, but would like to consider your response to these concerns in the form of a revised manuscript before we make a final decision on publication.

We therefore invite you to revise your manuscript taking into account all reviewer comments. Please highlight all changes in the manuscript text file. At this stage we will need you to upload a copy of the manuscript in MS Word .docx or similar editable format.

We are committed to providing a fair and constructive peer-review process. Do not hesitate to contact me if there are specific requests from the reviewers that you believe are technically impossible or unlikely to yield a meaningful outcome.

*2) If you have not done so already please begin to revise your manuscript so that it conforms to our Article format instructions, available

[here](http://www.nature.com/ng/authors/article_types/index.html).

*3) Include a revised version of any required Reporting Summary: <https://www.nature.com/documents/nr-reporting-summary.pdf>

EXTENDED DATA FIGURES

Link Redacted

Nature Genetics is committed to improving transparency in authorship. As part of our efforts in this direction, we are now requesting that all authors identified as 'corresponding author' on published papers create and link their Open Researcher and Contributor Identifier (ORCID) with their account on the Manuscript Tracking System (MTS), prior to acceptance. ORCID helps the scientific community achieve unambiguous attribution of all scholarly contributions. You can create and link your ORCID from the home page of the MTS by clicking on 'Modify my Springer Nature account'. For more information please visit please visit www.springernature.com/orcid.

Sincerely,

Tiago

Tiago Faial, PhD
Chief Editor
Nature Genetics
<https://orcid.org/0000-0003-0864-1200>

Reviewers' Comments:

Reviewer #1 (Remarks to the Author):

Summary

Luo and colleagues have undertaken a detailed study of the H2BNTac modification in prostate cancer, and have linked the levels of this histone modification to cancer cell line's level of dependence on CBP/p300 degradation.

Overall comments

This is a strong paper that uses a potent new CBP/p300 PROTAC to elucidate the relationship between H2BNTac and "enhancer addiction" in cancer. The potency in animal models is compelling, and several "enzalutamide resistant" models were used to drive home how clinically relevant this new therapy could be. More importantly, the authors give strong evidence to support classifying cancers along the axis of "enhancer addicted" (or not) by using H2BNTac as a clear biomarker. The superiority of the PROTAC compared to HAT inhibition or CBP/p300 bromodomain inhibition is not novel or surprising, but is nevertheless well done.

Data and methodology

The data and methodology are of high quality. No major concerns were identified.

Novelty/notability claims

1. The parts of the paper that describe p300 as a co-activator of AR are not novel (PMID: 32668201, PMID: 11931767, among others), despite the analysis being well done.
2. "H2B acetylation is markedly elevated in androgen receptor (AR)-positive prostate cancer". This is in my view one of the more important and novel findings of the paper. The increase in H2BNTac over benign sections is clear, and the comparison to other histone modifications make it clear that H2BNTac is likely more important for defining "enhancer addicted cancers" than the much more popular H3K27ac mark.
3. "cancers with elevated H2BNTac levels, including AR-positive prostate cancer, exhibit acute dependence on p300/CBP activity"- the data presented here that spans a wide array of cancers in their response to this CBP/p300 PROTAC gives a sense of broad applicability to this work, and nominates a "meta class" of cancers based on this criteria. While this has been shown to varying degrees elsewhere, this may be the best catalog of "enhancer driven cancers".
4. There are thousands of AR+p300 enhancers described herein, and the phrase "neo-enhanceosome" is used to describe this work. However, it seems that in order to make such a claim, the authors would need to compare AR and p300 binding sites in normal prostate tissue, as it seems likely that some fraction of the sites being studied here are not "neo" to mCRPC.
5. The authors make it sound like the "neo-enhancers" are being selectively targeted by CBP/p300 degradation, but it is more likely that all enhancers are being decommissioned. Are AR+p300+ERG and mCRPC unique enhancers being shut down more than other sets of enhancers, such as p300 enhancers lacking AR? If CBPD-409 has a non-targeted, global effect, this in my view may support the idea of targeting the overall hyper-H2BNTac epigenetic state rather than targeting the

subset of enhancers that are “neo”.

6. It would be exciting to see the “neo-enhanceosome” shown in terms of H2BK20ac ChIP-seq signal normalized by exogenous chromatin spike in (ChIP-Rx), comparing prostate cancer tissue compared to benign tissue, or PCa AR+ cell lines compared to PC3/DU145, showing a global increase of enhancer strength on a “per cell” level. Not a requirement, just a suggestion.

Minor points

7. While it is exciting to see reduced tumor growth with the CBP/p300 degrader, the concern would normally be that PROTACs would function well to degrade the human proteins but not the mouse orthologs, preventing the observation of known toxicity associated with CBP/p300 HAT inhibitors. It is shown here however that they observe “potent degradation of mouse p300/CBP orthologs”. Are these mice engineered with a humanized Cereblon?

8. The authors note that “p300/CBP degraders exhibit exceptional cytotoxicity in only select cancer cell lines, particularly those driven by enhancer-binding transcription factors”. A similar observation was made with a CBP/p300 PROTAC by Asante et al (PMID: 38102136) for rhabdomyosarcomas driven by the enhancer-binding transcription factor PAX3-FOXO1, and could be discussed to strengthen this point.

9. The observation that AR bound sites lacking p300 have less acetylation than AR+p300 sites is not surprising and it doesn't seem novel – I'm not sure why the paper pays so much attention to this. Figure 2 in particular doesn't need to include so many nearly redundant heatmaps. In addition

10. Do genes associated with AR+p300 have a higher gene expression than genes that only have AR?

11. The paper could benefit from having a IGV-style genome browser example that shows a well known AR target gene and its surrounding enhancers, with the ChIP-seq and RNA-seq from cell treated with DMSO v CBPD-409.

Figure comments

12. Figure 2B – instead of “super enhancer score”, the y-axis should indicate what the number units are (reads per million mapped reads, RPM?). Something like “H3K27ac ChIP-seq signal (RPM)”.

13. In Figure 4D, the first row appears to be one of the only genes that still responds to R1881, even with CBPD-409 treatment. Could you add that gene's name to the plot, as you have already highlighted the more well known AR target genes?

14. The “hook” effect is visible in Figure 5B for LNCaP and 22Rv1, but not mentioned in the results section describing this data. It could be helpful to explain why this is an expected consequence of the PROTAC approach.

15. Throughout the main figures, text appears to be squeezed or stretched – I recommend trying to make the width/height ratio of all text the same.

Reviewer #2 (Remarks to the Author):

This paper explores the role of p300 in prostate cancer. The authors look for regulated histone modifications and focus on H2B N-terminal modifications, where acetylation events are known to occur. The authors show that p300 is required for cell viability in public datasets and associate p300 levels with H2B acetylation events. p300 binding was shown to correlate with the strongest AR and ERG binding sites and these co-bound regions were shown to be the most accessible and near the lineage-defining regulatory regions. Depletion of p300 caused a downregulation of H3 and H2B acetylation, with a further reduction observed following inhibition of both p300 and CBP. By exploiting two reader bromodomain inhibitors, the authors show that H2K27Ac was depleted but H2B acetylation was not, suggesting that these inhibitors of reader activity are not sufficient to completely block activity of p300/CBP. The authors therefore use one of the bromodomain inhibitors as the basis to make a PROTAC, which they call CBPD-409 and they conduct a series of experiments to gain confidence in the specificity and mode of action of this inhibitor. Importantly, the authors show that CBPD-409 can block acetylation events on both H3 and H2B targets, whereas the precursor inhibitors could not inhibit H2B acetylation events. This novel inhibitor was shown to target acetylation at key AR associated genes, although chromatin accessibility was not affected, suggesting that CBPD-409 does not target transcription factor activity on chromatin. As would be expected, p300/CBP degradation blocked AR transcriptional targets. A series of growth inhibitory experiments were conducted which showed that CBPD-409 can block prostate cancer cells, but not NEPC models of AR negative models. This was expanded to a large cell line panel which revealed that ~6% of all models showed sensitivity to the CBPD-409 inhibitor but most models were resistant, suggesting that p300/CBP dependence is not universal. Toxicity studies were conducted in mice and rats, confirming the encouraging in vivo profile of this inhibitor. Finally, a couple of efficacy studies were undertaken, supporting the use of CBPD-409 in xenograft models.

This is a very good paper and although the link with p300 in prostate cancer is well established, the development of the CBPD-409 inhibitor is a major advance. The work characterizing and validating this new inhibitor is compelling and the conclusions are valid and exciting.

- Does the cell line inhibitory profile match the CRISPR data (i.e. from DepMap) showing that p300 or CBP dependence is aligned with the responsiveness to CBPD-409? Or do both p300 and CBP need to be inhibited to see this effect?

- The development of CBPD-409 is a substantial achievement. As a resource to learn about AR biology, what does CBPD-409 treatment do to the recruitment of known AR-associated co-factors? Can the author learn about the hierarchy in co-factor recruitment using this reagent? And does the loss of p300/CBP induce recruitment of opposing (potentially repressive) co-factors?

Version 1:

Decision Letter:

Our ref: NG-A67279R

29th May 2025

Dear Dr Chinnaiyan,

First, please accept my apologies for the delay in returning this decision to you. Thank you for your patience.

Thank you for submitting your revised manuscript "Targeting histone H2B acetylated enhanceosomes via p300/CBP degradation" (NG-A67279R). It has now been seen by the original referees and their comments are below. The reviewers find that the paper has improved in revision, and therefore we'll be happy in principle to publish it in Nature Genetics, pending minor revisions to comply with our editorial and formatting guidelines.

Sincerely,

Safia Danovi, PhD
Senior Editor, Nature Genetics
ORCID: 0009-0007-7822-5479

Reviewer #1 (Remarks to the Author):

The authors have thoughtfully and thoroughly addressed all of my concerns, and have "gone the extra mile" at every opportunity. I have no remaining concerns, and highly recommend this important work for publication.

Reviewer #2 (Remarks to the Author):

The authors have produced an excellent manuscript. Although they didn't do the AR ChIP-seq as requested, the BRD4 data is sufficient.

NG-A67279: “Targeting histone H2B acetylated enhanceosomes via p300/CBP degradation”

Responses to Reviewers' Comments

Reviewer #1:

Overall comments

This is a strong paper that uses a potent new CBP/p300 PROTAC to elucidate the relationship between H2BNTac and “enhancer addiction” in cancer. The potency in animal models is compelling, and several “enzalutamide resistant” models were used to drive home how clinically relevant this new therapy could be. More importantly, the authors give strong evidence to support classifying cancers along the axis of “enhancer addicted” (or not) by using H2BNTac as a clear biomarker. The superiority of the PROTAC compared to HAT inhibition or CBP/p300 bromodomain inhibition is not novel or surprising but is nevertheless well done.

Data and methodology

The data and methodology are of high quality. No major concerns were identified.

We appreciate the reviewer's encouraging feedback on the importance of H2BNTac as a distinct biomarker for enhancer-dependent cancers and the clinical impact of our study. Their suggestions have strengthened our work. Below, we provide detailed point-by-point responses outlining all modifications made to the manuscript. Additionally, in the rebuttal figure legends, we have indicated the corresponding manuscript figures where revised data have been incorporated, with the relevant new text highlighted in blue in the manuscript.

Novelty/notability claims

1. The parts of the paper that describe p300 as a co-activator of AR are not novel (PMID: 32668201, PMID: 11931767, among others), despite the analysis being well done.

Response: We thank the reviewer for this comment. We have cited the indicated studies in our revised work. While p300's role in AR transactivation and the AR/p300/SRC3 complex structure is known, its genome-wide requirement for AR function remains unclear.

Our cistrome analysis reveals that only ~50% of AR enhancers are co-occupied by p300, suggesting its selective role in orchestrating enhancer activity rather than universal AR binding. Notably, AR/ERG hyperactive enhancers show strong p300 dependence, where its enzymatic function is critical for transcriptional activation.

Using CBPD-409, we demonstrate that p300 is essential for nucleosome acetylation and RNA Pol II recruitment but dispensable for AR binding and chromatin accessibility, highlighting its function as a transcriptional amplifier of oncogenic programs in prostate cancer.

2. “H2B acetylation is markedly elevated in androgen receptor (AR)-positive prostate cancer”. This is in my view one of the more important and novel findings of the paper. The increase in H2BNTac over benign sections is clear, and the comparison to other histone modifications

make it clear that H2BNTac is likely more important for defining “enhancer addicted cancers” than the much more popular H3K27ac mark.

Response: We appreciate the reviewer’s recognition of the significance of this finding regarding elevated H2BNTac levels in prostate cancer. As stated by the reviewer, our study highlights H2BNTac as a more specific enhancer mark than H3K27ac, strongly correlating with p300 levels and distinctively identifying enhancer-addicted cancers. Notably, p300/CBP degradation selectively and almost completely depletes H2BNTac, effectively disabling the AR enhanceosome. These findings position H2BNTac as a key regulator of enhancer function and a promising therapeutic target in prostate cancer. We further emphasized these novel findings in the revised manuscript text.

3. “Cancers with elevated H2BNTac levels, including AR-positive prostate cancer, exhibit acute dependence on p300/CBP activity”- the data presented here that spans a wide array of cancers in their response to this CBP/p300 PROTAC gives a sense of broad applicability to this work, and nominates a “meta class” of cancers based on this criterion. While this has been shown to varying degrees elsewhere, this may be the best catalog of “enhancer driven cancers”.

Response: We appreciate the reviewer’s recognition of the broad applicability of our findings. We agree that our study supports the classification of enhancer-driven cancers based on H2BNTac dependence, highlighting their sensitivity to p300/CBP degradation and reinforcing the role of elevated H2BNTac as a key biomarker of enhancer addiction.

4. There are thousands of AR+p300 enhancers described herein, and the phrase “neo-enhanceosome” is used to describe this work. However, it seems that in order to make such a claim, the authors would need to compare AR and p300 binding sites in normal prostate tissue, as it seems likely that some fraction of the sites being studied here are not “neo” to mCRPC.

Response: We appreciate the reviewer’s important comment regarding the definition of the neo-enhanceosome. To more carefully evaluate the effect of CBP/p300 inactivation on the neo-enhanceosome, we leveraged AR cistrome data from patient tissues to distinguish CRPC-specific AR enhancers from normal AR enhancers^{1,2}. Using AR ChIP-seq data from 15 CRPC patient samples and 7 benign prostate tissues, we identified 64,175 unique AR binding sites in CRPC tissues, 7,443 AR binding sites shared between CRPC and benign prostate tissues, and 3,755 AR binding sites exclusive to benign prostate tissues (shown below as **Figure R1A** and **Extended Figure 5F** in the revised manuscript). Based on this classification, we define the 64,175 CRPC-specific AR binding sites as **AR neo-enhancers** and the 7,443 shared AR binding sites along with the 3,755 benign-specific sites as **AR normal enhancers**, given their presence in benign prostate tissues.

We then analyzed p300 ChIP-seq peaks at AR neo-enhancers and normal enhancers in VCaP cells. As anticipated by the reviewer, our results show no significant difference in p300 enrichment between AR neo-enhancers and normal enhancers, suggesting that p300 is not an exclusive cofactor of AR neo-enhancers (shown below as **Figure R1B** and **Extended Figure 5G** in the revised manuscript). Instead, our data indicate that p300/AR co-bound enhancers in

mCRPC exist in a hyperactivated epigenetic state, characterized by significant enrichment of transcriptional cofactors, hyper H2BNT acetylation, and increased transcriptional output (**Figure 2A**). Thus, following the reviewer’s suggestion, we have replaced the phrase “neo-enhanceosome” with “hyper-activated enhancers” in the revised manuscript. This phrase more precisely defines the AR/p300 co-bound enhancers and their contribution to amplifying the oncogenic gene programs.

Figure R1. (A) The Venn diagram of AR ChIP-seq peaks in CRPC tissues and benign prostate tissues. (B) ChIP-seq read-density heatmaps of p300 at AR neo-enhancers and AR normal enhancers in VCaP cells.

This new data is added as Extended Figure 5F-G in the revised manuscript.

5. The authors make it sound like the “neo-enhancers” are being selectively targeted by CBP/p300 degradation, but it is more likely that all enhancers are being decommissioned. Are AR+p300+ERG and mCRPC unique enhancers being shut down more than other sets of enhancers, such as p300 enhancers lacking AR? If CBPD-409 has a non-targeted, global effect, this in my view may support the idea of targeting the overall hyper-H2BNTac epigenetic state rather than targeting the subset of enhancers that are “neo”.

Response: We appreciate the reviewer’s insightful comment. Our ChIP-seq data indicate that CBPD-409 not only represses H2BK20ac at ERG/AR/p300 co-bound neo-enhancers but also decommissions H2BK20ac at enhancers bound solely by p300, supporting the reviewer’s idea that CBPD-409 broadly disrupts all H2BNTac-marked enhancers (shown below as **Figure R2** and **Extended Figure 5D** in the revised manuscript). In light of this data and the analyses

conducted in response to comment #4, we have revised the text to replace the term 'neo-enhancer' with 'hyper-activated' enhancers, and now propose p300/CBP inactivation to target the elevated H2BK20ac epigenetic state in enhancer-addicted cancers. Additionally, we have updated the title and abstract of our paper to reflect this important conclusion.

Figure R2. ChIP-seq read-density heatmaps of H2BK20ac at ERG/AR/p300 co-bound sites and p300 only binding sites in VCaP cells.

This new data is added as Extended Figure 5D in the revised manuscript.

6. It would be exciting to see the “neo-enhanceosome” shown in terms of H2BK20ac ChIP-seq signal normalized by exogenous chromatin spike in (ChIP-Rx), comparing prostate cancer tissue compared to benign tissue, or PCa AR+ cell lines compared to PC3/DU145, showing a global increase of enhancer strength on a “per cell” level. Not a requirement, just a suggestion.

Response: We appreciate the reviewer’s insightful comment. Since no ChIP-seq grade antibody is available to detect H2BK20ac in both human and Drosophila cells, performing an ideal ChIP-Rx experiment, which requires an antibody recognizing the target in both the sample and spike-in species, is not feasible. To address this, we conducted H2BK20ac and H2BK16ac ChIP-seq using Drosophila chromatin as a spike-in reference, along with a Drosophila-specific H2A.v antibody, in AR-positive (VCaP) and AR-negative (PC3/DU145) cells.

Our results reveal significantly more than 5-fold higher H2BK20ac and H2BK16ac signals at AR/p300 co-bound regions in VCaP cells compared to DU145 and PC3 cells, indicating the elevated activity of hyperactivated enhancers in AR-positive prostate cancer (shown below as **Figure R3** and **Extended Figure 8N-O** in the revised manuscript).

Figure R3. (A) ChIP-seq read-density heatmaps of H2BK20ac and H2BK16ac at AR/p300 shared non-promoter regions in VCaP, DU145, and PC3 cells. (B) ChIP-seq tracks of H2BK20ac and H2BK16ac within the *MYC* gene locus in VCaP, DU145, and PC3 cells. *This new data is added as Extended Figure 8N-O in the revised manuscript.*

Minor points

7. While it is exciting to see reduced tumor growth with the CBP/p300 degrader, the concern would normally be that PROTACs would function well to degrade the human proteins but not the mouse orthologs, preventing the observation of known toxicity associated with CBP/p300 HAT inhibitors. It is shown here however that they observe “potent degradation of mouse p300/CBP orthologs”. Are these mice engineered with a humanized Cereblon?

Response: We appreciate this important comment. While our mice do not express humanized cereblon, IHC analysis across all tested vital organs confirms that CBPD-409 effectively degrades p300/CBP in CD1 mice (**Extended Figure 9A**). Despite its potent on-target degradation, no evident toxicity was observed, supporting CBPD-409’s favorable safety profile. Additionally, strong p300/CBP degradation was observed in CBPD-409–treated CD1 rats, further demonstrating its robust on-target activity in murine cells (**Extended Figure 9O**).

To thoroughly address this question, we profiled the safety of CBPD-409 in the humanized CRBN (hCRBN) mouse model (*Crbn*^{V380E/I391V} C57BL/6 mouse)³. In hCRBN mice, CBPD-409 exhibits marked degradation effects on p300/CBP proteins in different organs (shown below as **Figure R4A** and **Extended Figure 9J** in the revised manuscript). Notably, there were no significant changes in body weight in CBPD-409 treated hCRBN mice (shown below as **Figure R4B** and **Extended Figure 9K** in the revised manuscript). Serum biochemistry analyses and complete blood cell counts showed no significant changes following CBPD-409 treatment (**Figure R4C-D** and **Extended Figure 9L-M** in the revised manuscript). Histological

examination revealed no evident toxicity in hCRBN mice organs (shown below as **Figure R4E** and **Extended Figure 9N** in the revised manuscript), supporting the excellent safety profile of CBPD-409.

Figure R4. (A) Representative immunohistochemistry images showing staining of murine CBP and p300 in specified organs of humanized cereblon C57BL/6 mice treated with CBPD-409 (3 mg/kg, administered orally three days per week). n=6 per treatment. Scalebar = 20 μ m. (B) hCRBN mice body weight (%) measurements throughout the treatment period from vehicle and CBPD-409 treated groups (two-sided t-test). Data are presented as mean \pm SEM (vehicle: n = 5, CBPD-409: n=6). (C) Liver and kidney function tests from CBPD-409 hCRBN mouse PD10 study. ALB: albumin, CREA: creatinine; BUN: blood urea nitrogen; Tbili: bilirubin. Vehicle: n=5; CBPD-409: n=6; two-sided t-test. (D) Complete blood count from CBPD-409 hCRBN mouse PD10 study. (E) Representative H&E staining of major organs from hCRBN mouse study. Scalebar = 20 μ m.

This new data is added as Extended Figure 9J-N in the revised manuscript.

8. The authors note that “p300/CBP degraders exhibit exceptional cytotoxicity in only select cancer cell lines, particularly those driven by enhancer-binding transcription factors”. A similar observation was made with a CBP/p300 PROTAC by Asante et al (PMID: 38102136) for

rhabdomyosarcomas driven by the enhancer-binding transcription factor PAX3-FOXO1 and could be discussed to strengthen this point.

Response: We appreciate the insightful suggestion from the reviewer. The study by Asante et al. demonstrated that p300/CBP degradation selectively impairs RMS cells harboring the PAX3-FOXO1 fusion, which aligns with our findings suggesting that targeting p300/CBP preferentially suppresses enhancer-binding transcription factors⁴. We have cited this important study in the revised manuscript and added mention of the findings in the Discussion section.

9. The observation that AR bound sites lacking p300 have less acetylation than AR+p300 sites is not surprising and it doesn't seem novel – I'm not sure why the paper pays so much attention to this. Figure 2 in particular doesn't need to include so many nearly redundant heatmaps.

Response: We thank the reviewer for their comment. While it is well-established that p300 is a key histone acetyltransferase (HAT) that functions as a cofactor for AR transactivation, it is not the only HAT involved in AR signaling. For example, PCAF has also been shown to function as an AR cofactor, catalyzing histone acetylation at AR enhancers^{5,6}. Our data demonstrate that at AR-bound sites lacking p300 co-occupancy, H2BNTac is dramatically reduced, confirming that H2BNTac is uniquely dependent on p300 activity and suggesting that p300 is the dominant HAT required to maintain histone acetylation and the chromatin landscape at hyperactivated AR enhancers. As suggested by the reviewer, to improve clarity and avoid redundancy, we refined **Figure 2** to present only the most essential histone mark heatmaps (H3K27ac and H2BK20ac).

In addition, do genes associated with AR+p300 have a higher gene expression than genes that only have AR?

Response: We appreciate the reviewer's insightful question. To address this, we analyzed our previously published Hi-ChIP dataset (PMID: 34937944) to identify genes regulated by AR/p300-shared enhancers versus those regulated by AR-only enhancers. We defined these gene sets based on enhancer-promoter interactions captured by the H3K4me3 HiChIP-seq assay. Notably, we found a significantly higher proportion of AR/p300 co-bound enhancers loop to active promoter regions compared to AR-only enhancers (**Figure R5A** and **Extended Figure 2F** in the revised manuscript), with over 40% of AR/p300 sites making promoter contacts. Notably, AR/p300 target genes include several well-established AR target genes like *KLK2*, *TMPRSS2*, and *ZBTB16*, highlighting their predominant role in enhancing the AR target gene programs. Consistent with this data, in the nascent RNA-seq data, we found that target genes of AR/p300-shared enhancers exhibit higher mRNA expression than those regulated by AR-only enhancers. This confirms that AR/p300 co-bound enhancers are more transcriptionally competent and underlines that role of CBP/p300-catalyzed H2BNTac as a marker of hyperactivated enhancers in AR+ prostate cancer (**Figure R5B** and **Extended Figure 2G** in the revised manuscript).

Figure R5. (A) Bar chart showing the percentage of enhancers looping to promoter versus non-promoter regions. (B) Bar chart illustrating the nascent RNA expression levels of genes regulated by AR-only enhancers versus AR/p300-shared enhancers.

This new data is added as Extended Figure 2F-G in the revised manuscript.

11. The paper could benefit from having an IGV-style genome browser example that shows a well-known AR target gene and its surrounding enhancers, with the ChIP-seq and RNA-seq from cells treated with DMSO v CBPD-409.

Response: We appreciate the reviewer's insightful comment. As shown below in **Figure R6** and **Extended Figure 6F** in the revised manuscript, we present an IGV-style genome browser view illustrating AR and p300 co-occupancy at the *TMPRSS2* enhancer region. Our data show that treatment with CBPD-409 leads to a significant reduction in H3K27ac and H2BK20ac levels at this enhancer. RNA Pol II loading at the *TMPRSS2* gene body is inhibited by CBPD-409. This visualization provides clear evidence of how CBPD-409 disrupts AR enhancer function and transcriptional output.

Figure R6: ChIP-seq tracks of AR, p300, H3K27ac, H2BK20ac, and Pol II within the *TMPRSS2* gene locus in VCaP cells treated with 100nM CBPD-409 or 1 μ M GNE-049 for 4 hours.

This data is added as Extended Figure 6F in the revised manuscript.

Figure comments

12. Figure 2B – instead of “super enhancer score”, the y-axis should indicate what the number units are (reads per million mapped reads, RPM?). Something like “H3K27ac ChIP-seq signal (RPM)”.

Response: We thank the reviewer for this suggestion. The super-enhancer score represents the summed H3K27ac ChIP-seq signal across constituent enhancers. Since the signal is normalized, the y-axis has been relabeled as suggested to “H3K27ac ChIP-seq signal (RPM)”.

13. In Figure 4D, the first row appears to be one of the only genes that still responds to R1881, even with CBPD-409 treatment. Could you add that gene’s name to the plot, as you have already highlighted the more well-known AR target genes?

Response: We appreciate the reviewer’s important observation. In CBPD-409–treated cells, *HPGD* (hydroxyprostaglandin dehydrogenase) remains R1881-responsive and is now labeled in **Figure 4D**.

14. The “hook” effect is visible in Figure 5B for LNCaP and 22Rv1, but not mentioned in the results section describing this data. It could be helpful to explain why this is an expected consequence of the PROTAC approach.

Response: We appreciate the reviewer’s insightful observation. The hook effect, where target degradation is reduced at higher PROTAC concentrations, is a well-documented phenomenon in PROTAC-mediated protein degradation⁷. This occurs due to excess PROTAC molecules leading to a decrease in ternary complex formation, which is necessary for effective ubiquitination and proteasomal degradation.

In Figure 5B, we observe this effect in LNCaP and 22Rv1 cells, where cytotoxicity is less at the highest PROTAC concentrations. This is an expected consequence of the PROTAC mechanism, as at high concentrations, PROTACs can saturate the target protein and E3 ligase independently, reducing the probability of ternary complex formation and subsequent ubiquitination.

We clarified this observation in the results section to explicitly discuss the hook effect and its implications for PROTAC dosing.

15. Throughout the main figures, text appears to be squeezed or stretched – I recommend trying to make the width/height ratio of all text the same.

Response: We thank reviewer the constructive feedback. We appreciate reviewer’s attention to detail and revised the figures to ensure a consistent width/height ratio for all text.

Reviewer #2:

This paper explores the role of p300 in prostate cancer. The authors look for regulated histone modifications and focus on H2B N-terminal modifications, where acetylation events are known to occur. The authors show that p300 is required for cell viability in public datasets and associate p300 levels with H2B acetylation events. p300 binding was shown to correlate with the strongest AR and ERG binding sites and these co-bound regions were shown to be the most accessible and near the lineage-defining regulatory regions. Depletion of p300 caused a downregulation of H3 and H2B acetylation, with a further reduction observed following inhibition of both p300 and CBP. By exploiting two reader bromodomain inhibitors, the authors show that H2K27Ac was depleted but H2B acetylation was not, suggesting that these inhibitors of reader activity are not sufficient to completely block activity of p300/CBP. The authors therefore use one of the bromodomain inhibitors as the basis to make a PROTAC, which they call CBPD-409 and they conduct a series of experiments to gain confidence in the specificity and mode of action of this inhibitor. Importantly, the authors show that CBPD-409 can block acetylation events on both H3 and H2B targets, whereas the precursor inhibitors could not inhibit H2B acetylation events. This novel inhibitor was shown to target acetylation at key AR associated genes, although chromatin accessibility was not affected, suggesting that CBPD-409 does not target transcription factor activity on chromatin. As would be expected, p300/CBP degradation blocked AR transcriptional targets. A series of growth inhibitory experiments were conducted which showed that CBPD-409 can block prostate cancer cells, but not NEPC models of AR negative models. This was expanded to a large cell line panel which revealed that ~6% of all models showed sensitivity to the CBPD-409 inhibitor but most models were resistant, suggesting that p300/CBP dependence is not universal. Toxicity studies were conducted in mice and rats, confirming the encouraging in vivo profile of this inhibitor. Finally, a couple of efficacy studies were undertaken, supporting the use of CBPD-409 in xenograft models. This is a very good paper and although the link with p300 in prostate cancer is well established, the development of the CBPD-409 inhibitor is a major advance. The work characterizing and validating this new inhibitor is compelling and the conclusions are valid and exciting.

We appreciate the reviewer's positive feedback on CBPD-409 as a potent inhibitor of p300/CBP activity and its selective ability to suppress cancer cell growth. Their suggestions have helped refine our work. Below, we provide point-by-point responses detailing all revisions made to the manuscript. In the rebuttal figure legends, we have also noted the corresponding manuscript figures where revised data have been incorporated, with the updated text highlighted in blue in the manuscript file.

1. Does the cell line inhibitory profile match the CRISPR data (i.e. from DepMap) showing that p300 or CBP dependence is aligned with the responsiveness to CBPD-409? Or do both p300 and CBP need to be inhibited to see this effect?

Response: We thank the reviewer for this important comment. As shown in **Figure 5F**, our analysis integrating DepMap CRISPR screening data revealed a strong positive correlation between p300 gene dependency and CBPD-409 sensitivity. This trend is weaker and not

statistical significant for CBP gene dependency across cancer cell lines, likely suggesting that p300 is the dominant enzyme that catalyzes H2BNTac in most cancer models in the DepMap database. Notably, this is corroborated by us using genetic approaches in prostate cancer models, where knock out of the p300 gene alone markedly reduced the levels of H2BNTac compared to CBP inactivation in LNCaP cells (**Figure 3A**).

To address the second half of the reviewer question, we believe that both p300 and CBP need to be inhibited for maximal cytotoxicity in tumor cells. While p300 appears to be the dominant writer of H2BNTac histone marks, using genetic tools, in 22RV1 cells we found both p300 and CBP to equally contribute to maintaining the hyper-H2BNTac epigenetic state (**Extended Figure 3A**). Consistently, inhibiting both p300 and CBP achieved the strongest suppression of growth in 22RV1 cells. Altogether, this suggests that enzymatic contributions from either p300 or CBP paralogs might vary across cancers, thus requiring combined inhibition of both enzymes to trigger a cytotoxic response. We have now added this interpretation of our data in the discussion section.

The development of CBPD-409 is a substantial achievement. As a resource to learn about AR biology, what does CBPD-409 treatment do to the recruitment of known AR-associated cofactors? Can the author learn about the hierarchy in co-factor recruitment using this reagent? And does the loss of p300/CBP induce recruitment of opposing (potentially repressive) cofactors?

Response: We appreciate the reviewer's suggestions. To assess the effect of p300/CBP degradation on AR cofactor recruitment, we performed ChIP-seq of the key AR cofactor BRD4. CBPD-409 treatment significantly reduced BRD4 loading at AR enhancers (shown below in **Figure R7A** and **Extended Figure 5J** in the revised manuscript). Integrating our ChIP-seq (AR, FOXA1, histone modifications, RNA Pol II) and ATAC-seq data, we conclude that p300/CBP regulates histone modifications, BRD4, and transcriptional machinery loading without affecting AR and FOXA1 binding. These results support a hierarchical assembly of the AR enhancer complex in the following order: AR/FOXA1 → p300/CBP → BRD4 → RNA Pol II.

Furthermore, co-immunoprecipitation experiments revealed that CBPD-409 treatment increases the interaction between AR and HDAC3, which is a core component of the NCOR1 repressive complex^{8,9}, suggesting that loss of p300/CBP promotes the recruitment of repressive cofactors (shown below in **Figure R7B** and **Extended Figure 5K** in the revised manuscript).

Figure R7: (A) ChIP-seq read-density heatmaps of BRD4 at AR/p300 co-bound cis-regulatory elements in VCaP cells with 4 hours of 100 nM CBPD-409 treatment. (B) Immunoblot analysis of the co-immunoprecipitation (Co-IP) assay showing that 100 nM CBPD-409 increases the interaction between AR and HDAC3 proteins.

This new data is added as Extended Figure 5J-K in the revised manuscript.

References

1. Pomerantz, M.M. *et al.* The androgen receptor cistrome is extensively reprogrammed in human prostate tumorigenesis. *Nat Genet* **47**, 1346-51 (2015).
2. Pomerantz, M.M. *et al.* Prostate cancer reactivates developmental epigenomic programs during metastatic progression. *Nat Genet* **52**, 790-799 (2020).
3. Sellar, R.S. *et al.* Degradation of GSPT1 causes TP53-independent cell death in leukemia while sparing normal hematopoietic stem cells. *J Clin Invest* **132**(2022).
4. Asante, Y. *et al.* PAX3-FOXO1 uses its activation domain to recruit CBP/P300 and shape RNA Pol2 cluster distribution. *Nat Commun* **14**, 8361 (2023).
5. Blanco, J.C. *et al.* The histone acetylase PCAF is a nuclear receptor coactivator. *Genes Dev* **12**, 1638-51 (1998).
6. Reutens, A.T. *et al.* Cyclin D1 binds the androgen receptor and regulates hormone-dependent signaling in a p300/CBP-associated factor (P/CAF)-dependent manner. *Mol Endocrinol* **15**, 797-811 (2001).
7. Pettersson, M. & Crews, C.M. PROteolysis TArgeting Chimeras (PROTACs) - Past, present and future. *Drug Discov Today Technol* **31**, 15-27 (2019).

8. Cheng, S., Brzostek, S., Lee, S.R., Hollenberg, A.N. & Balk, S.P. Inhibition of the dihydrotestosterone-activated androgen receptor by nuclear receptor corepressor. *Mol Endocrinol* **16**, 1492-501 (2002).
9. Mottis, A., Mouchiroud, L. & Auwerx, J. Emerging roles of the corepressors NCoR1 and SMRT in homeostasis. *Genes Dev* **27**, 819-35 (2013).